# NADPH Oxidase 3: Beyond the Inner Ear

**DOI:** 10.3390/antiox13020219

**Published:** 2024-02-08

**Authors:** Marc Herb

**Affiliations:** 1Institute for Medical Microbiology, Immunology and Hygiene, Faculty of Medicine, University Hospital Cologne, University of Cologne, 50935 Cologne, Germany; marc.herb@uk-koeln.de; 2German Centre for Infection Research, Partner Site Bonn-Cologne, 50931 Cologne, Germany; 3Cologne Cluster of Excellence on Cellular Stress Responses in Aging-Associated Diseases (CECAD), 50931 Cologne, Germany

**Keywords:** NADPH oxidase, Nox3, reactive oxygen species, oxidative stress, inner ear, vestibular system, cochlea, ototoxicity, lung diseases, cardiovascular diseases

## Abstract

Reactive oxygen species (ROS) were formerly known as mere byproducts of metabolism with damaging effects on cellular structures. The discovery and description of NADPH oxidases (Nox) as a whole enzyme family that only produce this harmful group of molecules was surprising. After intensive research, seven Nox isoforms were discovered, described and extensively studied. Among them, the NADPH oxidase 3 is the perhaps most underrated Nox isoform, since it was firstly discovered in the inner ear. This stigma of Nox3 as “being only expressed in the inner ear” was also used by me several times. Therefore, the question arose whether this sentence is still valid or even usable. To this end, this review solely focuses on Nox3 and summarizes its discovery, the structural components, the activating and regulating factors, the expression in cells, tissues and organs, as well as the beneficial and detrimental effects of Nox3-mediated ROS production on body functions. Furthermore, the involvement of Nox3-derived ROS in diseases progression and, accordingly, as a potential target for disease treatment, will be discussed.

## 1. NADPH Oxidase 3

### 1.1. The Family of NADPH Oxidases

Reactive oxygen species (ROS) were once described as mere byproducts of metabolism and as an unavoidable harming effect that cells have to cope with [1,2,3,4,5]. ROS is the summative term for a group of molecules that all contain oxygen but show more reactivity toward biological molecules than molecular oxygen [6,7,8]. A few early studies have observed beneficial effects of ROS during egg fertilization processes, but only suggestions for ROS sources were made [9,10,11]. Intriguingly, the discovery of an enzyme family present in nearly every form of life [12,13,14,15,16], including bacteria [17,18,19,20], mammals [13,21,22,23,24,25,26,27,28,29,30,31,32], fish [33,34], insects [35], plants [36,37,38], fungi [39,40,41,42] and worms [43], namely, the family of NADPH-dependent oxidases (Nox) was a surprise. This is because the sole function of this enzyme family members is the production of ROS [6,22,44], or to be precise, superoxide (O_2_^−^) [21,45,46,47,48,49], which was associated only with detrimental effects on cellular structures at the time. The first described family member was Nox2 [50,51,52,53,54], also named gp91^phox^ (after its main subunit) or phagocyte NADPH oxidase [55,56,57] (after the most representative cell type, where it is expressed). Nox2 is responsible for the respiratory burst in phagocytes [55,58,59,60,61,62], and the ROS produced inside the phagosome of macrophages, neutrophils and monocytes fell back into the same functional role as before, i.e., a destructive or at least inactivating factor, but inside the phagosome at least directed to a specific target, which is the engulfed pathogen [45,63,64,65,66,67,68,69,70,71]. It is noteworthy that pathogen engulfment as process is not the sole pre-requisite for the respiratory burst. Phagocytosis itself triggers ROS production into the phagosome, nevertheless with varying intensities in dependency of the cargo [72,73,74]. Furthermore, Nox2 is not restricted to phagocytes, but also found in other cells and tissues [75,76,77,78]. In a short time, other Nox family members were discovered, and the enzyme family consists of seven members to date [6,13,79,80], namely Nox1 [81,82], Nox2 [83], Nox3 [84], Nox4 [52,85,86], Nox5 [52,87] and the two Dual oxidases (Duox), Duox1 and Duox2 [88,89,90,91,92]. It became quickly evident that the other Nox family members were either not only present or even absent in phagocytes, but likewise expressed in cells of the adaptive immune system [93,94] as well as in non-immune cells [22,48,66,86,89,95,96,97,98,99,100,101,102,103,104]. ROS production in non-phagocytes has more subtle functions [89,105,106,107,108] in contrast to the vast amounts of ROS (~2 nmol/min per 10^6^ human cells) [55,62,109,110] that are produced in the phagosome during the respiratory burst [22,25,70,111]. These effects of ROS, which strongly diverge from their destructive power in the phagosome, regulate many important processes, such as cell growth and transformation [36,81,92,112,113,114,115,116,117,118,119,120,121,122], angiogenesis [123,124], vasodilatation [125,126,127,128], hormone synthesis [129], tissue remodeling [130], signaling transduction [31,131,132,133,134,135,136,137,138] neuronal development [139,140,141,142], and the list is continuously expanding [80,89,143,144,145,146]. Notably, in addition to their various functions, all isoforms are involved in immune responses during pathogenic invasion [35,147,148]. While oxidative stress describes an imbalance of the cellular redox status in general, the beneficial effects of an oxidative milieu, as listed before, are summarized under the term “oxidative eustress” [7,8]. Of course, when ROS production occurs in an uncontrolled manner or in the wrong subcellular location [149,150,151], a phenomenon termed oxidative distress [7,8], it leads, independently of the ROS source, to cell-, tissue- and organ damage or death [152,153,154,155,156,157,158,159]. Oxidative distress can finally contribute to the development of diseases, such as atherosclerosis [160,161]; cardiovascular diseases [100,108,162,163,164,165,166,167,168,169,170], such as stroke [130,171,172,173,174,175,176] or diabetes [177,178]; cancer [113,122,179,180,181,182,183,184] and neurodegenerative diseases [98,185,186,187,188,189].

#### Structural Components of Nox Enzymes

Since Nox3 is a remarkable exception concerning the usage of Nox-related subunits, a general overview, which covers the similarities and differences of the Nox isoforms, is necessary and will support a better understanding of the latter parts of this review (Figure 1). All Nox family members share a membrane-bound catalytic core structure, a glycoprotein consisting of six trans-membrane α-helical domains (the actual gp91^phox^ in Nox2), which contains two conserved heme groups near the N-terminus [109,190,191,192]. This core component is synthetized as a 65-kilo Dalton (kDa) precursor protein in the endoplasmatic reticulum (ER) [193] and gains its name-giving molecular weight of 91 kDa after heavy glycosylation during the transport through the Golgi network [194,195,196,197]. All Nox core structures end in a long cytosolic C-terminal tail, where the FAD- and NADPH-binding regions are located [198,199]. The gp91^phox^ core unit forms a heterodimer with the membrane-bound protein p22^phox^ [50,51,194,195,197,200,201] called *b*_558_ when fully assembled [51,53,54,202,203,204]. The heterodimer was named after the characteristic spectrum peak at 558 nanometers (nm) [53,202,205,206]. p22^phox^ is an integral part of the Nox family members Nox1-4 [16,96,207,208] but is absent in Nox5, Duox1 and Duox2 [16]. Structurally, p22^phox^ consists of four trans-membrane α-helices [209,210] and a proline-rich cytosolic region, which functions as docking site for other cytosolic adaptor subunits for the Nox enzymes [16,211]. The core subunit p22^phox^ does not only serve as docking site for the cytosolic adaptor subunits of the Nox enzymes [207,212,213] but also has crucial functions for the flavocytochrome *b*_558_ core complex of Nox1-4 itself [27,207,214]. It has an important effect on gp91^phox^ stabilization and loss of p22^phox^ leads to retention of gp91^phox^ in the ER [193,194,195,215,216,217]. p22^phox^ further mediates the localization of gp91^phox^ to cellular membranes in general [217,218,219] and the localization to the plasma membrane in particular [16,217,220]. While p22^phox^ is not essential for all Nox isoforms, the subunit gp91^phox^ represents the obligatory core component for all Nox enzymes [221], which contains the electron-shuttling apparatus. Electrons are transported from NADPH to FAD through the heme-containing domains and react with molecular oxygen to O_2_^−^ [13,222] (Figure 1).

If and what adaptor proteins are necessary for enhanced activation or basal enzymatic activity greatly vary between the Nox isoforms [55,83,172,223,224,225]. Nox2 is activated by the cytosolic regulatory proteins p47^phox^ [215,226,227,228], p67^phox^ [227,228,229,230,231] and p40^phox^ [232,233,234,235]. Nox1 utilizes NADPH oxidase organizer 1 (NOXO1) [25,236] and NADPH oxidase activator 1 (NOXA1) [25,236]. Furthermore, Nox1 and Nox2 both need the Ras-related C3 botulinum toxin substrate (Rac) enzymes, small guanosine triphosphate phosphohydrolases (GTPases), for enzymatic activity [96,237,238,239,240] (Figure 1).

p47^phox^ is not active in unstimulated cells due to its auto-inhibitory region [212,241,242]. However, after stimulation (e.g., by pathogenic or chemical molecules), p47^phox^ is phosphorylated on several serine residues [212,243,244,245,246,247] and binds to p22^phox^ [211,213,246,248,249,250,251,252]. The kinase enzyme responsible for p47^phox^ phosphorylation can be one of the various isoforms of the Protein kinase C (PKC) family. Which PKC isoform is activated depends on the stimulus, but p47^phox^-dependent Nox activation was discovered for PKCβ [253,254] and PKCζ [255,256]. p47^phox^ phosphorylation leads to a conformational change, which allows its binding to p22^phox^ [213,248,249,250,257]. In the cytosol, p47^phox^ and p67^phox^ are already tethered together via tail-to-tail interactions [252,258,259,260,261] and recruited to the plasma membrane/phagosomal membrane-residing flavocytochrome *b*_558_ complex [257,260,262,263,264]. p67^phox^ is critical for the oxidase activity itself, since it regulates the electron flow from FAD to the two heme groups [232]. p67^phox^ also binds directly to Rac, therefore facilitating its transport to the plasma membrane [255,265,266]. Similar to p47^phox^, p67^phox^ is phosphorylated on many sites by various agonists, e.g., by members of the mitogen-activated protein kinases (MAPK) like p38 and extracellular signal-regulated kinase 1/2 (ERK1/2) [267,268,269,270]. While the direct activation of Nox2 strictly depends on p67^phox^, the recruitment of p67^phox^ to the p22^phox^-gp91^phox^ heterodimer is completely dependent on p47^phox^. In the absence of p47^phox^, p67^phox^ will not translocate to the heterodimer and Nox2 is not activated [271,272]. Therefore, p47^phox^ serves as important recruitment unit for the other subunits (Figure 1).

p40^phox^ [258,273] and Rac [274,275], which beforehand transits into its active GTP-bound state [96,237,239,240,276,277,278,279,280], are both likewise recruited to the forming active Nox complex together with p47^phox^. While p67^phox^ and NOXA1 are termed activator subunits [222,281,282] and p47^phox^ and NOXO1 are regarded as organizer subunits (for Nox2 and Nox1/Nox3, respectively) [79,222,245,248,252,283,284,285], p40^phox^ serves as scaffold-like platform, facilitating the translocation of the other subunits at least in the case of Nox2 [236,284]. While p40^phox^-mediated scaffolding is not essential for Nox2 enzyme activation per se [286,287], the involvement of p40^phox^ leads to a two-fold increased ROS production [236] by facilitating the recruitment of p67^phox^ to the plasma membrane [236] and phagosomal membrane [274,288,289].

The Rac proteins, specifically Rac1 (in human monocytes and macrophages) [290,291,292] and Rac2 (in human neutrophils) [293,294,295,296,297,298,299], are crucial for the activation of Nox1 and Nox2 [238,265,293,296,300,301,302], while they are completely dispensable for Nox4, Nox5 and the Duox enzymes [218,303,304]. The Rac enzymes serve two purposes in the context of Nox activation [83,290,305,306]. Firstly, they physically tether p67^phox^ [265,302,307,308,309] to the plasma membrane [280] and the cytochrome b_558_; [310,311], and secondly, they induce a conformational shift of p67^phox^, thereby inducing its activation [231,232,265,302,312,313,314,315,316]. Initial experiments in cell-free systems have suggested a complete subunit-independent translocation of Rac enzymes to gp91^phox^ [96,273,274,290,311]; however, it was later shown that Rac enzymes at least interact with and support the translocation of the subunits p67^phox^ [266,307,312] 2004) and NOXA1 [274,300]. For further reading about the complex topic of the different Rac isoforms and their specific roles during Nox activation, I redirect the interested readers to other excellent reviews [275,317].

The accepted most current model for the activation of Nox2 as most representative isoform depicts as follows: The adaptor proteins for Nox2, namely p67^phox^ and p47^phox^, exist in the cytosol as preformed complexes together with p40^phox^ [234,318], which functions as linchpin between these two subunits [234,319,320,321]. After phosphorylation, the SH3 region of p47^phox^ is exposed to the cytosol and binds to p22^phox^ [246,249,252,271,283]. Since p67^phox^ is tethered to p47^phox^, the heterodimer translocates together to p22^phox^ [236,259,260,261,322,323], where p67^phox^ also associates with gp91^phox^ [232,255,312,314,315,324]. Hence, it is reasonable that no localization of this complex near the plasma membrane is observed when either gp91^phox^ or p22^phox^ is missing [234,273,319,320,321,325]. After dissociation of its inhibitory factor Rho GDP-dissociation inhibitor (RhoGDI) [238,311,326], Rac translocates to the membrane, where it binds to p67^phox^ [265,266,307] and to the flavocytochrome b_556_ core complex [327]. p47^phox^ as well as Rac are not absolutely essential for Nox2 activation [310,328], but play the role of important support units. They bind and orientate p67^phox^ for optimal electron flow and activation of the Nox2 complex [24,109,232,312,314]. Nox1 is activated in a similar manner, however by utilizing its unique organizer and adaptor subunit NOXO1 and NOXA1, respectively (Figure 1). It is noteworthy that Nox1, while also being dependent on Rac for activation, cannot utilize the Nox2-related subunits p47^phox^ and p67^phox^ for functioning (Figure 1).

After full assembly, the Nox complex transfers electrons as hydride ions (H^−^) from NADPH to FAD. This step is mediated by the recruitment the p67^phox^ subunit [231,232,255,329,330]. From FAD, the electrons are shuttled by the two heme molecules through the membrane-spanning part of the complex [109,331]. On the other site of the membrane, the electrons are transferred to molecular oxygen and form O_2_^−^ [24,200,207,218]. So far, only the release of H_2_O_2_ instead of O_2_^−^ has been clearly proven for Nox4 [332,333], while O_2_^−^ is still the first-generated ROS subspecies at the Nox4 enzyme [334,335,336]. Nox4 contains a special E-loop on the extracellular site, which slows down the diffusion of O_2_^−^ until it is dismutated to H_2_O_2_ [220,333]. Nox4 is a unique isoform in terms of regulation since no stimuli or regulatory subunits are necessary to directly induce Nox4 activity [218,337]. Nox4 is defined as being permanently active, as long as p22^phox^ for the complete core structure is present [84,218,220,334,338] (Figure 1). The major adaptation for Nox4-derived ROS production is achieved by degradation- or new expression-induced various stimuli or stress conditions [339,340,341,342,343,344,345]. Nevertheless, some stimuli, like insulin [346] or LPS [347], can quickly trigger Nox4-mediated ROS production, which cannot be explained by expression of the protein itself. Accordingly, a few years after its discovery, polymerase (DNA-directed) delta-interacting protein 2 (Poldip2) [348,349] was identified as positive regulator, which directly binds to p22^phox^ and increases Nox4-mediated ROS production [350,351] (Figure 1). Some other regulating proteins, e.g. Toll-like receptor (TLR) 4 [347,352] or protein disulfate isomerase [353], were identified, slowly revising the view of Nox4 as not being regulated by other factors except its expression [85]. Nox5, Duox1 and Duox2 have EF-hand domain-containing extensions on the cytosolic N-terminus, which bind Ca^2+^ [87,88,354] (Figure 1). Indeed, Ca^2+^ is the main activating factor for ROS production of these three Nox family members [87,88,355]. Additional adaptor proteins, Dual Oxidase Maturation Factor 1/2 (DuoxA1/2), were identified as factors necessary for maturation of Duox1/2 [356,357]. Duox1/2 also contains an additional peroxidase-like domain that extrudes to the extracellular site [88,109,358]. However, so far, only the Duox isoform of *Drosophila melanogaster* has shown an active peroxidase function of this domain, similar to the myeloperoxidase reaction The Duox isoform processes the produced H_2_O_2_ to generate hypochlorous acid (HOCl) [35]. Since this review summarizes new and old findings of Nox3, the reader is directed to other excellent reviews about Nox enzymes in general and in detail [13,21,22,23,24,25,26,27,28,80,359,360,361].

### 1.2. Nox3: Structure and Subunits

Nox3 combines many features of Nox1, Nox2 and Nox4 in terms of basal activation and regulatory subunit involvement, which is unique among the Nox enzymes. However, it took some years of intensive research to shed light on this most flexible Nox isoform. Nox3 was discovered together with Nox4 and Nox5 during a genetic screen in search for homologs of the firstly discovered Nox2, or more precisely, for proteins similar to the membrane-bound subunit gp91^phox^ [362,363]. This research field was intensively investigated after the discovery that non-phagocytic cells also produce ROS and that phagocytes are not the only cells capable of this process. After Nox3, Nox4 and Nox5 were identified, Nox1 was cloned and described by Suh and colleagues [81], followed by Duox1 and Duox2 [88,109,364], therefore completing the enzyme family. The *NOX3* gene is located on chromosome 6 in humans (gene locus 6q25.3), and suggestions were made that this Nox isoform appeared after the emergence of fish and amphibians [16]. The protein structure of Nox3 is very similar to Nox1, Nox2 and Nox4. Indeed, Nox3, which consists of 568 amino acids (aa), shows the strongest sequence similarity with gp91^phox^ (58%) [109,365]. Initially, Nox3 was only weakly detected in human fetal kidney and the placenta [52,363], and further research related to Nox3 was dampened afterwards. New insights, like the predominant tissue locations or exact protein structures of Nox4 [84,334], Nox5 [87,354,366] and Duox1/2 [43,367,368], were unraveled shortly after their identification [81,82]. In contrast, it took 3 years until Nox3-focused research achieved a new momentum, mainly by three studies from the labs of Prof. Lambeth and Prof. Krause [355,369,370]. Nox3 shares many similarities with Nox2 concerning their protein structure (like the dependency on p22^phox^) and regulatory subunits (i.e., p47^phox^/NOXA1 and p67^phox^/NOXO1). Initial experiments of several research groups, which exclusively focused on Nox research, delivered the first observations where, if and how Nox3 is located and activated. All of these studies did not investigate ex vivo cells but instead used human cancer cell lines and co-expression approaches to combine the Nox3 core protein and various Nox subunits. Most of the findings are consistent between these initial studies, but some differences emerged back then, probably due to differences in the used culture cells. These differences were quite at the awareness of the researchers and discussed in the community back then [371]. Nevertheless, these first studies gained impactful insights into the Nox3 protein and its regulation, which will be discussed now.

#### 1.2.1. Adaptor Subunits of Nox3

In 2004, two research groups investigated and published findings regarding the regulation of Nox3 nearly simultaneously [355,372]. Cheng and colleagues from the Lambeth lab used in vitro experiments, in which different combinations of human Nox3 (and Nox2/gp91^phox^ and Nox1) and different Nox adaptor proteins were expressed in HEK293-H cells and COS-7 cells as “experimental vessels” [369]. They investigated the subunits associated with Nox2 and Nox1 at basal conditions and after stimulation with Phorbol 12-myristate 13-acetate (PMA). This chemical stimulates association of the Nox subunits by activation of the PKC, which results in robust ROS production [73,74,373,374]. Cheng et al. found that p67^phox^ alone was not sufficient for basal or PMA-stimulated ROS production, while the expression of p47^phox^ was sufficient for moderate ROS production by Nox3. Notably, this ROS production could not be further increased by PMA treatment. The combined presence of p47^phox^ and p67^phox^ led to the highest ROS production, which could be further increased by PMA stimulation. The presence or absence of Rac did not change the activation rate of Nox3. Interestingly, the expression of the Nox1 subunit NOXO1 also led to a strong activation of Nox3, which could not be further increased with PMA. In contrast, NOXA1 only slightly induced ROS production. Combinatory expression of adaptor proteins either for Nox2 (p47^phox^, p67^phox^) or Nox1 (NOXO1, NOXA1) led to maximal ROS output of Nox3. NOXO1 in combination with p67^phox^ showed only minimally increased ROS production in comparison to sole NOXO1 presence. The combined expression of NOXA1 and p47^phox^ in this system only led to PMA-dependent activation of Nox3. These data nicely showed that Nox3 is much more flexible than Nox1 or Nox2. While Nox2 strictly needs both p47^phox^ and p67^phox^ for activation, p47^phox^ alone leads to moderate Nox3-derivedderived ROS production. Nox1, on the other hand, needs NOXO1 and NOXA1 for activation, while NOXO1 alone induced strong ROS production together with Nox3. Combinations of different adaptor proteins (e.g., NOXO1/p67^phox^ or NOXA1/p47^phox^) only resulted in the low activation of Nox1 and Nox2, but induced a strong ROS production of Nox3. Taken together, these first experiments revealed the high flexibility of Nox3 in terms of adaptor protein usage (Figure 2).

Banfi and colleagues confirmed the flexibility of Nox3 in terms of adaptor protein utilization. The group analyzed mouse tissue samples via qRT-PCR and used histological staining to investigate the localization of Nox3 and the different subunits for the first time in vivo [355]. They detected mRNA expression of Nox3, NOXA1/p47^phox^ and, to a lesser extent, also NOXO1/p67^phox^ in the inner ear of mice. They followed a similar approach as Cheng and colleagues and analyzed the molecular function and regulation of Nox3 by a co-expression system in HEK293-H cells as “empty vessels”, which are devoid of Nox3. They confirmed ROS production by Nox3 in the complete absence of any adaptor subunit. In contrast to the findings of Cheng and colleagues, the group saw that p47^phox^ or NOXA1 alone are not sufficient to increase Nox3-derived ROS production. Similar to Cheng and colleagues, they measured the strongest increase in ROS production, when either NOXO1 and NOXA1 or when NOXO1 and p67^phox^ were expressed together. This Nox3-mediated ROS production could not be further enhanced by PMA stimulation. The combination of p47^phox^ and p67^phox^ or the combination of p47^phox^ and NOXA1 resulted in a robust PMA-induced ROS increase, while basal ROS production was only minimally increased [355]. The most obvious discrepancy between the two studies from Cheng et al. and Banfi et al. was the Nox3-derived ROS production with NOXO1 as sole subunit. In the study from Banfi et al., only a small increase in ROS production was measured after PMA stimulation [355], while Cheng and colleagues described a strong increase in basal ROS production, which could not further be enhanced by PMA [369]. One has to consider that both studies did not use ex vivo cells of any kind, in which the actual Nox3 is present and active. It was and is common practice to use artificial co-expression systems in cancer cell lines to obtain initial insights into protein function. Of course, with varying cell lines, the experimental outcomes can also differ, which explains the contrasting results of the two groups. Nevertheless, it is impressive that both studies gathered the mostly similar results for Nox3, which delivered important first hints of the regulation (and location) of this Nox family member. The flexibility of adaptor unit utilization by Nox3 is especially distinct from other Nox family members (Figure 2). The Lambeth lab later discussed that the potential or non-potential adaptability of different Nox enzymes reflect their functions and tissue locations [370]. In contrast to phagocytes, for non-immune tissue cells, it may be more biologically relevant to have a redundant subunit protein backup in case of gene mutations or deletions to keep the Nox enzyme functional and suppress disease development in case one or more regulatory Nox subunits are altered. Nox3, with its flexibility, is a shining example for this statement.

Takeya and colleagues analyzed and compared different splicing variants of the *NOXO1* gene (termed α, β, Δ, γ) and its subsequent protein products [96,229,367] in the context of Nox3 activation. While the most abundant variant NOXO1β was investigated before [369], Takeya et al. reported that also the splicing variant NOXO1γ was sufficient to induce Nox3-mediated ROS production [371]. They also described a strict dependency on the p22^phox^ subunit for NOXO1γ-mediated activation of Nox3. Furthermore, they showed that binding of either NOXO1β or NOXO1γ to phosphatidylserins in the plasma membrane is mediated by a specific amino acid sequence, called the PX motif. This PX motif is required for membrane binding of Nox subunits in general [237,369,375,376,377]. However, a basal ROS production by Nox3 was detected even after destruction of the PX domains in the NOXO1 subunits [371]. Following studies intensified the details of the interplay between Nox3 and its various subunits. Miyana and colleagues observed slightly enhanced Nox3 activation by p67^phox^ alone in HeLa, CHO and COS-7 cells and detected enhancement with NOXA1 in CHO, COS-7, but not in HeLa cells [378]. Maehara et al. showed that the SH3 domain of p67^phox^, which is necessary for the full activation of Nox2 [379], is interestingly not needed for Nox3 activation [380]. In a follow-up study, the group further identified a highly conserved activation domain of p67^phox^ (in the aa 190–210), which is crucial for activation of Nox1, Nox2 and Nox3. The identified crucial residues were Tyrosine 198, Leucin 199, Valin 204, Leucin 193 and Asparagin 197 [222]. However, this domain did not have any influence when Nox3 was activated by p67^phox^ together with p47^phox^, explaining the previous observation that p67phox and its SH3 domain alone are of no significance for Nox3 activation [380]. Taura et al. investigated the SH3 domain of p47^phox^ and NOXO1 and observed that the domain is important for full Nox3 activation after PMA-stimulation [381]. This occurred even in the absence of p67^phox^, therefore arranging NOXO1 and p47^phox^ above p67^phox^ in the hierarchy of Nox3 activation. The group further identified the amino residue Interleukin 152 in a short N-terminal tandem region in the SH3 region of p47^phox^. This residue was found to be crucial for the activation of p47^phox^-mediated activation of Nox3, even in the absence of p67^phox^. Notably, the residue was also found in NOXO1 [381].

#### 1.2.2. The Nox3-p22^phox^ Complex

While the interactions of Nox3 and the diverse Nox subunits obtained some new and fascinating insights during that time, Ueno and colleagues investigated the interplay of the other membrane-bound subunit of the Nox1/2/3/4 complex namely p22^phox^ [382]. Ueno et al. also used co-expressing systems in COS-7, CHO and HEK293-H cells to investigate this topic. Most importantly, they demonstrated, for the first time, that Nox3 physically interacts with p22^phox^ and that both the presence and the interaction of ph22^phox^ are essential for Nox3 activity. They detected minimal, but constitutive ROS production in the absence of any adaptor subunit, confirming previous findings [355,372] (Figure 2). They also measured enhanced ROS production after co-expression of Nox3 with p47^phox^, which could be further enhanced by combined expression with p67^phox^. Like Cheng et al., but in contrast to Banfi et al., the group found that the sole expression of NOXO1 leads to a strongly enhanced ROS production independently of PMA stimulation. Interestingly, they could show that NOXO1 is constitutively bound to p22^phox^, which explains the PMA-independent increase in ROS production. NOXA1 alone only slightly increased ROS production like seen before [355,372]. In contrast, the p47^phox^-mediated Nox3 activation could be further increased by PMA stimulation. The group also described no necessity for Rac proteins in Nox3 activation as reported previously [370].

Kawahara et al. not only characterized the role of p22^phox^ during activation of Nox1-5 in co-transfected HEK293 cells, but also investigated the interaction of p22^phox^ with various adaptor subunits [337]. They saw that Nox3 activation was strongly diminished in p22^phox^-silenced HEK293 cells despite the co-transfection of any subunit confirming the crucial role for p22^phox^ in Nox3 basal activation. Notably, they also saw a strong activation of Nox3 with NOXO1 or p67^phox^ alone, a minor but detectable activation with NOXA1 alone and the strongest activation after combined transfection with either NOXO1/NOXA1 or p67^phox^/p47^phox^. In addition, Kawahara and colleagues generated a C-terminally-truncated p22^phox^ protein, which still formed a complex with gp91^phox^, but could not bind to the organizer subunits NOXO1/p47^phox^. HEK293 cells, which co-expressed this truncated p22^phox^ protein together with NOXO1, showed a strongly diminished activity, suggesting that p22^phox^ has not only a direct stabilizing effect for Nox3, but is also important for binding of organizer subunits and subsequent Nox3 activation. As shown by two groups before [370,372], Nox3, in contrast to Nox1 and Nox2, has the remarkable ability to be activated only by the presence of an activator subunit (p67^phox^, NOXA1) without any organizer subunit (p47^phox^, NOXO1). This phenomenon is also unique among the Nox enzyme family members (Figure 2) Interestingly, during additional co-expression of NOXA1 together with Nox3, NOXO1 and the truncated p22^phox^ protein, the group observed a nearly restored activity of Nox3. These findings indicated the possibility that Nox3, as exclusive exception in the Nox enzyme family, might be able to bypass the p22^phox^-mediated binding of the organizer subunits NOXO1/p47^phox^ by the direct binding of an activator or organizer protein to the Nox3 core structure itself. This again demonstrates the fascinating flexibility of Nox3 [337]. Nakano et al. characterized the role of p22^phox^ for the actual biosynthesis of Nox3 in co-expression systems with HEK238 and CHO cells [217]. They firstly described the characteristic spectrum peak at 558 nm for Nox3, which suggested that the structure of Nox3 probably resembles Nox2. Chemical inhibition of heme synthesis in HEK293 cells transfected with Nox3 and p22^phox^ resulted in a completely blunted ROS production demonstrating that heme is crucial for Nox3 functioning. In vitro translation with cDNA in a rabbit reticulocyte lysate system resulted in a 53 kDa-sized protein product, which was the first size description of Nox3. This product further underwent N-linked glycosylation as previously discovered for gp91^phox^ [197] and knock-down of p22^phox^ via small interfering (si)RNA resulted in reduced ROS production, as described previously [337]. The group also observed that p22^phox^ was crucial for plasma membrane targeting of Nox3, which remained diffusely distributed in the cytosol in the absence of p22^phox^, which was also in line with previous observations [274,337,382]. In addition, the study confirmed other previous results concerning NOXO1 and p22^phox^, i.e., that NOXO1 interacts with p22^phox^ at the plasma membrane [274,337,382] but does not necessarily directly bind to it [372,383]. Miyano and colleagues confirmed the N-glycosylation and p22^phox^-dependent maturation of Nox3 in CHO cells [384], hence completing the picture of the interplay between Nox3 and p22^phox^.

#### 1.2.3. Nox3 and Rac

Previous observations of Rac-independent activation of Nox3 [370,382] were revised and challenged by a study from Ueyama et al., which investigated this topic by a co-expression system in HEK293-H and CHO-K1 cells [274]. The group could confirm many of the previous findings, like constitutive Nox3 activity without any subunit and maximal activity enhancement with NOXO1 alone or with p47^phox^/p67^phox^ combined [96,355,372,382]. Interestingly, the authors described a strong increase of ROS production after co-expression of Nox3 and Rac1 alone, which is in contrast to previous results. Additional expression of p67^phox^ further enhanced the activity of Nox3. This study confirmed again the flexible use of Nox activator or organizer proteins and underlines the observation that Nox3 does not strictly need organizer and adaptor units for moderate ROS production. Accordingly, Miyano and colleagues revised their findings concerning Rac dependency for Nox3 activation of their previous study in 2005 [382]. In co-expression experiments with various cancer cell lines (HeLa, CHO, COS-7), they observed a small p67^phox^/NOXA1-dependent enhancement of Nox3 activation by Rac. They further showed that p47^phox^, either in combination with p67^phox^ or NOXA1 was necessary for maximal Nox3 activation. In contrast, the combined expression of NOXO1 and either p67^phox^ or NOXA1 showed no dependency on Rac. Moreover, Nox3 was activated even more strongly when Rac binding was inhibited by site mutation of p67^phox^ or NOXA1 [378]. The group next focused on the role of all Rac isoforms (Rac1-3) in respect to Nox1 and Nox3 activation [275]. For that again a co-expression systems with HeLa and HEK293 cells was used. In addition, human neutrophil fractions and the macrophage-like cancer cell line RAW246.7 were analyzed to investigate this topic. The group could confirm previous results of the Rac1-dependent enhancement of ROS production mediated by p67^phox^ or NOXA1 [274,378] and showed that Rac2 and Rac3 can function redundantly in this process. For this reason, Rac is not a crucial component for Nox3-derived ROS production but can enhance Nox3 enzyme activity in combination with a defined set of subunits (Figure 2).

A complete new set of organizer subunits, not only for Nox3, but also for Nox1, was identified by Gianni and coworkers [385]. They investigated a possible role of the two Tyrosine kinases (tyrosine kinase substrate with five SH3 domains), Tsk4 and Tsk5 [386,387,388], for Nox enzyme activation. Co-expression of Nox3 and Tsk4/5 alone increased the ROS production by Nox3 in HEK293 cells, similar to NOXO1 or p47^phox^. Unfortunately, while also a role in Nox4 activation was discovered [353,389], to date, no further investigation of these new interesting subunits and Nox3 was conducted.

Taken together, the heterodimer consisting of Nox3 and p22^phox^ is the basic minimal structure independent of the investigated species and alone is sufficient to produce substantial amounts of ROS without any adaptor subunit. Both p22^phox^ and Nox3 depend on each other for proper maturation in the ER and for plasma membrane translocation, and either one is degraded when the other part is missing. Similar to the flavocytochrome b_558_ of Nox2, the Nox3-p22^phox^ heterodimer shows the characteristic spectrum band at 558 nm, contains the two essential heme groups and undergoes heavy glycosylation during maturation. However, in contrast to Nox2 or Nox1, if and to what extent the basal ROS production of Nox3 can be enhanced highly depends on the regulatory subunit set and the investigated species [172,390]. In human cells, the basic ROS-producing activity of Nox3 can be enhanced by p47^phox^, NOXA1 or NOXO1 alone, with NOXO1 showing the strongest effect [391]. The ROS production can be maximized by presence of both NOXO1 and NOXA1 or p47^phox^ and p67^phox^. p67^phox^ alone is not sufficient for ROS production enhancement of Nox3 and either needs Rac1 or p47^phox^ in addition. In mice, NOXA1, but not NOXO1 alone, is sufficient to slightly enhance the basal ROS production, however in dependency on Rac. A combination of NOXA1 and NOXO1 leads to maximal ROS production. p67^phox^-mediated enhancement of ROS production can be further increased by Rac, however not when both p67^phox^ and p47^phox^ are present. In summary, Nox3 is unique among all Nox isoforms since it can utilize all Nox organizer or activator subunits or a combination of them, while also showing basal activity like Nox4 (Figure 2). As a critical side note, all described findings and molecular interactions of Nox3 and its adaptor subunits have not been investigated to date in any ex vivo cell type that naturally expresses these proteins.

## 2. Location of Nox3

### 2.1. Location of Nox3 in Organs and Tissues

While often cited in many articles [12,13,172,285,361,392], including previous articles from our own lab [393,394], as “only/exclusively expressed in/restricted to the inner ear”, Nox3 was detected in various organs and cell types over time. This restriction of Nox3 presence to the inner ear, the continuous transmission through the literature and the subsequent underestimation of influence on its cellular processes were due to the first in vivo investigation in animals that lacked Nox3. The animals showed a remarkable head-tilting phenotype [370], and the inner ear as a localization of Nox3 was swiftly discovered [355]. However, several years of excellent research on Nox3 revealed other important locations of Nox3, for example the lung [395] and the liver [396]. Nevertheless, this review will firstly focus on the inner ear as the first location where Nox3 was discovered in vivo and then continues with a broad overview of organs and cell types where Nox3 could or could not be detected.

#### 2.1.1. Nox3 in the Inner Ear

The inner ear of mammals provides two crucial functions for the orientation of the organism, namely the sensation of sound and the sensation of balance and orientation [397,398]. While the cochlea is responsible for sound processing [399,400], the vestibular system maintains balance and orientation [401,402,403]. Paffenholz and colleagues discovered the probably most intriguing phenotype for Nox3, since Nox3-deficient mice showed a strong head-tilting behavior, targeting the inner ear as only research focus for Nox3 for some years [370]. Since this review focuses on Nox3, descriptions of the vestibular system and cochlea are mandatory at this point. It should be mentioned that as sure as Nox3 is not only expressed in the inner ear, Nox3 is also not the only Nox isoform expressed in the inner ear [404]. Cheng and Lambeth detected the expression of Nox2 and Nox4 besides Nox3 in the murine inner ear, while Nox1, Duox1 and Duox2 were absent [369]. Nox2 is expressed in the microglia, which reside in the spiral ganglion [405,406], while Nox4 is expressed in the vascular endothelium, which also supports the stria vascularis [407]. More importantly, studies that investigate the expression profile of Nox enzymes must carefully distinguish between the vestibular system and the cochlea and not generalize their findings to the whole inner ear.

##### Nox3 in the Vestibular System

In the vestibular system, three semicircular canals and the cristae ampullas form a functional unit to detect and coordinate angular (rotational) acceleration [408]. In the otolith organs (consisting of the saccule and the utricle) the neuroepithelial maculae, a layer of sensory epithelial cells, detect gravity and linear acceleration [409,410]. An extracellular gelatinous matrix is located on top of the maculae and embedded in this matrix layer are crystalline, polymorphic structures called otoconia [411]. The otoconia are formed directly above the sensory hair cells, which are mechanoreceptors that transfer the sensory information to the ganglion cells via chemical synaptic activation. Ganglion cells show discharge patterns in the absence of any stimulation [412,413,414], which are mediated by a steady neurotransmitter release from the pre-synaptic hair cells in a calcium ion (Ca^2+^)-dependent manner [415,416,417,418,419,420]. Otoconia function as solid masses, which are affected by change of gravity or linear acceleration [401,408,410,421]. Calcium carbonate (CaCO_3_) is the main inorganic compound forming the crystalline structure of the otoconia [401,422]. Indeed, the protein Pendrin, a HCO_3_^−^/Cl^−^ exchanger channel, as well as Otopetrin1, a proton channel [423,424], are crucial for proper otoconial formation [425,426,427,428,429]. The otoconia are not completely inorganic, since a number of proteins were identified as organizing or structuring components, and the list of otoconia-relevant genes is expanding [430,431,432,433]. The major proteinaceous component found in the otoconia is otoconin 90/95 (OC-90/95) [434,435,436]. OC-90/95 is a 90–95 kDa-sized glycoprotein that belongs to the family of secretory phospholipases A2 [435,436]. OC90/95 is produced by the non-sensory epithelial cells of the inner ear from where it is secreted into the endolymph [433,434,436,437]. It is necessary for proper formation of the inorganic CaCO_3_ crystallites into the otoconial organic mass [438,439], and OC-90-deficient mice lose nearly 50% of their otoconial structures, leading to imbalance. Importantly, the hearing capabilities remain intact in these animals [438,440]. Moreover, a disturbed longitudinal flow of OC-90 from the vestibule to the endolymphatic sac also leads to otoconial malformation, meaning that not only the presence, but also the location of OC-90/95, is of importance for otoconial formation [441,442]. OC-90 also recruits other proteins, such as Otolin-1 [438,443], a component of the gelatinous matrix. Other examples are Otogelin, which is found in the surrounding extracellular layer [444,445], and Otoancorin, which is located between the sensory hair cells and the overlaying extracellular matrix [446]. The concerted action and coordination of the various inorganic and organic components are necessary for the proper formation of functional otoconia [409,410,431,447,448].

For the investigation of the vestibular system, mice (or other model organisms), which harbor mutations in gene loci that affect the otoconial formation are obviously the most useful [421,449,450,451,452]. Several altered gene loci that led to loss, disturbed size or dislocation of otoconia and, subsequently, to a malfunctioning vestibular system were identified and phenotypically described [408,410,425,447,452,453]. The first-described gene locus associated with the head-tilting phenotype in mice was logically named “Tilted-head” (*thd*) [454]. Unfortunately, besides the phenotypical description of the mice, this locus was not further investigated. In the second detected locus “tilted” (*tlt*), the gene that encodes otopetrin 1 is localized. Otopetrin 1 is also crucial for otoconia development [425,426,427,428]. The analyzed third locus “head tilt” (*het*) containing two mutated alleles, *het* [455] and *het^2J^*, was characterized and both mutated alleles were associated with loss of otoconia [456]. After further characterization [370], this locus was logically renamed *Nox3^het^* [457]. The *Nox3^het−3J^* allele was generated during a mutagenesis project [458] and later investigated and associated with Nox3 by Paffenholz and colleagues [370]. The *Nox3^het−4J^* allele was also generated during a mutagenesis program in C57BL/6J mice [459] and the *Nox3^het−5J^* allele spontaneously appeared at a Jackson Laboratory in the CBySmn.CB17-Prkdcscid/J mouse strain [459].

Paffenholz et al. analyzed some other natural occurring and mutagenesis-induced mutated alleles in the *het* locus, which were named *het^R96^*, *het^R542^* and *het^3J^* [370]. Several affected genes were identified, one of them with a high homology to the previously described human NADPH oxidase 3 gene *NOX3* [52,363]. The *het^R96^* mutant allele resulted in a Nox3 protein, which lacked three of the trans-membrane α-helices, a complete catalytic domain and the binding sites for NADPH and FAD (see Section 1.2 and Figure 1). Also, a region responsible for heme binding was disturbed. The homologous deletion of Nox3 manifested itself by an obvious heat-tilting phenotype and lowered motor coordination (i.e., disturbance during balancing and swimming). Notably, while the vestibular system was clearly disturbed in Nox3-deficient mice, the hearing capacity was unaffected, at least in these investigated animals. Histological analysis of the vestibular system in Nox3-deficient mice revealed that the observed phenotype was based on the complete lack of otoconia in homozygous (but not heterozygous) mice throughout the complete lifespan (embryonic stage to adult) [370]. Paffenholz and colleagues described Nox3 as a ROS-producing enzyme in the inner ear that is crucial for the morphogenesis of the otoconia and subsequently for a properly functioning vestibular system. However, at that time, the molecular mechanism of the Nox3-derived ROS, which is responsible for otoconia formation, was pure speculation [370].

A parallel study of Banfi and colleagues also reported Nox3 presence in the inner ear of mice and rats by cloning experiments with cDNA [355]. The group also detected Nox3 expression at low protein levels in the brain, the skull and the fetal kidney. Nox3 expression in the fetal rat kidney was later confirmed by Reinehr and colleagues [460]. The predicted murine amino acid structure showed 81% sequence similarity with the human sequence. The group could also confirm the vestibular system as Nox3-expressing tissue [355,370] and further specified, for the first time, the sub-tissue location, i.e., the non-sensory epithelial cell layer of the saccule, by in situ staining [355].

All of the so far described mutant alleles of Nox3 (namely *Nox3^het^*, *Nox3^het−2J^*, *Nox3^het−3J^*, *Nox3^het−4J^*, *Nox3^het−5J^*, *Nox3^hetR96^* and *Nox3^hetR542^*) lead to otoconial and/or vestibular-evoked potential responses, which can be measured by a non-invasive method developed by Jones et al. as reliable tool to identify loss-of-function mutations for Nox3 [408,461]. The results of these measurements were comparatively analyzed and summarized in the work of Flaherty and colleagues [457] and recommended for further interested readers.

A few years later, Mohri and colleagues generated mice that expressed Nox3 coupled to the red fluorescence tag dtTomato to re-investigate the precise locations of Nox3 in the inner ear in a ground-breaking study for the field [462]. They reported the “tilted head” phenotype and otoconial defects in Nox3-deficient animals as described before [355,370]. Additionally, they observed strong Nox3 protein expression in the endolymphatic sac and duct at early embryonic stages (at day 18.5). However, right after birth and 3 days after birth, only weak Nox3 expression was detected in the semicircular canals and the vestibule. Importantly, the group further showed that Nox3-derived ROS are majorly produced by non-sensor epithelial cells [355], which face the lumen of the endolymphotic sac and duct, as well as the semicircular canals and vestibule. A mechanism of Nox3-derived ROS for otoconial development was not made during this investigation. Together, these studies clearly showed that Nox3 is located in the vestibular system and is crucial for the proper development of the otoconia and, accordingly, for balancing (see Section 4.3.1).

##### Nox3 in the Cochlea

The cochlea is the organ responsible for hearing [399,400,463], and several studies have described Nox3 expression in this area of the inner ear [355,404,462]. Banfi et al. detected expression of Nox3 mRNA in parts of the adult mouse cochlea, precisely the organ of Corti and the spiral ganglia, while Nox3 was not expressed in dorsal root ganglia [355]. However, in contrast to Banfi and colleagues, who analyzed mouse samples, Nox3 was not detected in the spiral ganglion neurons of the rat cochlea [370]. However, while the loss of Nox3 and the correlative deficiency of otoconia is detrimental for balance, head positioning and gravity sensing [370,408,464,465,466], the loss of Nox3 in the cochlea leads to a rather protective outcome for the tissue and the hearing capacity (see Section 5.1). Overproduction or production of ROS in the wrong location can lead to irreversible cell and tissue damage, called oxidative distress [7,8]. This phenomenon was also described in previous studies, which showed that excessive ROS production in the cochlea in general has a great impact on age-, noise- and drug-induced hearing loss (see Section 5.1.1, Section 5.1.2, Section 5.1.3 and Section 5.1.4) [467,468,469,470,471,472,473,474]. Since Nox3 was firstly discovered in the inner ear, it was only reasonable during the time of early Nox3-related research to assume that Nox3 is most probably responsible for the destructive ROS production in the cochlea [355,370]. However, it took several years until this correlation was proven true [462,475,476,477,478]. Similar to Nox isoform expression in the vestibular system, Nox3 is not the only Nox isoform expressed in the cochlea. Vlajkovic and colleagues detected all seven isoforms, Nox1-5 and Duox1-2 in the rat cochlea [479]. The group further investigated the specific cellular expression of the Nox isoforms, which will be discussed later in this review (Section 2.2.1). Mohri and colleagues used their well-established mouse strain, in which Nox3 is coupled to the red fluorescence tag dtTomato [462,480,481]. They detected no Nox3 expression in the cochlea after 1 and 2 months after birth. Nox3 expressions started, at the earliest, after 6 months accompanied by outer hair cell (OHC) loss. Further analysis revealed an increasing Nox3 expression in supporting cells between 1 and 6 months, while OHCs showed no Nox3 expression. This is a ground-breaking study for Nox3-related research, since Mohri and colleagues not only investigated the exact location of Nox3 in the cochlea, but also described its role for different forms of hearing loss, which will be discussed in Section 5.1. This was further completed by Rousset and colleagues who detected expression of Nox2 and Nox3 mRNA in the mouse cochlea, but more importantly, in the human cochlea [404].

#### 2.1.2. Nox3 in Other Organs

Many studies have investigated the topic of Nox3 expression in various organs and tissues. Surprisingly, during years of intensive research, it became clear that Nox3 is present in many organs and cell types with a plethora of different functions, which will be discussed later in Section 4. Unfortunately, most expression data available for Nox3 are restricted to mouse or rat tissue, and information of Nox3 expression patterns in human tissues is scarce.

In addition to the inner ear, Nox3 was detected either as protein or, mostly, as mRNA in mouse lung tissue [482,483], in mouse testes [484], in mouse white adipose tissue [485] and in the mouse upper circumvallate papillary epithelium of the tongue [486]. Nox3 mRNA could not be detected in the naïve mouse fetal or adult liver [487].

In the rat, Nox3 mRNA expression was detected in rat skeletal muscle, testis, lung, prostate, colon [488], brain [488,489,490], spinal cord neurons [491] and the adult rat kidney [492].

In contrast to murine or rat tissue, Nox3 is expressed in the avian liver [493].

The few studies which investigated Nox3 expression in ex vivo human tissue samples have described Nox3 expression in human placental tissue [494], as well as in non-tumor and tumor pancreatic tissue (with no significant differences in dependency of these two settings) [495]; Nox3 expression was detected in the human fetal, but not in the adult kidney [487]. Juhasz et al. investigated the expression of Nox enzymes in various human cancer cell lines and, importantly, in ex vivo tumor tissues [113]. Nox3 mRNA was absent in all isolated tumor tissues derived from the colon, liver, lung, kidney, prostate, stomach, ovary, breast, testis and brain.

### 2.2. Expression of Nox3 in Cell Types

While detection in tissues or whole organs was and is a challenging task, the investigation of Nox3 protein expression in specific cell types, especially in cell lines, was extensively performed and delivered a broad catalogue of data addressing the topic where Nox3 is expressed and where it is absent. I should note that I do not share the opinion of cell lines of cancerous origin as “normal cells” for in vitro investigations as a sole line of evidence. Primary isolated ex vivo cells should be preferred; however, their isolation and cultivation remain difficult. Notably, most of the in vitro studies which addressed Nox3 have used cancer-derived cell lines like HepG2 (as hepatocyte model) or HEI-OI (as an inner ear hair cell model). Therefore, I listed only cancer cells under Section 2.2.7, which were clearly addressed as cancer cells in a context of tumor-associated research.

#### 2.2.1. Nox3 in Cells of the Inner Ear

It is not surprising that the most detailed knowledge of cellular Nox3 expression accumulated around the cells of the inner ear and, as mentioned before, Nox3 is not the only Nox isoform expressed in the inner ear. Vlajkovis and colleagues first described a detailed overview of Nox isoform expression in the rat cochlea [479]. In detail, Nox1 mRNA was found in OHCs and Deiters’ cells; and Nox2 mRNA was expressed in OHCs and Claudius’ cells, Deiters’ cells and inner border cells, but was strongest in inner sulcus cells. Nox3 mRNA was strongly expressed in the inner sulcus cells but only weakly expressed in cells of the organ of Corti. Neither Nox2 nor Nox3 were detected in the lateral wall tissues or spiral ganglion neurons, which was confirmed for Nox3 protein expression by Zuhang et al. [496]. Nox4 was expressed in Hensen’s cells and inner sulcus cells but strongest in the blood vessels of the cochlear lateral wall and the Rosenthal’s canal. Duox1 was only weakly detected in sensory inner hair cells (IHCs) and supporting cells of the organ of Corti. Duox2 was strongly expressed in the inner sulcus cells and weakly expressed in the organ of Corti. The location of Nox3 in inner sulcus cells is especially notable, since these epithelial cells line the endolymphatic compartment where they clear the endolymph from cell debris, which occurs, for example, after severe acoustic trauma [497]. Accordingly, these cells play a pivotal role for cochlear repair and ion homeostasis [498].

Mohri and colleagues analyzed Nox3 expression in vivo using their Nox3-coupled dtTomato fluorescence system [462]. They described Nox3 expression in non-sensory epithelial cells of the endolymphatic sac and duct, of the vestibule and of the semicircular canals, but no Nox3 expression in the hair cells of maculae or ampullae. They also saw Nox3 expression after 7 days of birth in the root cells of the lateral cochlea wall. After 2 months, Deiters’ cells, Claudius’ cells and OHCs started to express Nox3. After 12 months, Nox3 expression further increased in Deiters’ cells, Claudius’ cells and outer and inner phalangeal border cells. IHCs showed Nox3 expression for the first time after 12 months. While these studies delivered excellent detailed information of Nox3 expression in rats and mice, so far, no detailed description of the cellular expression patterns of Nox3 has been conducted in the human inner ear.

#### 2.2.2. Nox3 in Lung Cells

Nox3 was weakly detected in mouse lung endothelial cells [453,483,499,500,501] and in primary human lung fibroblasts [502].

#### 2.2.3. Nox3 in Liver Cells

The human liver cell line HepG2 naturally expresses Nox3 mRNA and protein [363,487,503,504], which is of critical importance, since this cell line serves as cellular model for most of the Nox3-related research on liver diseases (see Section 5.4.1 and Section 5.4.2). This is in notable contrast to the absence of Nox3 in the naïve murine fetal or adult liver [487].

#### 2.2.4. Nox3 in Fibroblasts, Endothelial and Epithelial Cells in General

Ahmarani and colleagues expanded the list of cells in which Nox3 is naturally expressed [102]. They detected Nox3 in human endocardial endothelial cells (hEECs), human vaginal endothelial cells (hVECs) and vascular smooth muscle cells (hVSMCs). Interestingly, they reported a heterogeneous distribution in dependence of the cell type. In hEECs, Nox3 was found in clusters at the intracellular cell membranes, while in hVEVs and hVSMCs, it was equally distributed in intracellular membranes, including the nuclear membranes. Moreover, in all cell types, Nox3 was more abundant at the nuclear membranes compared to all intracellular membranes. Among the cell types, hVECS showed the strongest density of Nox3. Nox3 mRNA was further detected in late endothelial progenitor cells (EPC) together with Nox1, Nox2, Nox4 and Nox5 [505], in human nasal polyp-derived fibroblasts [506] and expressed as protein in the fibroblast-like cell line 3T3-l1 [485]. Notably, Zhang et al. found that Nox2 is the main ROS source in primary human dermal fibroblasts. All other Nox isoforms were at least expressed at the mRNA level, while Nox3 was not detectable at all [507]. Nox3 was also not detected in human umbilical endothelial cells (HUVECS) [508].

#### 2.2.5. Nox3 in Cells of the Eye

Not many studies have investigated Nox3 as a possible ROS source in the eye. Brown et al. analyzed Nox enzymes in rabbit conjunctival fibroblast in the context of the fibrotic response [509]. The group found that Nox2, Nox4 and Nox5 and Nox3 mRNA were strongly expressed in this cell type, while Nox1 or the Duox enzymes were not detectable. Transforming growth factor (TGF)-β treatment, which was used as a stimulating factor in this study, did not stimulate the expression of Nox3; therefore, the role of Nox3 in this context was not further investigated. Furthermore, O Brian and colleagues could not detect Nox3 mRNA or protein expression in human corneal stromal cells [510]. As a result, if and how Nox3 might play a role during human eye diseases is completely unknown.

#### 2.2.6. Nox3 in Cells of the Nervous System

Olguin-Alberne et al. investigated the involvement of Nox-derived ROS during the cell death of murine astrocytes induced by Staurosporin [511]. They could not detect Nox3 mRNA in astrocytes cultured for 2 weeks, while Nox1, Nox2 and Nox4 were detected. Nox3 absence in astrocytes was later confirmed by Reinehr et al. [460]. Oddly enough, Olguin-Alberne et al. further investigated Nox3-deficient mice and, not surprisingly, there was no difference between WT astrocytes and Nox3-deficient astrocytes. Notably, Acette et al. detected Nox3 mRNA expression in the oligodendrocyte cell line MO3-13 [512]. herefore, Nox3 should not be fully excluded from neuronal research.

#### 2.2.7. Nox3 in Cancer Cells

During a previous analysis of ex vivo human cancer tissues, Nox3 was not detected. Further screening of various cancer cell lines, however, showed strong Nox3 mRNA and protein expression in the cell lines H28 (mesothelioma), H358 (bronchoalveolar) and A549 (adenocarcinoma). Nox3 was weakly expressed in H157 (squamous), H727 (carcinoid) and H838 (adenocarcinoma) [513]; in the cervix cancer cell line HeLa; in the lung cancer cell line GLC-82 [503]; in the human pancreatic cancer cell line Panc-1 [514], as well as in the human adenocarcinoma cancer cell lines MDA-MB-231, MDA-MB-468 and Hs578T [515]. Nox3 mRNA was also detected in the murine breast cancer line 4T1 [516].

In addition to these cancer cell lines, in which Nox3 was readily detectable, the majority of studies have described the absence of Nox3 in cancer cells, i.e., in the cancer cell lines H322 (bronchoalveolar), H520 (squamous), H1299 (large cell carcinoma), H2122 (adenocarcinoma) and HT29 (colon cancer) [513]; in the squamous carcinoma cell lines HSC-2, HSC-3, HSC-4, SAS and OSC-19 [517]; in the osteosarcoma cell lines HOS, MOS, MG-63, NOS-1 and HuO 9N2 [518]; in the malignant pleural mesothelioma cell lines ACC-MESO-1, ACC-MESO4, Y-MESO-8A, MSTO-2211H, NCI-H28, NCI-H290 and NCI-H2052 and the untransformed mesothelial cell line (Met-5A) [519]. Furthermore, no Nox3 expression was detected in the myeloid leukemia cell line K-562 [508] and, finally, in several other cancer cell lines (LS180, Caco2, LS174T, HT-29, PC-3, LNCap, DU145, MCF-7, BT474, ZR-75, MB-468, K562, HL-60, OVCAR-3, Skov-3, SK-Mel 5, A2058, HepG2, HEK293, TC-71) investigated in a broad screening study by Juhasz and colleagues [113].

#### 2.2.8. Nox3 in Immune Cells

The first description of Nox3 expression in an immune cell type was made by van Buul et al., which detected Nox3 in the T-cell cancer line Jurkat [508]. Miyano and colleagues firstly showed that Nox3 is expressed and active in innate immune cells, namely the macrophage-like cancer cell line RAW 246.7 [275], which was confirmed in later studies [520,521]. In contrast, Nox3 mRNA was not detected in ex vivo Kupffer macrophages [460], and since no other ex vivo cell analysis was performed until now, it remains unclear if Nox3 belongs to the basic Nox repertoire of macrophages or if it is more part of the cancerous phenotype of RAW cells.

Feng and colleagues reported, for the first time, Nox3 expression on the mRNA and protein level in murine spleen B cells and in the human B cell line BAL17 [522], while Nox3 was not detected in the human B cell line Ramos [508]. Therefore, these findings remain somewhat contradictory.

Gaurav et al. investigated the role of eosinophils during allergic asthma [523] and detected high amounts of Nox2, Duox1 and Doux2 mRNA in human peripheral blood eosinophils, but only minor mRNA levels of Nox3 and Nox5.

Li et al. investigated the role of Nox enzymes in murine mast cells after UVA-induced Ca^2+^ fluctuations [524]. They detected strong mRNA expression of Nox2 and of its subunits p22^phox^, p47^phox^, p67^phox^, p40^phox^ and Rac 1/2, as well as moderate expression Duox1 in the rat mast cell line RBL-2H3. All other Nox isoforms, including Nox3, were not detected.

The rarity of studies which have investigated Nox3 in immune cells in general and the partially contradicting findings of the already conducted studies clearly demonstrate that this topic represents a vast empty field for future research.

#### 2.2.9. Nox3 in Other Cell Types

Nox enzymes were reported to be expressed in placental tissue before [525,526,527], but Polettini and colleagues dug deeper into this topic and analyzed human amniochorions, i.e., fetal membranes [494]. Expression of Nox2, Nox3 and Nox4 mRNA were detectable in healthy patients and in patients with either preterm premature rupture of membranes or preterm birth with intact membranes. Patients with chorioamnionitis were excluded from this investigation, since infiltrating immune cells would have confounded the obtained data. Nox1 and Nox5 mRNA was not detectable in the samples. Notably, the localization of Nox3 protein expression was present in both amnion and chorion cells.

Morimoto et al. described, in stably proliferating germline stem cells, strong expression of Nox1, while Nox3 and Nox4 were only weakly expressed [484,528]. However, dependent on the presence or absence of growth factors, the germline stem cells displayed a strongly fluctuating Nox isoform expression, with Nox3 as majorly expressed protein (see also Section 4.2). Issa et al. detected Nox3 mRNA and protein in the adipocyte cell line 3T3-1L [529]. Nox3 could not be detected in human induced pluripotent stem cell (iPSC)-derived CD34+ hematopoietic precursor cells [530], in immortalized primary human myometrial or in fibroid uterine cells [531].

### 2.3. Subcellular Locations of Nox3

While the expression either on the mRNA or the protein level was extensively described for Nox3 in tissues and cells in general, only a few studies have investigated the exact location of Nox3 in cells. For other Nox isoforms cellular locations were extensively investigated. Nox2 shows a rather restricted placement at the plasma membrane and at the membrane of phagosomes/endosomes, while Nox4 is broadly distributed over many intracellular structures [84,207], such as the nucleus [338] or the ER [218].

Uemaya and colleagues first described Nox3 localization at the plasma membrane, together with p22^phox^, p67^phox^ and, as described before [369], NOXO1 in co-transfected HEK-293 cells [274]. The authors also suggested a mainly extracellular ROS production based on this observation. Nakano and colleagues also reported p22^phox^-dependent localization of Nox3 at the plasma membrane in co-expression systems with HEK-293 and CHO cell lines [217]. During their analysis of the general Nox3 expression in cells, Ahmarani and colleagues reported a heterogeneous distribution of Nox3 in dependence of the cell type [102]. In hEECs, Nox3 was found in clusters at intracellular cell membranes, while in hVEVs and hVSMCs, Nox3 was equally distributed in intracellular membranes including the nuclear membranes. Moreover, in all cell types, Nox3 was more abundant at the nuclear membranes compared to all intracellular membranes. The exact location of Nox3 for most of the cell types is still unclear and represents a highly interesting research field.

Taken together, a plethora of studies have investigated and reported Nox3 expression (some on the protein level, but most of them only on the mRNA expression level), in many organs, tissues (in vivo or ex vivo as explants) and cell types (as primary cells or cell lines). These findings revise the often-cited statement of Nox3 as “only expressed in the inner ear”. Sadly, studies which have investigated the exact subcellular location that obviously is dependent on the cell type, are scarce. Nevertheless, it seems that Nox3 might also exploit an interesting variability in terms of the subcellular location. Considering the vast amount of research, which was conducted so far to determine the structure (Section 1.2), induction/regulation (Section 3) and functions (Section 4) of Nox3, as well as possible therapeutically treatment options (Section 5) that target Nox3, it is highly surprising that nearly nothing is known about Nox3 in humans except for the expression in some organs [363,372,404,487,494]. No human material from organs, where Nox3 was clearly involved in pivotal functions in other species, such as rats and mice (e.g., from the inner ear, lung or liver, Section 4 and Section 5) was investigated, let alone that any treatment option, which targets Nox3 in a mouse or rat model went into a clinical trial so far. Thus, in the nearly complete lack of information for Nox3 in ex vivo human tissue lies a huge potential for new and fruitful research.

## 3. Activation and Regulation of Nox3

Considering the expression of Nox3 in various cell types and tissues, logically, each cell type of a specific organ or body compartment reacts differently to external and internal stimuli. These factors can be of endogenous origin, e.g., growth factors, cytokines and hypoxia or enter from the exterior, like pathogenic infection and physical or chemical hazards. When, how and if Nox3 is activated by these stimuli will be discussed in this section. A strict separation was made between the actual activation of the Nox3 enzyme, i.e., induced ROS production, and the regulatory processes, which also include modifications of Nox3 mRNA expression in any way [13,390]. Nox3 resembles Nox4 in terms of basal ROS production. Accordingly, an increase of Nox3 protein expression can correlate with higher ROS production and might influence the subsequent cellular events. However, this is not actually an induction of the enzymatic activity.

### 3.1. Activation of Nox3

Undoubtedly, the reader will swiftly notice that only a few studies have investigated and experimentally showed Nox3 activation, which is ROS production after cdefined stimuli. Most of the studies only analyzed mRNA or protein expression in this context, which both do not necessarily correlate with actual enzyme presence [530,532,533,534], activation and directed production of ROS. Therefore, when studies only performed expression analysis without providing clear evidence of Nox3 being the actual ROS source (e.g., via knock-out or knock-down) and/or without any ROS measurements at all, these studies will be discussed in Section 3.2, which summarizes the regulation of Nox3.

Nox3 was found to be activated by various stimuli involved in diseases progression, such as insulin in HepG2 cells [487], cisplatin treatment in the organ of Corti and the associated cells [355] and, for the first and only time so far, in B cells, via BCR-ligand triggering [522]. Li and colleagues described a direct activation of Nox3 after TNF treatment, which was mediated by PKC activation and subsequent p47^phox^ translocation to Nox3 at the plasma membrane [396] (Figure 3A,B).

Similar to the knowledge about subcellular Nox3 location, also a clear scientific picture of Nox3 activation and ROS production, which does not always correlate with increased expression, is sadly very low. Considering the many discovered organs, tissues and cells in which Nox3 is expressed aside from the inner ear, a lot of interesting research potential lies in the question by which stimuli Nox3-derived ROS production is activated, especially in ex vivo cells.

### 3.2. Regulation of Nox3

#### 3.2.1. Nox3 Regulation on the Expression Level

As mentioned before, most of the studies that investigated Nox3, especially in the context of in vitro or in vivo functions, only analyzed mRNA expression of Nox3. First of all, mRNA content does not necessarily reflect the presence of the build protein [530,532,533,534], making the few studies that took the extra work of depicting the Nox3 protein expression much more conclusive. Secondly, many studies did not confirm Nox3 as precisely responsible for the observed effects, since no genetic evidence, i.e., by knock-out or knock-down, was performed. Nevertheless, regulation of mRNA and protein expression is an important factor of Nox3-mediated ROS production, which will be summarized in the following sections.

##### Up-Regulation Nox3 on the Expression Level

A number of endogenous factors such as cytokines, growth factors, hormones or altered body homeostasis lead to the up-regulation Nox3 expression (Figure 3C). In germline stem (GS) cells, Nox3 protein expression was up-regulated after stimulation with the cytokines glial cell line-derived neurotrophic factor (GDNF) and fibroblast growth factor 2 (FGF2) [484]. Issa et al. described an increase of Nox3 protein after three hours of TGF-β treatment in the adipocyte line 3T3-1L [529]. Similarly, Yasuoka and colleagues detected an increase in Nox3 mRNA after TGF-β or integrin beta-5 (IGBT-5) treatment in primary human lung fibroblasts [502]. Nox3 mRNA expression was increased in the murine breast cancer line 4T1 after isolation from an established tumor setting in mice [516]. These animals were additionally treated with TWS119, a substance that leads to glycogen synthase kinase-3 β (GSK-3β) phosphorylation. GSK-3β is a protein kinase with a high correlation to cancer transformation [535,536]. TWS119 treatment led to a further up-regulation of Nox3 mRNA in the isolated 4T1 tumor cells.

Insulin treatment increased Nox3 protein levels in HepG2 cells, a commonly used cell line for investigation of liver diseases. This phenomenon was also observed 3T3-L1 cells and white adipose tissue in mice [485]. Palmitate treatment also increases Nox3 protein levels in an adipose animal model [537]. Michihara et al. also found that Nox3 mRNA and protein levels were increased in the brain of hypertensive rats [489]. Adipositas, as well as hypertension, can contribute to cardiovascular diseases and the role of Nox3 in this context will be discussed in Section 5.4.

Li and colleagues reported Nox3 mRNA up-regulation after treatment of HepG2 cells with the pro-inflammatory cytokine TNF [396] (Figure 3A,B). Kathanal et al. observed Nox3 mRNA up-regulation after treatment with the Gram-negative bacterial cell wall component lipopolysaccharide (LPS) [521]. Both findings suggest a possible role for Nox3 during infection and inflammation.

Many exogenous factors, most of them physical or chemical inducers of inflammation, were described to increase Nox3 mRNA and protein levels. The most prominent substance is probably the anti-cancer drug cisplatin, which induces toxic damage by many correlative events that all increase the inflammatory profile of the inner ear, especially in the cochlea [538,539]. Accordingly, several studies have described an increase of Nox3 mRNA [477,540] or protein [462,476,541,542] after cisplatin treatment (Figure 3C).

Exposure to physical hazards also influences Nox3 expression. Carbone monoxide (CO) exposure (3000 parts per million [ppm]) induced Nox3 mRNA expression in the rat striatum [543], and Wang et al. saw a strong increase of Nox3 protein after 1 hour of heavy ion irradiation (1–4 gray) of HeLa, HepG2 and GLC-82 cells [503]. Habashy and colleagues investigated the oxidant and antioxidant responses in chicken livers after mild heat stress (35 °C) [493]. The group detected a basal mRNA expression of Nox3 in liver tissue, which was up-regulated after 1 and 12 days of applied heat stress. Finally, as reported by various studies [462,476,478,544], noise exposure leads to an increase in Nox3 mRNA and protein levels in the cochlea (Figure 3C).

Chemical exposure can also lead to altered Nox3 expression. Kim et al. described an up-regulation of Nox3 mRNA after treatment with endosulfan [545], a widely used pesticide that is associated with immune response dysregulation [546,547]. Ye and colleagues investigated the interplay between oxidative and anti-oxidative responses in rat kidney after phenol-induced kidney injury [492] and detected an increase of Nox2, Nox3, p22^phox^ and p47^phox^ mRNA in isolated brain nuclei. Kim et al. detected a protein up-regulation of Nox3 after mono sodium urate crystal treatment in RAW cells [520].

Some bioactive, substances isolated from medical plants, such as Brevilin A [548] or Genipin [521], also induced Nox3 mRNA and/or protein up-regulation.

Zuhang et al. observed, as the only incidence so far, an increase in Nox3 protein levels in ex vivo spiral ganglion cells after an infection, namely with the Cytomegalo virus [549] (Figure 3C).

While Nox3 involvement during various body functions and disease progression was intensively investigated (Section 4 and Section 5), this last example [549] dramatically displays the vast gap of knowledge of Nox3 in the context of immunity and infection.

##### Down-Regulation or No Effect on Nox3 Expression

Owens and colleagues noted a correlation of Nox3 mRNA levels and the Rieske-Iron-Sulfur protein (RISP) in the Complex III of the mitochondrial respiratory chain. After RISP knock-down in various breast cancer cell lines they detected a decrease in Nox3 mRNA [550].

In contrast to other studies [462,476,478,544], Vlajkovic et al. observed that Nox3 expression is down-regulated in the rat cochlea after noise exposure (100–110 decibels [dB]). More precisely, they showed that Nox3, but not Nox2, is down-regulated in the inner sulcus cell region [479]. Li and colleagues detected Nox3 mRNA in late EPCs together with Nox1, Nox2, Nox4 and Nox5. Angiotensin-II treatment resulted in a strong increase in the mRNA expression of Nox2, Nox4 and Nox5, but no expression changes were detected for Nox3 [505]. Finally, the antioxidative substances Simvastatin and curcumin reduced Nox3 mRNA levels [504].

##### Nox3 Regulation via Other Factors

Qian et al. showed a regulatory role of nitric oxide on the direct enzymatic activity of Nox3 [551]. In COS-7 cells, which were co-transfected with Nox3, as well as NOXO1 and NOXA1, the addition of the NO donator DETA-NONOate inhibited Nox3-mediated superoxide production in a dose-dependent manner. The group of Kiss et al. reported dependency of PKC during p47^phox^-mediated activation of Nox3 [464] (Figure 3A,B), confirming the findings of Li and colleagues [396].

## 4. Functions of Nox3

It is not surprising that Nox3-derived ROS, in regard to Nox3 expression in many different tissues and cell types, fulfill various functions in the body. In this section, the beneficial functions of Nox3-derived ROS will be discussed, while the causes of ROS overproduction or ROS production in the wrong locations, which lead do various diseases, will be summarized in Section 5.

### 4.1. Signaling Functions of Nox3

Remarkably, three very convincing and nicely conducted studies, which investigated Nox3-derived ROS in cellular signaling processes, all investigated the signaling functions of ROS in the context of diabetic liver diseases. The fourth study investigated several cancer cell lines, and these four studies are, so far, the only research conducted for Nox3-derived ROS in the context of signaling pathway modifications.

Previous studies have reported a swift increase of H_2_O_2_ production after insulin treatment [552,553,554] in liver cells and Carnesecchi et al. investigated possible ROS sources involved in this context in the hepatocyte-like cancer cell line HepG2 [487]. The group measured a basal H_2_O_2_ production without any stimulus and a robust increase (28–40%) of H_2_O_2_ production after treatment with 100 nM insulin. Down-regulation of Nox3 by siRNA nicely solidified Nox3 as the source of ROS, since Nox3 knock-down led to the abolishment of H_2_O_2_ production. After insulin treatment, HepG2 cells showed increased phosphorylation of the signaling kinases ERK1/2 and Akt. While Akt phosphorylation was not altered after Nox3 knock-down, phosphorylation of ERK1/2 was decreased through the whole time course of insulin treatment. Insulin-induced ERK1/2 activation leads to Vascular Endothelial Growth Factor (VEGF)-A mRNA and protein expression in HepG2 cells and keratinocytes [555,556,557]. The group further investigated this topic in the context of Nox3-derived ROS production. Indeed, an increase in VEGF-A mRNA and protein expression after insulin treatment was detected, which was strongly decreased after Nox3 knock-down. Notably, the exogenous addition of H_2_O_2_ rescued this effect, thus connecting Nox3, H_2_O_2_ and VEGF-A expression. Finally, the group observed a decreased binding activity of the transcription factor Specific protein 1 (Sp1) [558], which plays a central role in VEGF-A expression [559,560]. This study is one of the few examples during the time course of Nox3-focused research, which clearly shows a consistent line of evidence for Nox3-derived ROS involvement. All critical parameters for Nox-related research were investigated, i.e., the ROS production-inducing stimulus (insulin), confirmation of the ROS source by genetic evidence (via siRNA-mediated knock-down); furthermore, an actual decrease in ROS production confirmed by ROS measurements (same stimulus, same cell type), a connection of the produced ROS and the regulated signaling pathways (ERK and Akt signaling) and finally the influenced cellular outcomes (transcription factor regulation) (Figure 4A).

A study from Li and colleagues investigated the effect of Nox3-derived ROS on the glycogen levels in HepG2 cells [396]. Insulin resistance is a key feature of type 2 diabetes and several studies have documented the involvement of elevated ROS production in insulin resistant cells and tissues [561], which lead to disturbed signaling pathways that regulate the intracellular glycogen levels [562,563]. The group focused on TNF-induced signaling as an inhibiting factor of insulin signaling [564,565]. Wistar rats were fed with a high-fat diet (HFD) for 12 weeks to induce insulin resistance. This was accompanied by increased TNF plasma levels, decreased hepatic glycogen levels and enhanced hepatic ROS production. To link these correlative data sets, the researchers switched to an in vitro model. HepG2 cells were treated with TNF (4–6 ng/mL for 4 days) and showed decreased intracellular glycogen levels and enhanced total cellular ROS production. qRT-PCR analysis revealed Nox3 as the only expressed Nox isoform on the mRNA level in HepG2 cells, together with p22^phox^, p47^phox^, p67^phox^ and Rac1. TNF treatment increased the mRNA expression of Nox3, which was also noticed in the liver in vivo after a HFD. Since not NOXO1, but and only p47^phox^ as possible regulatory subunit of Nox3 was detected in HepG2 cells, the group tested the previously suggested involvement of PKC during Nox3 activation [464]. Indeed, the PKC inhibitor hypericin abolished TNF-induced ROS production. Since PKC signaling induces p47^phox^ translocation from the cytosol to the plasma membrane, this was also investigated. Fluorescence microscopic and Western blot analysis of membrane protein extractions confirmed the translocation of p47^phox^ to the plasma membrane after TNF stimulation. These data show that Nox3-mediated ROS-production is increased by two independent mechanisms in HepG2 after TNF treatment: Firstly, the mRNA expression of Nox3 is increased after TNF treatment, and secondly Nox3, is activated via TNF-mediated PKC activation and p47^phox^ translocation (Figure 3B). The group next confirmed via siRNA-mediated knock-down Nox3 as the sole ROS source after TNF stimulation in these cells. Glycogen levels also remained stable after knock-down of Nox3 in contrast to not transfected cells, showing the involvement of Nox3-meditated ROS in this process. They further investigated the C-Jun-N-terminal Kinase 1/2 (JNK1/2) signaling pathway as the link between ROS and the observed glycogen decrease, since this pathway is not only modulated by ROS [566] but is also involved in insulin sensitivity in mice [567,568]. TNF treatment resulted in phosphorylation of JNK1/2, which could be reversed via Nox3 knock-down. This nice publication identified Nox3 as sole ROS source in TNF–stimulated HepG2 cells, the JNK-pathway as ROS-mediated target, the involvement of ROS in cellular insulin resistance and a possible interplay of TNF, PKC and p47^phox^-mediated activation of Nox3. Furthermore, these findings reveal a clear contrast to the regulation of Nox3 in the inner ear, where Nox3 is only activated via NOXO1 [355,372]. Finally, the group unraveled two very distinct possibilities to regulate Nox3-derived ROS production, i.e., on the expression level or by direct signaling-mediated activation.

A follow-up study from the same lab further focused on the role of free fatty acids (FFA) during insulin resistance and the role of Nox3 in this context [537]. It was previously shown that elevated ROS levels in general are correlated to insulin resistance [569,570] and an involvement of FFA was suggested [571,572,573,574]. The group saw elevated insulin, glycohemoglobin and FFA levels in plasma, as well as decreased hepatic glycogen levels and increased hepatic ROS levels in leptin-deficient mice (db/db mice). This mouse strain is a commonly used model for type 2 diabetes investigations [575]. In vitro studies with HepG2 cells revealed an increase of gluconeogenesis and an impaired cellular glycogen content after palmitate treatment, which mimics insulin resistance in vivo. In this context also increased total cellular ROS levels were observed. A previous study documented the expression of Nox3, p22^phox^, p67^phox^, p47^phox^ and Rac1 in HepG2 cells, but not of other Nox isoforms or subunits [396]. Indeed, expression of Nox3 was up-regulated after palmitate treatment in HepG2 cells and in livers of db/db mice, while Nox1, Nox2, Nox4 or Nox5 were not expressed. Knock-down of Nox3 via siRNA in HepG2 cells reduced Nox3 mRNA expression and ROS production in untreated and palmitate-treated cells, nicely establishing Nox3 as the ROS-producing enzyme in this context. Previous studies have discovered critical roles of the MAPKs, JNK1/2 [396,576,577] and p38 [578] during insulin resistance. ROS-mediated modifications of these pathways [566,579], especially during insulin resistance [580] were also suggested. Indeed, palmitate treatment led to increased JNK1/2 and p38 phosphorylation in HepG2 cells. The activation of these two kinases subsequently led to phosphorylation of the kinases Akt, glycogen synthase kinase-3 (GSK3) and the transcription factor Forkhead box protein O1 (FoxO1), finally resulting in increased gluconeogenesis and reduced glycogen levels. Knock-down of Nox3 reduced phosphorylation of JNK1/2 and p38 as well as suppressed gluconeogenesis (Figure 4B). Again, this is an example of a nice and convincing study in which both, the exact ROS source and the mode of action were clearly described in a cellular system. However, in contrast to the lab’s previous study [396], a direct link of Nox3-derived ROS in vivo was unfortunately not found.

Maletter et al. investigated a completely different topic, i.e., the role of Nox3-derived ROS during cell death signaling. The group focused on the effects of the CD95/Fas ligand CD95L on the human adenocarcinoma cancer cell lines MDA-MB-231, MDA-MB-468 and Hs578T [515]. CD95L treatment, previously cleaved by a metalloproteinase [581,582,583], resulted in a switch from an apoptotic [584] to a pro-motile metastatic phenotype [585]. Binding of cleaved CD95L to the Fas receptor led to subsequent Ca^2+^ release mediated by the transcription factor c-yes [586,587]. The elevated Ca^2+^ levels activated PI3K [585,588] and induced total cellular ROS production. Although the used cancer cell lines expressed Nox2, Nox3 and Nox4, only Nox3 was recruited to the membrane-located signaling platform, which formed after CD95L treatment. Silencing of Nox3 by siRNA abrogated Ca^2+^ release and cell migration in CD95L-treated cells. Unfortunately, no ROS measurements were performed in Nox3-silenced cells. Therefore, no evidence in this otherwise convincing study for a direct link between Nox3-derived ROS and the observed signaling effects in this context could be made.

### 4.2. Functions of Nox3 in Cell Differentiation

Sasaki and colleagues firstly investigated the involvement of Nox3-derived ROS during cell differentiation [589]. They used RAW246.7 cells to investigate a possible role of Nox-derived ROS during osteoclast differentiation. Previous studies for this topic were contradictory. Osteoclasts express Nox2 [590]. However, Nox2-defcient osteoclasts still produce O_2_^−^, and Nox2-deficient mice show no abnormalities in their bone structure [86]. As redundant ROS sources, Nox4 in differentiated osteoclasts [86,591] and Nox1 in osteoclast precursors [592] were suggested. Notably, in Nox1-deficient [125,593] and Nox3-deficient mice [370], no bone abnormalities occur. Sasaki et al. detected small amounts of Nox3 mRNA in RAW246.7 cells (0.001% in comparison to the highly expressed Nox2), while NOXA1 mRNA could not be detected. After treatment with Receptor Activator of NF-κB Ligand (RANKL), which is an osteoclast differentiation factor, Nox2 mRNA expression was strongly down-regulated, while Nox3 expression only slightly decreased. Nox1 expression on the other hand was strongly increased. Notably, expression of NOXO1, an important enhancer of ROS production of both Nox1 and Nox3, decreased. Accordingly, the O_2_^−^ production was reduced but did not vanish completely. This suggests a flexible adaptive switch of Nox enzymes for ROS production during differentiation of osteoclasts. Unfortunately, there was no direct evidence of Nox3-derived ROS during this process, since only p22^phox^ or p67^phox^ were down-regulated via siRNA.

Several lines of evidence suggest that ROS in general are necessary for the differentiation of cells of the nervous system [594,595,596], which was shown in detail for the PC12 cell line [597], glia cells [598], neuroblastoma cells [599] and oligodendrocytes [600]. Previous studies, which investigated a possible role for Nox-derived ROS during oligodendrocyte differentiation [598], only used the very unspecific inhibitors apocyanin [601,602,603] or DPI [7,13,604,605,606]. No genetic evidence (knock-out or knock-down) was provided for Nox enzyme involvement [79], so this issue has remained unresolved. Acette and colleagues further investigated this issue in the oligodendrocyte-like human cancer cell line MO3-13 [512]. They found that MO3-13 cells express Duox1, Duox2, Nox5 and Nox3. Nox3 knock-down reduced the expression of Myelin Basic Protein and the nuclear factor Olig-2, which are two important markers of oligodendrocyte differentiation [607,608,609]. Unfortunately, no other cell responses, especially the ROS production, were analyzed after knock-down of Nox3. Hence, again, a direct link of Nox3-derived ROS and the expression of differentiation markers in oligodendrocytes could not be made.

Morimoto and colleagues investigated in a nicely conducted study a putative function for Nox3-derived ROS during proliferation of murine GS cells in vitro and in vivo [528]. In a previous study, the group identified Nox1 as the majorly expressed ROS source in stably-growing GS cells. Cellular knock-down of Nox1 in cells or a full body knock-out of Nox1 in mice led to a reduced proliferation activity of GS cells [484]. In this follow-up study, the group reported a strongly dynamic expression pattern of Nox enzymes in dependency on their proliferation status. Stably proliferating GS cells strongly expressed Nox1, while Nox3 and Nox4 were only weakly expressed, as shown previously [484]. Without any proliferation-stimulating factors, i.e., the growth factors FGF2 and GDNF, Nox1, Nox2 and Nox3 expression was strongly up-regulated. Notably, when FGF2 and GDNF were added to actively proliferating GS cells, Nox1 and Nox2 mRNA levels were down-regulated and only Nox3 mRNA was up-regulated. The expression of Nox3 was modulated by the MAPK and PI3K signaling pathways, since chemical inhibition of both pathways led to a strong down-regulation of Nox3 mRNA expression. Knock-down of Nox3 resulted in decreased ROS production and reduced gene expression of *ld4*, *etv5*, *Nanos3*, *Neurig3*, *Blc6b*, *Ztb16*, *Cdkn1a*, *CCnd2* and *Ccnd3*. An increase of gene expression was detected for *Ccnd1*, *Sohlh1* and *CDkn1b*. Nox3 knock-down finally led to apoptotic cell death and a defect in active proliferation. The group furthermore expanded their findings by analyzing the testes from 7- to 10-day-old mice. Isolated testicular cells were treated with small hairpin (sh)RNA against Nox3 and subsequently showed increased self-renewal activity in comparison to the control cells. This study firstly described the presence of Nox3 in testesand analyzed its contribution to the self-renewal capability after cytokine stimulation as well as during GS cell maintenance under non-stimulated conditions. Sadly, a direct mechanistic link between Nox3- or Nox1-derived ROS and the proliferation capacities was not investigated.

Mazzonetto and colleagues investigated of the interplay of Nox3 and Sonic Hedgehog (SHH)-mediated signaling during the development of granule cell precursor differentiation [610]. Purkinje cells produce and secrete SHH [611], a protein that is the major proliferatory stimulus for granule cell precursors [611,612,613,614]. After binding to the SHH receptor called Patched, the intracellular signaling pathway is activated and leads to induction of proliferation [615]. Dysfunction of this pathway and the subsequent disturbance of cerebellar neurons during development can cause ataxia, which manifests in neurological malfunction and motor discoordination [616]. Mazzonetto et al. characterized a BALB/c mutant mouse line that showed a phenotype which resembled ataxia and was established in a previous mutagenesis screening [617]. The most obvious phenotype of the mutant animals was a lack of motor coordination. This phenotype did not get worse with age indicating a developmental defect. Linkage analysis revealed the location of the mutation in chromosome 17, which is the syntenic region of the human chromosome 6. Several other studies have located mutations linked to Nox3 in chromosome 6 in patients with developmental disturbances and hearing loss [618]. The candidate genes present in the mutated regions were *Tiam2*, *Tfb1m*, *Cldn20* and *NOX3*. After applying a singular nucleotide variant filtering and further data analysis, only one singular nucleotide variant remained in exon 3 of the *NOX3* gene. The mouse line was subsequently named *NOX3^eqlb^* after the newly discovered allele. *NOX3^eqlb^* mice showed an unaltered *NOX3* gene expression, while the *NOX1* gene was strongly expressed in comparison to WT mice. The group isolated cerebellar and neural stem (NS) cells and observed a slight increase in total cellular ROS in *NOX3^eqlb^*-derived cells after 7 days and much less ROS production after 12 days in cell culture. The cerebellum of *NOX3^eqlb^* mice showed no abnormalities, but the group reported a thicker external granular layer, a disorganized Purkinje cell monolayer and more Bromodeoxyuridine (BrdU)-positive cells indicating increased proliferation. Organotypic in vitro cultures of cerebella and granular precursor proliferator cells isolated from in *NOX3^eqlb^* mutant mice showed a higher proliferation rate in comparison to WT mice. Cultured neurospheres from isolated NS increased in size much earlier when derived from *NOX3^eqlb^* mice but normalized at day 10 to a similar degree as in WT mice. Other organs such as the heart, liver, muscles, kidney and other brain regions showed no increased proliferative activity. Microarray analysis detected 116 up-regulated and 40 down-regulated genes at day 6 after birth. At day 15, 64 genes were up- and 5 were down-regulated. All of these genes were involved in proliferation and cell growth, e.g., *Cdkn2a*, *Cd133*, *CCnb1*, *Cdk1*, *Rb1*, *Cdc25*, *Akt1* and *Sox2* [619]. Increased levels of SHH protein, the main mitogenic driver in this context, was detected in the cerebella of *NOX3^eqlb^* mice. Additionally, increased expression of genes down-stream of the SHH pathway (*Ccnd1* and *Gli1*, *2*, *3* [620,621,622]) was detected. Since SHH-mediated signaling is activated via ROS [623], a connective mechanism was suggested. Unfortunately, in this otherwise excellent study, only the unspecific inhibitor apocyanin was used, and no direct evidence was given for the involvement of Nox3-derived ROS in this context.

Feng and colleagues investigated, for the first time, the presence and function of Nox3 in B cells [522]. Upon exposure to antigens, B cells undergo proliferation and activation mediated by a complex signaling cascade [624,625,626,627]. The involvement of ROS during cellular signaling was established in various immune cells before [4,80,145,146], including B cells [628,629,630,631,632]. While mitochondria [632] and Nox2 [629] were identified as activated ROS source in B cells, other Nox enzymes were not investigated. Notably Nox2-deficient B cells normally proliferate, which suggests that ROS produced by other Nox isoforms might be more important in this process [629]. Feng et al. analyzed this topic and found that in ex vivo murine splenic B cells and in the B cell line BAL-17 the mRNA expressions of the *NOX1*, *NOX3*, *DUOX2*, *NOXA1* and *NOXO1* genes were up-regulated after B cell receptor activation. Nox4 and Duox1 mRNA could not be detected. The group measured no total cellular ROS production in Nox2- or DuoxA1/2-deficient B cells in the early phase (1 h), but prolonged ROS production at later time points (4–6 h). Additionally, no disturbance in proliferation was reported, suggesting no role for the early ROS production mediated by Nox2 and Duox2. Interestingly, prolonged ROS production and proliferation in B cells was abolished in p22^phox^- or Nox3-deficient BAL-17 cells, but not in Nox1-deficient cells. Via the CRISPR/Cas9 knock-out system, the group nicely identified Nox3 as responsible ROS source. However, instead of using the nicely established knock-out cell lines, the group only used the globally working ROS scavenger NAC to investigate the role of ROS in the signaling cascade, which mediates B cell activation. Therefore, a direct mechanistic link between Nox3-derived ROS and the signaling cascade necessary for B cell activation was not demonstrated.

Park et al. investigated the role of 8-hydroxy-2′-deoxyguanosine (8-OHdG), an oxidatively modified DNA base and biomarker of oxidative distress [179,633,634,635], and its paradoxical role as exogenous anti-inflammatory and anti-oxidative component [636]. Treatment of human pancreas cancer cell line (Panc-1) cells with 8-OhdG resulted in decreased total cellular ROS production and a reduction of Nox1, Nox2 and Nox3 mRNA expression. [514]. However, none of these Nox isoforms were confirmed as involved ROS sources via knock-down or knock-out experiments.

Al-Sabbagh et al. investigated the functions of Nox enzymes during decidualization, a process which summarizes the cellular changes for pregnancy preparation of human endometrial stromal cells [637]. After 8-bromo-cAMP stimulation, which induces signaling events that lead to decidualization, the group observed a p22^phox^-dependent and Rac1-independent response of stromal cells. Despite the fact that Rac1 is not completely necessary for full Nox3-mediated ROS production, they excluded Nox3 as ROS source and focused on Nox4, which was confirmed as major ROS source via siRNA knock-down experiments. Unfortunately, since no broad experimental screening via qRT-PCR or Western blot experiments were conducted, it remains elusive if and how Nox3 is expressed or if Nox3-derived ROS fulfill other important functions in human endometrial stromal cells.

### 4.3. Functions of Nox3 in the Inner Ear

Before Nox3 can be put in the context of ROS-associated benefits or malfunctions for the inner ear, it should be mentioned that ROS and their subsequent effects on the inner ear were described before the discovery of Nox3 [469,472]. Moreover, as mentioned earlier (Section 2), Nox3 is also not the only Nox isoform expressed in the inner ear [479]. It is also noteworthy that the striking overlap of NOXO1 and Nox3 mRNA expression patterns, as well as the observed similar phenotypes of NOXO1- or Nox3-deficient animals led to a synonymous use for NOXO1 and Nox3 deficiency in some occasions [355,370,464].

#### 4.3.1. Functions of Nox3 in the Vestibular System

While the effects of Nox3 deficiency on otoconial development were described in vivo by Paffenholz and colleges [370], it took 2 years until an indirect hint for the precise mechanism of Nox3-derived ROS in this process was discovered by Kiss et al. [464]. The group analyzed a spontaneously emerged mouse mutant line with severe balance deficits, named “head slant” (*hslt*), described by Gagnon and colleagues in 2013. (short report available as PDF on the Jackson Laboratory Website, https://www.informatics.jax.org/downloads/Reference_texts/J86035.pdf, accessed on 2 February 2024). Kiss et al. further characterized this mutant mouse line and confirmed strong balance and orientation deficits, while the hearing capacities were not altered. All of the *hslt* mutant mice were homozygous for a mutant *NOXO1* gene allele (therefore named *NOXO1^hslt^*). The group elegantly showed via a transgenic rescue with a functional *NOXO1* gene allele that the dysfunctional NOXO1 subunit is indeed responsible for this severe phenotype, since all transgenically rescued animals showed normal gravity and balance perception similar to wild-type (WT) animals. Strikingly, in all *NOXO1^hslt^* animals, a complete absence of CaCO_3_ and otoconia in the inner ear was reported. Instead of functional otoconia, otoconia-like unstructured conglomerates were spotted directly above the sensory hair cells. Other compartments of the inner ear, like the sensory epithelia, the tectorial membrane of the organ of Corti and the ampullae of semicircular canals, were all intact. The group also reported a broad expression of NOXO1 mRNA in the sensory and nonsensory epithelial cell layers of the saccule, in the ampullae of semicircular canals, in the epithelium lining of the scala media and in spiral ganglion neurons. In vitro expression of the NOXO1^hslt^ protein in HEK293 cells resulted in an abolished Nox3-mediated ROS production in comparison to the cells, which expressed the NOXO1^wt^ protein. This was also observed after co-transfection of NOXA1 and NOXO1^hslt^, while the co-expression of NOXA1 and NOXO1^wt^ showed maximal ROS production via Nox3. This study firstly showed a direct and not a correlative connection between NOXO1, Nox3-derived ROS and otoconia formation. The group suggested changes in the OC-90/95 protein itself or during the delivery of OC-90/95 to the forming otoconia. These suggestions supported the observations of Paffenholz et al., which showed that H_2_O_2_ leads to disulfide linkage and conformational changes in the secreted OC-90/95 protein [370]. During the period in which they conducted their study, Paffenholz and colleagues hypothesized that Nox3-derived ROS might lead to peroxidation of the lipid vesicles in which the globular substance for otoconia formation is stored. The lipid vesicle peroxidation then could lead to Ca^2+^ release on the one hand and accessibility of OC-90/95 to the globular substances on the other hand. Also, the involvement of Nox3-derived ROS and their influence on CaCO_3_ concentrations at the otoconia-forming regions of the vestibular system was suggested as mechanism by a later study [462]. Although the impressive phenotype of Nox3-deficient animals regarding otoconia was intensively investigated and described in vivo, the study by Kiss and colleagues firstly provided a deepened mechanistic explanation how the Nox3-derived ROS might contribute to otoconia formation. The suggested theory of Paffenholz et al. and Kiss et al. of ROS-mediated disulfide-linkage of OC-90/95 and its important effect on otoconial formation should be proven right, however 15 years later [466].

This follow-up study after nearly 15 years was conducted by Xu and colleagues, who firstly investigated the suggested direct mechanistic interplay of Nox3-derived ROS and OC-90/95 in vitro and in vivo [466]. Considering the vast amount of time which passed from the first discovery of the correlative presence of Nox3 and otoconia [355,370], it is notable and laudable that at least one study has investigated this research issue directly. The group therefore generated OC-90/Nox3 double knock-out mice via cross breeding of previously described mouse strains [370,438] and compared WT, single knock-out and double knock-out animals with various experimental approaches. OC-90 and Nox3-deficient animals stayed for a shorter period of time on the rotarod, a testing device for balancing [638], in comparison to WT animals. OC-90/Nox3-deficient animals endured for an even shorter period. Notably, while WT mice adapted from experiment to experiment, all other mouse strains did not adapt, indicating a permanent deficit of balance functions. Measurements of the vestibular evoked potentials, which reflect the activity of the vestibular nerve, were completely absent in OC-90/Nox3-deficient mice. The vestibular nerve and its information relay to the subsequent neuronal network depend on the proper function of the utricle and saccule, where otoconia are located. The absence of any vestibular-evoked potential in the double-deficient animals suggested a severe impairment of gravitational reception [409,461]. Accordingly, in Nox3- and OC-90/Nox3-deficient animals, otoconial structures were completely absent as depicted via scanning electron microscopy. Double-deficient animals also displayed loss of hair cell bundles. This loss was not present from birth but appeared after 3 months. The group used a co-expression system in the NIH/3T3 cell line cultured under extracellular calcification conditions in vitro. Strikingly, transfection with an empty vector or expression of Nox3 alone induced no or minor calcification, respectively. Sole expression of OC-90 induced a stronger formation of calcified nodules on the cell surfaces. Finally, the co-expression of OC-90 and Nox3 together resulted in the strongest calcification process. This simple but nicely conducted study is the one and only research performed so far, which clearly showed the importance of Nox3 for the process of otoconia formation directly and not as a correlative observation. The only experiments, which would have added important information to these findings, are (i) ROS measurements, to prove that the transfected Nox3 indeed produces ROS into the extracellular milieu, and (ii) the addition of a ROS scavenger into the medium, e.g., N-acetyl cysteine (NAC), to prove that the Nox3-derived ROS are the potentiating factor of OC-90-mediated otoconial formation. OC-90 has a remarkable number of cysteine residues [435,639,640,641], and Xu et al. suggested a mechanism in which disulfide bond-dependent multimer formation of OC-90 in the endolymph, which is otherwise a soluble monomer, then serves as scaffolding platform for otoconial growth (Figure 4C). So far, this is the most reasonable mechanism of Nox3-derived ROS for otoconial formation, and future studies will hopefully further investigate this important topic that is still not fully resolved.

Jones et al. characterized a number of mouse strains, which all lack otoconia, namely *head tlt*, *het-Nox3*, *tilted* and *tlt-Otop1* [642]. Otoconia-deficient mice failed to swim and orientate like described before [370,464,466]. The group measured spontaneous activity of the vestibular primary afferents, which innervate the maculae, even in the absence of otoconia [643]. The vestibular primary afferents further displayed higher discharge rates in comparison to WT animals. These data suggest that in absence of stimulation due to otoconia loss the resting activity in macular primary afferents and the ribbon synapses present in hair cells of otoconia-deficient mice are still functional. Basaldella and colleagues further investigated the interplay of the vestibular and proprioceptive system and body balance in an impressive study [644]. For this purpose, the group used Nox3-deficient mice as in vivo model system. As mentioned above, these mice are devoid of otoconia in the utricle and the saccule of the inner ear [370], which leads to defects in perception of gravity and linear acceleration, while the auditory system remains intact. The research group analyzed the communication with the lateral vestibular nucleus and other motor neuron pools. The lateral vestibular nucleus is one of the four major nuclei that form the vestibular complex. This complex is essential for maintaining the head position and clear vision during movement [645,646]. In this context, the group reported that the lateral vestibular nuclei (LVe) neurons maintain the synaptic input to motor neurons even in Nox3-deficient mice, in which otoconia are absent and vestibular signaling is non-functional. They also reported a higher synaptic input density, but no differences in the synaptic inputs mediated from the LVe neurons to other motor neuron pools. This study demonstrated that genetic distortion via Nox3 deficiency of vestibular input channels or silencing of the synaptic output of vestibular neurons leads to comparable connectivity defects between LVe neurons and flexor motor neurons. Ward and colleagues conducted a comparative study with WT mice and Nox3-deficient (*head tilt*, *Nox3^het−3J^*) mice as model for otoconial-deficient inner ear lesions [465]. The group described no eye movement in response to static body tilts about the earth-horizontal axis in Nox3-deficient mice. Through application of a magnetic field, nystagmus occurrence in mice can be studied [647,648,649]. Using this technique, the group saw that WT mice showed different variants of nystagmus. Nose-first entry into the magnetic field induced a left-beating nystagmus, tail-first entry resulted in a right-beating nystagmus. Nox3-deficient mice showed no nystagmus in any of these tested positions. The group nicely showed that the nystagmus occurrence, usually observed in mice with intact vestibular functions, was absent in Nox3-deficient mice and concluded that a functional otoconial structure is critical for the development of a nystagmus in magnetic fields.

#### 4.3.2. Functions of Nox3 in the Cochlea

Interestingly, while Nox3-derived ROS are crucial for otoconia formation and a functional vestibular system [355,370,408,464,466] (Section 4.3.1), in the cochlea, no physiological functions of the Nox3-derived ROS have been described since its discovery [355,392]. On the contrary, Nox3-deficient mice showed normal hearing capacities [217,355,650]. Instead, non-physiological ROS overproduction by Nox3 results in cochlear damage with severe outcomes [404,476,651]. Nox3-mediated ROS overproduction can be easily triggered, e.g., by cisplatin-treatment [652,653,654], by noise exposure [462,470,655] or when blood flow [656,657] or oxygen tension decrease [656]. The correlative involvement of ROS in the cochlea and destruction of hair cells as cause for hearing loss was shown in many studies before [470,658,659]. Of course, the most obvious way to treat this ROS-induced damage is to counter-act with anti-oxidants [469,660], such as methionine [661,662], lipoic acid [659] or NAC [663,664,665] that reduce the global oxidative distress [659,666,667,668,669,670]. No beneficial role for Nox3-derived ROS in the cochlea was discovered so far. Moreover, all studies, which investigated this research topic always reported overproduction of ROS via Nox3 and subsequent cochlea damage and hearing loss. Therefore, this field will be completely discussed in the next section, which summarizes Nox3 involvement in diseases. Remarkably, this topic, i.e. investigation of a possible beneficial effect of Nox3-derived ROS in the cochlea is one of the most intriguing areas for future studies in the Nox3 research field.

## 5. Roles of Nox3 in Diseases

Since research on Nox3 mainly focused on the most prominent expression region, namely the inner ear, most of the research of Nox3-associated diseases satellite around ear-associated illnesses [462,476,478,651]. Therefore, this section will start with this topic. Nevertheless, tremendous research exploited important roles of Nox3 during lung and cardiovascular diseases, again revising the view of Nox3 as “restricted to the inner ear”.

### 5.1. Role of Nox3 in Hearing Loss

Hearing loss affects one out of six people and it is one of the major common sensory impairments of humans worldwide [656,671,672]. Hearing loss can be caused by various extrinsic and intrinsic factors, i.e., noise exposure, drug application (including cisplatin), infections and age-related degeneration [672,673,674,675]. The hearing loss in general results from compromised functioning of the organ of Corti in the cochlea and/or the nerve pathways connected to the auditory part of the brain [676]. Several research studies have reported that the nerve connection from the auditory system of the brain to the sound detecting cells (i.e., the hair cells) of the organ of Corti are the most vulnerable parts damaged by endogenous or exogenous sources [404,677,678,679]. The organ of Corti is built up from IHCs and OHCs surrounded by inner and outer phalangeal cells (or Deiters’ cells), inner and outer pillar cells, Hensens cells and Claudius’ cells, all summed up under the term “supporting cells” [398] (Figure 5A). The hair cells detect low- or high-frequency sounds in dependency on their position [680]. Sensory hair cells, in general, do not regenerate in mammals [681,682] and continuous damage results in the permanent loss of hair cells [683]. Oxidative distress is a major driver of hair cell death and subsequent cochlear damage [470,539,651,673,684,685,686,687], which can be induced alongside noise [688,689,690,691], antibiotics [653,692,693], ototoxic anticancer drugs [653,694], infection [695,696] and aging [686,687,697,698,699]. Theses exogenous or endogenous stress factors all result in increased metabolic activity of the cochlea and increased ROS production [667,690,700]. In some cases, like low blood pressure and/or oxygen deprivation, ROS production waves were measured, which started at the luminal surface of the marginal cells in the stria vascularis [657] and re-occurred after reperfusion of the cochlea. The increased ROS levels can last for a long period of time, for example, up to 10 days after noise exposure [469,667,700,701]. This continuous oxidative distress ultimately contributes to death of OHCs and spiral ganglion cells [702,703,704], irreversible cochlea damage and, tragically, permanent hearing loss [404,705,706]. There are many ROS sources in cells with mitochondria [707,708,709] and Nox enzymes as the most prominent ones [6,13,361]. Importantly, mitochondria of OHCs increase their respiratory activity after noise exposure and generate increased amounts of ROS as byproduct [470,710,711], which also contribute to the harmful oxidative damage besides Nox enzymes in general and Nox3 in particular. I point to various excellent reviews about ROS in the inner ear [712,713] or Nox enzymes in this context [686,714] and focus on Nox3-derived ROS. Notably, many studies have used in vivo Wistar rat models, whose hearing ranges are from around 200 Hertz (Hz) to 90 kHz [715] and measured auditory brainstem responses (ABR) for determining the hearing capacity as major experimental output [462,716].

Nagamani et al. first reported a correlation of four patients with interstitial deletion in the 6q region of the long arm of chromosome 6 and Nox3 expression [618]. Deletions of the 6q region were reported before to be associated with ear anomalies [717,718,719,720], but hearing loss was rarely reported [721]. The study suggested that hearing loss occurred because of interstitial or terminal deletions in the 6q25 region, precisely between the regions 6q25.2 and 6q25.3. This area harbors 12 protein-coding genes, with the *NOX3* gene among them. The study described for the first time a possible involvement of Nox3-related inner ear diseases in humans, which started the investigation of Nox3 as harmful ROS source and possible therapeutic target (Section 6) for patients.

#### 5.1.1. Noise-Induced Ototoxicity

Prolonged exposure to noise is the most common cause of hearing loss worldwide [722,723,724,725,726,727] responsible for 20% of all cases of hearing loss [675]. Exposure to sound pressure levels that exceed 85 dB or immediate exposure to noise impulses lead to irreversible cochlea damage. Exposure to moderate sound levels over a prolonged time period can also harm the spiral ganglion neurons [728,729]. Noise-induced hearing loss is a result from the combined damaging effects of synaptic damage and cochlear hair cell death [730,731]. The noise-induced hearing loss can be temporary or permanent in dependency of the duration, severity and combination of the damaging factors [404]. A number of additional factors can worsen the progress of hearing loss, e.g., other diseases [164,732], social [470,473] and work behavior [733,734] or working conditions [735]. The frequency ranges of the impairment lie between 3.4 and 6 kHz [735]. Previous studies have also suggested genetic components, which might influence the outcome and severity of noise-induced hearing loss [736,737]. For example, mice that already showed age-induced hearing loss were more susceptible to additional noise-induced hearing loss [738]. Furthermore, several mouse lines, which were deficient for antioxidant components, such as superoxide dismutase 1 (SOD1) [700], glutathione peroxidase 1(GPX1) [700], plasma membrane calcium ATPase 2(PMCA2) [739] or Cadherin Related 23 (CDH23) [740] showed also increased sensitivity to noise-induced hearing loss. These findings suggest an important role for ROS in this context in general. Accordingly, a previous study from Ramkumar et al. reported that noise exposure resulted in an increase of ROS levels, oxidative distress and increased pro-inflammatory responses in the chinchilla cochlea [741]. The pro-inflammatory status in the cochlea is mainly attributed to infiltrating immune cells, mainly monocytes [742,743,744], which respond to the cochlear tissue damage and the previously released chemokines from cochlear cells. Together with the already increased ROS production by Nox3 and mitochondria, the pro-inflammatory environment induces a vicious cycle that further increases the cochlear damage instead of dampening it [688,744,745,746]. Importantly, this pro-inflammatory, pro-oxidative setting is not restricted to noise-induced ototoxicity but can be applied to any effect that leads to increased ROS production and cochlear tissue damage. This scenario represents a complex network of cellular mechanisms and communication in the cochlea that still is incompletely understood and needs further investigation [747].

A number of studies performed genetic screens to identify possible factors that might contribute to noise-induced hearing loss. Lavinsky et al. used a well-established Hybrid Mouse Diversity Panel [748,749,750] to investigate possible loci for susceptibility towards noise-induced hearing loss [650]. The *Nox3^het^* allele on the murine chromosome 17 was identified as candidate factor. *Nox3^het^* mice were exposed to noise and ABR threshold shifts (4, 8, 12, 16, 24 and 32 kHz) were analyzed. The group measured a reduction in the ABR threshold shifts of WT mice in comparison to *Nox3^het^* mice at 8 kHz suggesting a role for Nox3 during noise-induced hearing loss in the lower frequency spectrum. Zhao and colleagues performed a genome wide association study (GWAS) in 614 patients of a case-control study to investigate the interplay of noise kurtosis and lifestyle factors with noise-induced hearing loss [751]. Complex noise induces greater damage to the auditory system than steady noise in both animals and humans [752,753]. A complex noise is defined as continuous background noise with temporal appearance of randomly occurring high-level noises [754]. By transforming time-domain variables, like pulse interval distribution or duration, into simple variables by kurtosis [755,756,757], this experimental approach allows to assess the biological effects of complex noise in animal models [752,753,756]. The group reported that the risk of acquiring noise-induced hearing loss was 0.806-times higher for people, which were exposed to complex noise, as shown previously [753,756]. They detected an increased Guanine-to-Tyrosine polymorphism (single nucleotide polymorphism [SNP] rs12195525, GG phenotype) in the locus, which is located in the coding region of the *NOX3* gene. They also observed an increased risk for noise-induced hearing loss in GG phenotype patient groups in which further risk factors, such as smoking or high-volume outputs of technical devices, occurred.

The first study that connected the several correlative dots, i.e., Nox3 expression in the cochlea, per se [355], genetic correlations of noise-induced hearing loss with Nox3 [650,751], increased ROS levels in the cochlea as damaging factors [470,539,651,673,686,687], induction of ROS production by noise exposure [667,690,700] and the subsequent hearing loss, was conducted by Mohri and colleagues [462]. The group investigated the role of Nox3 during noise-induced hearing loss in their dtTomato-Cre reporter system for Nox3 detection in mice [462]. The group exposed 2-month-old WT and Nox3-deficient mice (Nox3 marked with the dtTomato fluorescence tag) to harmful noise at 120 dB for three hours and analyzed the ABR thresholds. At day 7, a lower ABR threshold shift at a high frequency (32 kHz) was measured in Nox3-deficient mice in comparison to WT animals. This was accompanied by a reduced OHC loss in Nox3-deficent animals directly linking Nox3 as damaging factor to hearing loss during noise exposure. A recent study from Rousset and colleagues revised previous findings [462,650] concerning the role of Nox3-derived ROS during hearing loss after white noise exposure [478]. Rousset et al. used the previously described C57BL/6J-*NOX3^het−4J^* mouse strain [457], which carries a loss-of-function allele of Nox3. They applied RNAscope in situ hybridization on murine cochlea explants and detected strong Nox3 mRNA expression in the spiral ganglion, while Nox3 was only weakly expressed in the stria vascularis and not detectable in the organ of Corti. The latter is contradictory to several previous studies [355,462,475,479]. Additionally, they detected Nox3 mRNA in the peripheral auditory neurons in Rosenthal’s canal. After noise exposure, Nox3 mRNA expression was increased in cochlear explants, precisely in the medial and the apical cochlea turns. The group also analyzed the hearing capacities of Nox3-deficient mice and observed no difference in the audiograms in comparison to WT animals after 6 weeks of age confirming not a general deficit of hearing in Nox3-deficient animals. Deafening noise exposure (116 dB) led to an elevation of hearing thresholds at frequencies between 16 and 32 kHz after 24 h in WT mice. A protective effect in Nox3-deficient animals was only observed for 32 kHz. After 7 days of noise exposure, ABR measurements showed a better recovery of hearing in Nox3-deficient mice, while WT animals showed no recovery. Histological examinations of cochlear explants further showed that Nox3-deficient animals had reduced hair cells loss, conserved auditory synapses and intact neuron integrity, which all were deceased in WT animals. This study nicely confirmed previous results [404,462,476,655,758], showing that Nox3 has no direct role for cochlear development and structures in sharp contrast to the otoconia formation in the vestibular system [217,370,464]. Even worse, after noise exposure, Nox3-mediated ROS overproduction results in increased oxidative distress and damage of cochlear structures (Figure 5A).

Goodarzi and colleagues investigated the combined effects of noise exposure and silver nanoparticles (Ag-NPs) on the cochlear function in rats [544]. The influence, either beneficial or detrimental, of nanoparticles, in general, on biological functions of the organism is a swiftly expanding research topic [747,759,760,761]. However, metallic nanoparticles, in particular, exploit toxic effects on cells by increasing the ROS production and pro-inflammatory cytokine release [762]. Ag-NPs can enter the body in various ways, e.g., via ingestion, inhalation or even skin contact [763]. Previous studies have reported toxic effects of Ag-NPs to the cochlea [764,765,766]. Goodarzi et al. compared completely untreated Wistar rats with rats exposed to loud noise (104 dB) for different time intervals. The animals either received not further treatment or were intra-peritoneally injected with Ag-NPs (100 mg/kg body weight). The group measured distortion product otoacoustic emissions (DPOAEs) for screening the inner ear function [767,768]. Animals showed a higher rate of hearing loss when exposed to both noise and Ag-NP at frequencies of 7.26, 8.47 and 9.86 kHz. Oppositely, malondialdehyde (MDA) and SOD levels in the serum were either increased by noise exposure or Ag-NP treatment alone but were not further increased by the combined treatment. qRT-PCR analysis further showed that *TNFSF2*, *IL6* and *NOX3* gene expressions in the cochlea were increased by one of the treatments alone but were not further increased by the combinatory treatment. Further investigations concerning Nox3-derived ROS were not made. A similar research topic was investigated by Shahtaheri et al. The group investigated the effects of white noise in combination with aluminum oxide (Al_2_-O_3_) nanoparticles (AO-NPs) on the cochlear structure in rats [769]. AO-NPs are widely used as thermal insulation material [770], and the exposure to workers that are involved in AO-NPs manufacturing [771,772,773] is correlated with many harmful effects on workers’ health [774,775,776]. Additionally, workers are often exposed to extreme noise levels. Regarding this harmful work environment, Shahtaheri and colleagues analyzed the combinatory harmful effects of AO-NPs and noise exposure (95 dB/20 Hz–20 kHz, 8 h per day) on the cochlea of Wistar rats. The group detected reduced auditory capacities analyzed by DOPAE measurements [727,777] and cochlear damage by histochemical analysis in rats exposed to noise. The damage was further increased by treatment with AO-NPs. AO-NP treatment alone did not alter the investigated parameters. Notably, Nox3 mRNA levels also increased after noise exposure in the cochlea, while AO-NP treatment alone did not change the mRNA expression levels of Nox3. The combinatory effect of both increased Nox3 mRNA expression significantly in comparison to noise exposure alone. This was accompanied by OHC and a supporting cell decrease, while IHC numbers showed no alterations. The authors suggested an enhanced damaging effect of white noise exposure and AO-NP treatment on the cochlea due to increased Nox3-mediated oxidative distress. Critically, neither Nox3 knock-down experiments nor ROS measurements were performed in this context. Hence, again direct evidence for a Nox3 involvement is missing in this study.

#### 5.1.2. Cisplatin-Induced Ototoxicity

Cisplatin is a commonly used chemotherapeutical agent against solid tumors [778,779,780,781,782]. Similar to most chemotherapeutical applications, cisplatin treatment results in strong side effects for the patients like nephrotoxicity and ototoxicity [473,783,784,785,786,787]. Cisplatin-induced nephrotoxicity can be treated with diuretics [788,789], while cisplatin-induced ototoxicity is a much more severe, cumulative and untreatable problem [539,786,790]. It manifests as sensorineural, irreversible hearing loss [791,792,793,794,795] due to damage of the organ of Corti in the cochlea [666,673,796,797]. Specifically, cell death of IHCs and OHCs [473,654,659,798], of spiral ganglion cells [652,799,800,801] and of marginal cells of the stria vascularis [802,803] is increased after cisplatin treatment. Inflammation after cisplatin treatment is another driving factor, which further progresses the cochlear damage [804,805,806,807,808,809]. On the sub-cellular level, cisplatin-mediated cytotoxicity induces DNA damage [810,811], mitochondrial dysfunction [812,813] and increased ROS production by various ROS sources [355,469,813,814,815,816,817]. The accumulating damage due to the oxidative distress further progresses the dysfunction of cochlea [653,654] and vestibular system [817,818,819].

Banfi and colleagues first reported cisplatin-induced Nox3-mediated ROS production by using a co-expression system in HEK293 cells [355]. Mukrerhajea et al. provided further evidence in vivo in the rat cochlea and in vitro in the OHC line UB-OC-1 [820]. Cisplatin treatment induced in both systems increased Nox3 expression and ROS production [541]. Kim and colleagues investigated the role of Nox enzymes during cisplatin-induced ototoxicity in general [540]. They used the mouse auditory cancer cell line HEI-OC1 and in vivo experiments for this approach. Cisplatin treatment induced Nox1 and Nox4 mRNA expression starting after 1 hour. Unfortunately, they claimed that Nox3 mRNA was not detectable; however, the data were not shown in the publication. Notably, in vivo injection of cisplatin for 4 days showed a strong induction of the already basally expressed Nox3 mRNA in the cochlea. However, the group focused on Nox1 and Nox4, and Nox3 as ROS source was not further analyzed. Mohri and colleagues investigated, besides several other important Nox3-related topics (see Section 2, Section 3 and Section 4), also the role of Nox3 during cisplatin-induced hearing loss [476,477]. The group used their well-established reporter system with the dtTomato-coupled Nox3 protein [462]. Tone-burst stimuli (8, 16, 24 and 32 kHz) were applied on 2-month-old WT and Nox3-deficient mice either treated with cisplatin or left untreated. ABR threshold shifts were measured, and WT animals showed deteriorated ABR thresholds at frequencies of 24 and 32 kHz after cisplatin treatment compared to Nox3-deficient animals, which showed no deterioration. WT mice also showed OHC loss, which was lower in Nox3-deficient mice. TdT-mediated dUTP-biotin nick end labeling (TUNEL) assays confirmed increased apoptosis of OHC in WT animals in this context as reported before [476,821]. In Nox3-deficient animals fewer TUNEL-positive OHC were detected. In the lateral wall of the cochlea and the stria vascularis no TUNEL-positive cells were seen in both WT and Nox3-deficient mice. The group furthermore showed that cisplatin treatment increased Nox3 expression in the cochlea predominantly at the basal turn and in the supporting cells. In detail, no Nox3-expressing OHC either with or without cisplatin treatment could be detected in WT animals, while weak Nox3-expression in IHCs and strong Nox3 expression in supporting cells could be observed at least after cisplatin treatment. Together, these studies provide solid evidence that cisplatin treatment increases the presence of Nox3 in the cochlea, which leads to a harmful elevation of ROS production and finally to ototoxicity. Interestingly, in vivo Nox3 is mainly present in the supporting cells and not the OHCs, which nevertheless suffer the greatest damage through the increased ROS production (Figure 5A).

Several studies have provided evidence for a protective role of the activating adenosine A1 receptors (A1ARs) [822] and its agonist adenosine during cochlea-related diseases [479,823,824,825,826,827,828,829,830]. In this context, Kaur and colleagues investigated the role of ROS for the A1AR signaling during cisplatin-induced ototoxicity [758]. They reported that activation of the A1AR signaling pathway by N6-R-phenylisopropyladenosine (R-PIA) prevents hearing loss induced by cisplatin and OHC damage in the rat in vivo. They used the OHC line UB-OC-1 to investigate a role for Nox3-derived ROS in vitro, since ROS have a pro-inflammatory effect during cisplatin-induced ototoxicity [355,541]. Cisplatin treatment for 24 h induced A1AR mRNA and protein expression and increased Nox3 mRNA as well as the total cellular ROS levels. Treatment with R-PIA reduced ROS generation and Nox3 mRNA expression in UB-OC1 cells and in the rat cochlea. Cisplatin treatment of UB-OC-1 cells also induced phosphorylation and nuclear translocation of Signal transducer and activator of transcription 1 (STAT1), which could be inhibited by additional R-PIA treatment. STAT1 signaling contributes to the pro-inflammatory response during cisplatin-induced ototoxicity [831]. Accordingly, treatment with R-PIA reduced cisplatin-induced expression of TNF in the rat cochlea. However, no experiments after Nox3 knock-down or knock-out were performed. Therefore, evidence for the identification of Nox3 as relevant ROS source in this context is missing.

#### 5.1.3. Cytomegalovirus-Induced Hearing Loss

Congenital Cytomegalovirus (CMV) infection often leads to sensorineural hearing loss accompanied by neurological and developmental disabilities [832,833,834,835]. Several studies have monitored apoptotic cell death in the murine cochlea [836] in neonatal mice after CMV infection, subsequently leading to sensorineural hearing loss [549,837]. A correlative increase in total cellular ROS levels was also described in this setting [549]. Due to these previous observations, Zhuang and colleagues picked this topic up and investigated the possible ROS sources and the effect of the anti-inflammatory substance Berberine [838] during CMV-induced ototoxicity [496]. The group detected an increase in apoptosis and total cellular ROS in neonatal murine ex vivo cultured spiral ganglion cells. An increase in Nox3 protein expression was also observed. Additional treatment of Berberine reduced apoptosis, ROS levels and Nox3 expression. However, no genetic evidence was given to validate Nox3 as ROS source. Most critically, the authors claimed that Nox3 was connected to mitochondrial ROS production. No specific mitochondrial ROS measurements were performed, and no co-localization studies of Nox3 with mitochondria, e.g., by immunolabeling and fluorescence microscopy, were conducted. Nevertheless, this is so far the one and only study that has described an induction of Nox3 protein expression as response to infection.

#### 5.1.4. Age-Induced Hearing Loss

Age-induced hearing loss (presbycusis) [839] affects, as the name implies, elderly people. This disease is associated with tremendous social consequences [840,841,842,843]. Similar to other causes for hearing loss, age-induced hearing loss can further progress due to prolonged noise exposure or ototoxic drugs [844]. On the cellular level, the loss of hair cells, spiral ganglion cells and cells of the stria vascularis leads to hearing loss majorly at higher frequencies [841,845]. An increase of age is also accompanied with a disturbance of redox homeostasis not only in the cochlea [686,697], but also in other organs, since gene expression of anti-oxidant systems decrease with age [687,698,699].

Du and colleagues investigated the effects of a HFD in combination with a D-galactosidase-induced rat animal model of aging [846,847] to investigate the cumulative effects on hearing loss [848]. In this animal model, the continuous administration of D-galactose leads to numerous detrimental effects based on metabolic disturbance that mimic the aging process [846,849,850,851]. These effects include dysfunctional mitochondria [850,852,853], increased apoptosis [437,854], neurotoxicity [850,855,856], a shortened lifespan [857] and, after 8 weeks of treatment, symptoms that mimic aging of the cochlea due to increased ROS production [858,859,860,861]. Furthermore, after 8 weeks of D-galactose treatment deletions in the mitochondrial DNA (mtDNA) in the cochlea increase and mitochondria show an oval round shape indicating massive damage. The isolated mtDNA from rat cochlea cells showed increased oxidative damage and subsequent common deletion, which are both biomarkers for oxidative distress, aging and age-related hearing loss [846,862,863,864,865,866]. Du et al. analyzed ABR thresholds and detected the highest ABR threshold shifts for four tested frequencies (4, 8, 16, 32 kHz) in groups treated with both HFD and D-galactose after 12 months. After sole D-galactose treatment Nox3 protein levels increased in the stria vascularis and the spiral ganglion. HFD treatment alone increased Nox3 protein levels only in the stria vascularis. The combined treatment of D-galactose and HFD led to the highest Nox3 expression not only in the stria vascularis and the spiral ganglion, but also in the organ of Corti. Apoptotic cell death in the inner ear was observed for all three conditions, but again the highest cell death rate was reported after the combined treatments. Additionally, all three treatments increased the accumulation of mitochondrial common deletion [867,868], which accompanies mitochondrial damage due to aging [869,870]. Du and colleagues deepened their findings from this previous study [848] with the same D-galactose-induced aging model via RT-PCR and Western blot analysis and reported an increase in Nox3 and p22^phox^ mRNA and protein expression in D-galactose-treated rats in the cochlea [475]. Additional Western blot analysis and TUNEL staining showed that apoptosis increased in the cochlea after D-galactose treatment. These two studies by Du and colleagues gave the first correlative insights of increased Nox3 expression during aging, an associated damaging effect to cochlear structures and the subsequent hearing loss. However, since mitochondria are heavily damaged during this aging model and neither in vivo experiments with Nox3-deficient animals nor in vitro experiments with Nox3 knock-down in cells were conducted, the explicit role and the contribution of Nox3-derived ROS in comparison to ROS produced by the damaged mitochondria remained elusive.

Rousset and colleagues used the A/J mouse strain nmf333, which carries a missense mutation in the p22^phox^ subunit [871], to characterize the role of Nox enzymes in the cochlea during age-induced hearing loss [404]. The group firstly defined age-induced hearing loss in WT animals in their experimental setting. They analyzed ABR threshold levels over an age range from 4 to 26 weeks and observed threshold shifts close to 45 dB after 4 weeks, which progressed up to 75 dB with age. They also detected a progressive hearing loss 32 kHz (in 4-week-old mice) and 5.7 kHz (in 26-week-old mice). In accordance with these data sets, a progressive degeneration of the sensory epithelium from the base to the apical turn was described with a more pronounced cellular degeneration in the basal region. Further analysis of IHC innervation revealed a dramatic decrease in the number of synaptic ribbons per IHC, as well as a decrease in the total neuronal density in the spiral ganglion, which also progressed with age. Since a deficiency of p22^phox^ affects Nox1-4, the group analyzed the presence and distribution of Nox mRNA expression in both the mouse and, highly notably, in the human cochlea. qRT-PCR and in RNAscope in situ hybridization measurements showed high mRNA expression of Nox2, Nox3 and Nox4 in mouse and human cochlea tissue. While Nox2 and Nox4 mRNA was evenly distributed throughout the whole cochlea, Nox3 mRNA was concentrated in the spiral ganglion and moderately expressed in the stria vascularis. Most interestingly, Nox3 mRNA was not detected in hair cells, which is in line with the study from Mohri et al. [462]. p22^phox^-deficient animals showed no disturbance in hearing at young age in comparison to WT mice. However, the loss of the hearing capacities at high frequencies observed in aged WT mice, was nearly absent in p22^phox^-deficient animals together with an intact sensory epithelium and preserved synaptic ribbons. The group further performed a transcriptome analysis of 6-week-old cochlea tissue and detected a down-regulation of ryanodine receptors (Ryr) 1, 2 and 3, which are important for Ca^2+^ homeostasis and accordingly for proper neuronal signaling. Several other genes, all revolving around Ca^2+^ homoeostasis, such as *Otoferlin*, *Vamp1*, and *Snap25* or the glutamate transport, such as *Slc17a6*, *Slc17a8* and *Gria2* were down-regulated in absence of p22^phox^. The group narrowed down the auditory neurons as main cell type where the down-regulation was observed. This remarkable study firstly analyzed the mRNA expression of Nox3 in the human cochlea and clearly solidified a rather detrimental effect of Nox presence on cochlear structures, precisely the neuronal part. Unfortunately, like in other previous studies of the Nox3 research field, the group did not clarify the exact interplay of Nox-derived ROS during Ca^2+^ signaling and the subsequent age-related hearing loss. Moreover, while nicely showing that also Nox2 and Nox4 mRNA is present in the cochlea, the analysis of p22^phox^-deficient animals only enabled suggestions considering the general role of Nox enzymes in the cochlea and not specifically the role of Nox3, especially since Nox2 and Nox4 might also play important roles in this organ [405,406,407]. Protein expression, for example, in cochlea tissue lysates, was not analyzed, Instead, the research group solely relied on mRNA-detecting techniques. Since the opinion that mRNA always correlates with protein presence or even activity of the protein is outdated [529,530,532,533,534], protein level analysis of the cochlea, especially from human samples would have been a ground-breaking contribution to the field of Nox3-related research. Human-related data sets of this topic are still largely missing to date. In their favor, the group mentioned and discussed these critical points already in their paper. In summary, the studies of Rousset and colleagues [478,651,872], together with Mohri et al. [462] represent milestone research articles considering Nox3 investigations in the inner ear. Continuing in this sense, Mohri and colleagues also investigated the topic of age-induced hearing loss with their generated mouse line, which expresses the fluorescent reporter dtTomato in cells that display Nox3 expression [462] (Section 2, Section 3 and Section 4). The group compared the ABR threshold shifts in WT and Nox3-deficient animals after 1, 2 and 6 months after birth. An increase of Nox3 protein in the cochlea as well as increased ABR threshold shifts at frequencies of 8, 24, and 32 kHz occurred in WT mice over time. Nox3-deficient mice showed no ABR threshold shift increase at all. Especially at high frequencies (24 and 32 kHz), the ABR thresholds were higher in WT mice in comparison to Nox3-deficient animals at 6 months from birth. In addition, histologic analysis of the organ of Corti showed that WT mice at 6 months after birth exhibited OHC loss, while hair cell loss in Nox3-deficient mice was significantly lower. These findings suggest that increased Nox3 expression in the organ of Cori leads to OHC destruction and subsequently contributes to age-related hearing loss (Figure 5A).

### 5.2. Role of Nox3 during Vertigo

The only study which investigated a rather harmful effect of Nox3-derived ROS on the vestibular system (in contrast to the crucial function of otoconia formation), was conducted by Zhang et al., who investigated factors that influence benign paroxysmal positional vertigo (BPPV) [873]. BPPV is the most common peripheral vertigo-related disease [874,875] occurring in 2.4% of people [876], which increases with age [877]. BPPV is characterized by the detachment of otolith particles, particle movement into the semicircular canal and subsequent loss of otoconial function [441,878]. BPPV is therefore also termed otolithiasis. In dependency how the proper function of the otoconia is impaired, BPPV can be classified in primary BBPV and secondary BPPV. Primary BBPV is induced by factors that directly damage the otoliths or their surroundings, e.g. hair cell damage or loss, endolymph ion changes, decreased otolith protein secretion and defects in otolith-anchoring proteins [441,879]. Secondary BBPV is defined as damage, which is induced as side effect of other harmful events, such as ear surgery, trauma, ototoxic drugs, Meniere’s disease [880] or vestibular neuronitis [881]. Systemic factors like osteoporosis [882], vitamin D deficiency, hypertension, diabetes or cerebrovascular diseases [883] can also contribute to the severity of this disease. Zhang and colleagues focused on vitamin D deficiency during BPPV, since vitamin D is important for proper Ca^2+^ homoeostasis in general [884,885] and for proper otolith formation and function in particular [881]. Overall, 48 patients with diagnosed BPPV and 48 control patients from the Affiliated Hospital of Inner Mongolia Medical University [886] were analyzed in this study. While no difference in age, body mass index, sex, occurrence of diabetes or hypertension was observed between the groups, BBPV-diagnosed patients showed a decreased bone density and plasma vitamin D levels. Notably, mRNA and protein levels of both OC-90 and Nox3 in the serum were decreased in patients with BPPV. To further analyze the role of vitamin D in this context, vitamin D receptor (VDR)-deficient mice were analyzed. In whole-tissue lysates of the inner ear, mRNA and protein levels of OC-90 and Nox3 were decreased in VDR-deficient mice suggesting a regulatory role of vitamin D in this context. A direct mechanism for VDR-mediated signaling for Nox3-derived ROS production and OC-90 assembly was not investigated.

### 5.3. Role of Nox3 during Lung Diseases

For a long time, Nox3-related research only focused on either the inner ear or studies focused on broad expression studies to improve the catalogue, which lists if, when and where Nox isoforms are expressed. Most of the latter studies have not focused explicitly on Nox3, but rather described its expression as additional finding. Zhang and colleagues investigated, for the first time, a possible connection between Nox3 and pulmonary emphysema, which is a major contributor to chronic pulmonary diseases [887,888] in a mouse model [483]. They described developing emphysemas in naive TLR4- and MyD88-deficient mice beginning at 3 months after birth and peaking between 6 months and 1 year. This was reflected by increased lung volumes, enlarged air spaces distal to the terminal bronchioles and by destruction of the normal alveolar architecture. These factors are typical for emphysema [887] and occurred in both knock-out animal strains. Notably, all mice strains did not show any significant differences in any pro-inflammatory parameter that was analyzed. However, TLR4-deficient animals showed a decreased elastase inhibitory capacity and increased elastolytic activity in the lung tissue. Since increased oxidative distress is an important correlative factor of emphysema [889] and lung injuries [890,891,892,893,894,895,896,897], the group analyzed the total antioxidant capacity, namely levels of glutathione (GSH) and other antioxidant components in the branchio-alveolar fluid. A strong decrease of GSH levels was detected in the fluid of knock-out animals. Moreover, isolated lungs and isolated lung cells from TLR4-deficient animals showed increased O_2_^−^ production in comparison to WT animals. The increased ROS levels further led to more oxidative DNA damage, which is also correlated with emphysema [898]. Interestingly, while Nox3 mRNA was only weakly expressed in WT animal lungs and isolated endothelial lung cells, TLR4-deficient lung samples and lung cells showed an increased Nox3 mRNA expression. Additionally, isolated lung cells from TLR4-deficient animals showed an increased elastolytic activity similar to the lung tissue. Knock-down of Nox3 via siRNA in TLR4-deficient lung cells led to a rescue effect of elastolytic activity, nicely confirming the involvement of Nox3. These results clearly demonstrated the connection of TLR4 deficiency, increased Nox3 expression, Nox3 as cause for the increased elastolytic activity and therefore the developed emphysema. A direct mechanism for Nox3-derived ROS was not investigated at that time. Nevertheless, the study of Zhang and colleagues broke the ”inner ear” stigma of Nox3 in terms of disease developement.

In a follow-up study from Zhang and colleagues, a role of Nox enzymes during hyperoxia was investigated. Hyperoxia can occur during sustained oxygen supply in critically ill patients, which can result in respiratory failure [899,900]. Hyperoxia is also an established model for oxidant-induced lung injury [901,902]. Previous reports of the group demonstrated that TLR4-deficient mice showed increased oxidant production in lung tissue and subsequent lung destruction [483], as well as enhanced susceptibility to hyperoxia-induced acute lung injury [903]. An increase in Nox3 mRNA was also reported in TLR4-deficient animals, and siRNA-mediated knock-down partially rescued the phenotype related to TLR4-deficiency [483]. WT mice exposed to hyperoxia showed increased TLR4 mRNA and protein levels in mouse lung endothelial cells and lung lysates. TLR4-deficient mice were more susceptible to hyperoxia, as reported before [903], but interestingly, Nox3-deficient animals showed an increased survival rate. Additional knock-out of Nox3 in TLR4-deficient animals (TLR4/Nox3 double-deficient mice) nearly rescued the animals comparable to WT controls. Hyperoxia conditions increased macrophage, lymphocyte and neutrophil infiltration into the lungs of WT animals, which was further enhanced in TLR4-deficient animals. Nox3-deficient animals, however, showed no differences compared to WT animals. Notably, TLR4/Nox3 double-deficient animals showed a partial rescue from this phenotype. In WT mice, increased lactate dehydrogenase release as well as increased H_2_O_2_ and lipid peroxidation levels were detected in lungs after hyperoxia exposure. TLR4 deficiency further increased these parameters, while Nox3-defcient animals showed reduced levels in comparison to WT animals. These data nicely show that TLR4 signaling somehow inhibits Nox3-mediated ROS production in lungs, which is uncoupled when TLR4 as regulating factor is missing. The Nox3-mediated uncontrolled ROS production then leads to lung destruction. When Nox3, as an ROS source, is removed, it either protects the mice in general from lung injury during hyperoxia, or it leaves the TLR4-dependent inhibition as the terminal factor without any effect. The group also discovered that the Heat Shock Protein 70 (Hsp70) [904,905] is necessary for the TLR4-mediated Nox3 inhibition, since mice and endothelial lung cells deficient for Hsp70 showed increased Nox3 mRNA and protein levels. Notably, mitochondrial matrix O_2_^−^ levels were decreased in TLR4-deficient lung cells and were not altered in Nox3-deficient cells, excluding mitochondria as a potential ROS source in this setting. In addition, this study firstly investigated possible transcription factors that might influence Nox3 mRNA expression. Chromatin immune-precipitation assays identified regions between −2534/−2360 and −1792/−1498 base pairs upstream of the Nox3 promoter as critical binding sites for STAT3 during Nox3 inhibition. In lungs and endothelial lung cells from endothelial STAT3-deficient mice, more Nox3 expression during both basal and hyperoxia conditions was detected. Electrophoretic Mobility Shift Assay (EMSA) analysis showed that Hsp70 induced the STAT3 binding to the Nox3 promoter region only in WT or Myd88-deficient endothelial lung cells, but not in TLR4- or TRIF-deficient cells. Taken together, this study by Zhang and colleagues is probably the most detailed report about Nox3 activation, regulation and function in a specific context so far. The results were solidified by genetic models and ROS measurements not only in vitro, but also in vivo and no cell type or tissue switching during the study was performed. This is a remarkable example of how to perform a scientific analysis about a Nox enzyme and its functions (Figure 5A,B).

Ruwanpura and colleagues further investigated the role of TLR4 and its adaptors MyD88 adapter-like/Toll/interleukin-1 receptor domain-containing adaptor protein (MAL/TIRAP) [906,907] for normal lung architecture and function in mice [908]. They confirmed the findings from Zhang and colleagues [483], i.e., enlargement of the distal air spaces and destruction of normal alveolar architecture without any inflammation in 6-month-old TLR4-deficient mice. Functionally, they found that the static compliance (pulmonary compliance during the inspiratory pause) was significantly increased in TLR4-deficient mice, which was determined by forced oscillatory technique [909,910,911]. The group further described increased oxidative distress in lung tissue, increased Nox3 mRNA and increased apoptosis of alveolar septal cells. Notably, TLR2 deficiency did not alter any of the observed parameters suggesting a TLR4-specific mechanism in this context.

Yasuoka et al. focused on the influence of ROS during the development of lung fibrosis [502]. During lung diseases, fibrosis is a common side effect, which poses a significant increase in morbidity and mortality in patients [912,913,914]. ROS have been implicated as drivers of fibrosis-related pathophysiology [915,916,917] and lung dysfunction [918,919,920]. Fibrosis is accompanied with tissue remodeling and tissue growth as well as development and is regulated by a plethora of growth factors. Yasukoa et al. focused on the insulin-like growth factor binding protein-5 (IGFBP-5), a prominent factor in this context [921,922], and its connection to lung fibrosis and ROS production. They found that primary human lung fibroblasts increased Nox3 mRNA levels and total cellular ROS production after IGFBP-5 or TGF-β treatment. siRNA-mediated knock-down of Nox3 reduced the ROS production in these cells to baseline levels. However, a role for Nox3-derived ROS in the investigated in vivo setting was not conducted.

The discovery of Nox3 as important player for the progression of lung diseases was furthermore confirmed by a series of genetic screens, which delivered correlative data between the *NOX3* gene and different lung diseases. Tremblay et al. conducted a GWAS to identify candidate genes as predisposing factors for genetic asthma association studies [923]. The scan, in combination with the Genes-to-Diseases computational analysis tool [924,925], analyzed 609 subjects from the Saguenay-Lac-St-Jean founder population in Quebec, Canada [926,927]. Amongst several other genes, the *NOX3* gene was identified as the only NADPH oxidase-related gene. Yin et al. investigated genetic etiology in the context of non-idiopathic pulmonary hypertension (PH) [928,929,930]. Overall, 208 patients were included, 109 patients were diagnosed with non-idiopathic PH and 99 healthy volunteers were included as controls. A total of 143 SNPs were detected in the 109 PH patients with the top hits located in the chromosome 6, precisely in the locus of the *NOX3* gene (SNP termed rs6557421). Notably, PH patients with the detected SNP rs6557421 genotype had a 10-fold-higher risk to develop PH in comparison to healthy control samples. Cantu et al. searched for genetic variations that might increase the risk of primary graft dysfunction (PGD) after lung transplantation by a SNP set analysis [931,932]. Rejection of the grafted lung and subsequent organ dysfunction is a major cause of death during the early transplantational period, affecting up to 30% of all patients [933,934,935]. One of the major pathophysiological aspects associated with PGD is increased oxidative distress occurring during ischemia/reperfusion events [936,937,938,939]. In total, 1039 lung transplant recipients and 392 donors were included in this study, and 314 of the 1038 recipients developed PDG and four genes were identified encoding glutathione peroxidase 1 (*GPX1*), nuclear factor (erythroid-derived 2)-like 2 (*NFE2L2*), nitric oxide synthase 3 (*NOS3*) and glutathione S-transferase mu 2 (*GSTM2*), which all are involved in antioxidant responses [940,941,942]. In the donor group, the genes for Nox3 (*NOX3*), nitric oxide synthase 1 adaptor protein (*NOS1AP*) and paraoxonase 1 (*PON1*) were associated with the development of PGD. Within the *NOX3* gene, the SNP rs3749930 had the strongest association with PGD. The detected SNP marks a nucleotide conversion, which resulted in a threonine to lysine aa substitution in a trans-membrane portion of the Nox3 protein. In addition, several intronic SNPs within the *NOX3* gene were associated with increased risk of PGD.

All of these studies clearly demonstrate a critical involvement for Nox3-mediated ROS production as rather destructive factor during lung diseases.

### 5.4. Role of Nox3 during Cardiovasclar Diseases

The term “cardiovascular diseases“ summarizes a broad catalogue of diseases that affect one or many components of the cardiovascular system directly. This includes the heart or the blood circulation system, but also simply all other organs and parts of the body as well, since oxygen and nutrient supply, mediated by the blood stream, are crucial for proper functioning of the organism. Thus, this topic intervenes with many other diseases, which are affected by the cardiovascular system. Similar to nearly any other disease outcome, as well as during any kind of cardiovascular disease, increased ROS production is a major contributing factor that worsens diseases progression [943,944,945,946,947,948]. Of course, the involvement of Nox enzymes as ROS sources was intensively investigated, including Nox3 [162].

#### 5.4.1. Nox3 and Type 2 Diabetes

While fatty acids are crucial components of cellular membranes, chronically increased levels of FFA, consumed with a HFD (Section 5.1.4) lead to obesity due to excessive depositing in non-adipose tissues, e.g. the liver [949,950,951,952]. Subsequently, the development of insulin resistance [537,571], type 2 diabetes [953] and other hepatic diseases [954] dramatically increases. *Diabetes mellitus* affects more than 300 million people worldwide and represents a disease with high morbidity [955,956]. Type 2 diabetes is associated with various chronic and acute toxic side effects, leading to dyslipidemia, hyperglycemia [957,958,959], diabetic retinopathy [960,961] and chronic hyperinsulinemia. All of these conditions can further induce or enhance adipositas, which is closely related to insulin resistance [561,567,962,963]. Type 2 diabetes and insulin resistance often correlate with increased oxidative distress and an increased systemic pro-inflammatory profile [964] in the according tissues and cells, especially in the liver. Of course, the roles of the Nox isoforms, as primary ROS producers, were investigated in this context [162]. Since Nox3 was identified as an important ROS source in association with diabetic diseases in vitro for HepG2 cells [396,487] (Section 5.1.4) and in a mouse model in vivo [537], further research mostly focused on treatment options. Cremonini et al. investigated the role of the flavanol (-)-Epicatechin [965,966] during HFD-induced insulin-resistance in mice [967]. The group detected a strong up-regulation of Nox3 (60%), Nox4 (274%) and p22^phox^ (237%) protein levels in the liver of mice, which received a HFD in comparison to normally fed mice. Supplemental Epicatechin in the diet prevented this up-regulation. On the in vitro level, similar results were observed in HepG2 cells treated with palmitate and Epicatechin, with exception of p22^phox^, which remained unaltered. The increased expression of Nox3 and Nox4 resulted in an increased total cellular ROS production. No genetic evidence was provided, and only inhibitors for Nox enzymes were used. Therefore, the specific role of Nox3 or Nox4 could not be determined.

Gupta et al. investigated the effects of Pancreastatin (PST) on adipocyte cells in vitro and in vivo [485]. PST is a peptide secreted by neuroendocrine cells [968], which exploits diabetogenic effects, such as glucose uptake inhibition in liver cells [969,970] or the pancreatic β cell response to insulin [971,972]. Accordingly, treatment with PST is associated with insulin resistance, type 2 diabetes and adipositas [973,974,975]. Since increased ROS levels are involved in lipolysis of adipocytes [529,553] and often correlate with type 2 diabetes progression in patients [162,976,977], the effects of PST on the oxidative distress and chronic insulin induced lipogenesis were also investigated in this study. Neither insulin treatment nor PST treatment alone were sufficient for induction of total cellular ROS production in the adipocyte-like cancer cell line 3T3-L1. Combined treatment induced a slight increase of ROS levels. This corresponded with increased Nox3 protein expression and JNK1/2 phosphorylation. An increase of Nox3 protein expression and JNK1/2 phosphorylation was also detected in white adipose tissue of mice with artificially induced insulin-resistance [978]. While these results nicely contributed to previous findings [396,487,537], no siRNA-mediated knock-down of Nox3 or Nox3-deficient animals were used to clearly confirm Nox3 as the responsible ROS source. Building up from their previous study, the group around Gupta and colleagues researched on possible treatment options with the Pancreastatin inhibitor PSTi8 against insulin resistance [979]. Palmitate treatment of HepG2 cells resulted in lipid accumulation, increased Nox3 mRNA expression, total cellular ROS production and decreased glycogen synthesis. All of these effects were reversed by additional treatment of PSTi. PA also induced phosphorylation of JNK1/2 and p38, which was again prevented by PSTi8 treatment. These findings mark PSTi8 as a potential candidate for diabetic treatment. However, since in both studies, Nox3 was not confirmed as a responsible ROS source, especially since Nox4 is also a prominent ROS source in adipocytes [346,980,981], a clear involvement for Nox3-derived ROS remains elusive.

Malik et al. investigated a previously described therapeutic role of Pterostilbene against insulin resistance [982]. Several studies already described anti-cancer and anti-oxidant effects of Pterostilbene [983], which is a methoxylated Reservatrol analogue [984]. An anti-diabetic effect was also described [985,986,987]. A mechanism of action was not investigated yet. Malik et al. treated HepG2 cells with palmitate, which induced cell death, lipid accumulation, Nox3 mRNA expression, total cellular ROS production and lipid oxidation. Additionally, PA treatment increased expression of genes for proteins involved in fatty acid metabolism, i.e., Sterol regulatory element–binding protein (SREBP1c), Carnitine palmitoyl transferase1 (CPT1), a mitochondrial PA transporter and its transcription factor Peroxisome proliferator-activated receptor alpha (PPARα). All of these effects were strongly reduced after additional treatment with Pterostilbene. While anti-oxidant effects were previously described for Pterostilbene, contradictory, the group observed down-regulation of anti-oxidative enzymes after additional Pterostilbene treatment, therefore outruling an anti-oxidative effect in this context. Since no siRNA knock-down of Nox3 was performed a direct effect of Pterostilbene on Nox3-derived ROS production was not investigated.

Type 2 diabetes negatively affects the outcome of wound healing [988,989] and increased ROS levels correlate with chronic open wounds in patients suffering from *Diabetes mellitus* [990]. Kim et al. investigated a possible treatment option for improved wound healing [991] by testing the anti-oxidative substance Edaravone. Edaravone was already in use for treatment of acute cerebrovascular diseases [992]. The group used primary human dermal fibroblasts from patients or healthy controls and used the human keratinocyte cell line HaCaT. Furthermore, they conducted a murine in vivo wound healing experiment [993]. Using this model, the group could analyze the expression of Nox3 in tissue flaps near the wound healing area and observed no differences between normo- and hyperglycemic mice after 5 days of operative wound creation. The addition of fibrin for wound healing stimulation or the application of Edaravone did not change Nox3 protein expression. Since no ROS measurements with siRNA knock-down of Nox3 or Nox3 deficient cells were performed, the role of Nox3-derived ROS during the wound healing process remains elusive.

As in the case of lung diseases [923,928,931], also for cardiovascular diseases, GWAS studies were conducted to identify possible risk factors which might influence the disease outcome [994,995,996]. Radowski et al. performed a GWAS to identify genes related to hypertension in 340 patients with type 2 diabetes [997]. Among the six identified genes, the *NOX3* gene was also detected, which was previously associated with hypertension [998]. Kwak et al. conducted a GWAS of people with type 2 diabetes to broaden the spectrum of factors, which could help identifying risk factors for cardiovascular diseases in general and type 2 diabetes in particular before the disease outbreak occurs [999]. In their pre-print, they described three variants in genetic loci associated with cardiovascular diseases, especially with type 2 diabetes. Among them, on chromosome 6, there was an intergenic variant between the genes *TFB1N* and *NOX3* (SNP termed rs335407).

#### 5.4.2. Nox3 and Adipositas

Similar to type 2 diabetes and insulin resistance, ROS also play a role during the inflammatory settings associated with adipositas [561,1000,1001,1002,1003]. In adipocytes, the presence of Nox3 was reported before [485]. Issa et al. investigated the influence of cytokines on ROS production and lipolysis in the adipocyte-like cell line 3T3-L1 [529]. Treatment with various pro-inflammatory cytokines (TNF, IL-1β, IFN-γ) induced a slight increase in cellular O_2_^−^ production after 8 h in differentiated 3T3-L1 cells. It was previously shown that Nox4-derived ROS play an important role for adipocyte differentiation [346,980,981]. Undifferentiated and differentiated 3T3-L1 cells expressed Nox3 as well as Nox4 mRNA. However, only differentiated cells contained the produced Nox3 and Nox4 proteins. While Nox4 expression remained unaltered after cytokine treatment, Nox3 protein levels strongly increased after 8 h. This study nicely showed a decrease in ROS production via Nox3-knockdown after cytokine treatment. Nox-derived ROS were associated with lipolysis in adipocytes before [553] and, indeed, Nox3 knock-down led to an increased lipolysis in 3T3-L1 cells. On the mechanistic level, the group identified an increased phosphorylation of the hormone-sensitive lipase, an enzyme which mediates lipolysis in adipocytes, at the serine residue 536.

#### 5.4.3. Nox3 and Stroke

Stroke is a major consequence of hypertension [1004,1005], and elevated ROS levels have been associated with cerebral hemorrhage [1006,1007,1008]. Michihara and colleagues therefore investigated the role of Nox enzymes during stroke development [489]. The group analyzed the cerebrum in a spontaneously hypertensive rat (stroke-prone) model (SHRSP) [1009]. These SHRSP animals show lower serum cholesterol levels [1010] and increased levels of oxidized proteins in the aorta, heart, kidney [1011] and brain [1012]. Furthermore, increased 8-OHdG levels in the urine and increased ROS levels in the brain of 16-week-old SHRSP animals were reported [1013]. Increased O_2_^−^ levels, enhanced general Nox activity and increased SOD protein levels were also detected in the brains of SHRSP animals [1014]. Michihara et al. analyzed the mRNA levels of Nox enzymes in the cerebrum of SHRSP animals and found increased mRNA levels of Nox2 and Nox3, while Nox1 and Nox4 were not altered and Nox5 was not detected. Notably, Nox3 protein levels were also increased, while Nox2 levels did not change in comparison to the control animals. This is a nice example that both mRNA and protein levels should always be investigated when suggesting changes in protein presence. However, again, no siRNA-mediated knock-down or Nox3 knock-out model was used to provide evidence that Nox3 is the responsible ROS source for the observed effects in SHRSP animals.

#### 5.4.4. Nox3 and Heart Failure

Several studies have investigated Nox enzymes and their roles for the cardiovascular system in general [102,108,1015] and during human [75,1016,1017,1018], mouse [149,1019,1020,1021] and rat heart failure in particular [186,1022]. While Nox1, Nox2, Nox4 and Nox5 were detected and investigated in this context, the role of Nox3 remained elusive until its detection in murine embryonic stem cell-derived cardiomyocytes by Li and colleagues [149]. The group mainly detected Nox4 mRNA expression, while Nox3 was only weakly expressed and accordingly focused on Nox4. Bkaily and colleagues further analyzed the role of Nox3 in this setting [1023]. For this purpose, they used the hereditary cardiomyopathy hamster model [1024,1025,1026], which is well established for cardiovascular disease studies [1027]. They detected Nox1, Nox2 and Nox4, but no Nox3 protein in the ventricular heart muscles of normal hamsters. In the ventricular heart muscles of cardiomyopathic hamsters, they observed a reduction of Nox1 and Nox4 protein levels and an increase of Nox3 protein, while Nox2 levels remained unchanged. These findings nicely demonstrate that Nox isoforms can show a dynamic expression in dependency of the tissue status. The fluctuation of Nox enzyme expression also demonstrates again that siRNA-mediated knock-down or knock-out experiments are strictly needed when claiming a specific role for a certain Nox enzyme as ROS source. Unfortunately, this was also not conducted in this study.

ROS production is also associated with the pathogenesis of ischemia/reperfusion (I/R)-induced heart injuries [1028,1029] occurring during a myocardial infarction. These injuries include myocardial cell damage and death, arrhythmias or microvascular dysfunction [1030,1031,1032]. Morimoto et al. investigated a putative interplay of ROS and the chemokine monocyte chemoattractant protein-1 (MCP-1) [1033,1034,1035] during I/R [1036]. In vitro experiments with neonatal cardiomyocytes showed that under normoxic conditions MCP-1 had no protective effect. However, after I/R induction, apoptotic cell death increased after 6 h and was reduced by treatment with MCP-1. They used Langendorff-perfused mouse hearts from MHC/MCP-1 mice, which overexpress MCP-1 in the heart for further in vivo investigations [1037]. The group reported an increase of MCP-1 mRNA and ROS production in WT mice after I/R, which was abolished in hearts from MHC/MCP-1 mice. Notably, the group observed mRNA expression of Nox1, Nox2 and Nox3 in the hearts of WT mice, which decreased after I/R. In the hearts of MHC/MCP-1 mice mRNA levels were lower at basal conditions and rose after I/R, again suggesting a dynamic interplay of Nox-derived ROS production. Unfortunately, no Nox silencing or knock-out neither in vitro nor in vivo was performed. Furthermore, no protein expression was analyzed for Nox3. Hence, if and how Nox3-derived ROS production is activated and if these ROS are involved in the context of MCP-1-mediated cardioprotection could not be clarified.

Vats et al. performed a retrospective cohort study [1038] from a population-based Malmö Diet and Cancer Study [1039] with 30,446 subjects over 24.3 years. The group analyzed SNPs to detect genetic variations in genes related to oxidative distress and vitamin intake. The study focused on abdominal aortic aneurysm (AAA) [1040] and unpredictable ruptured AAA [1041,1042], both manifesting in an irreversible and life-threatening dilation of the abdominal aorta [1040,1043,1044]. Accordingly, the study only included participants with occurrence of AAA (25,252 patients in total) [1039]. Oxidative distress has been suggested as a possible link between various factors that contribute to AAA, such as chronic inflammation and cell death [1045,1046], with Nox enzymes as correlated endogenous ROS sources [1045,1047]. During this study, 399 (1.6%) participants were diagnosed with AAA, and 71 (0.2%) were diagnosed with rAAA in general. Furthermore, an amazing effort was made in terms of sub-analytic parameter analysis such as sex, smoking status and physical activity by integrating patient information [1048]. The genetic loci were identified by GWAS and altered SNPs for the *NOX5* gene (rs150003957), and the *NOX3* gene (rs3749930) were detected. The according male patients showed elevated hazard ratios for AAA, while female patients showed no alterations. Furthermore, participants with the dominant *NOX3* gene SNP rs3749930 showed an increased risk for rAAA in the overall study. The group additionally performed subgroup analysis to investigate if the detected oxidative distress-related genotypes had an influence on the effect of antioxidant vitamin intake. They reported that men with the *NOX3* gene variant rs3749930 showed an inverse association between higher riboflavin vitamin uptake and a hazard risk for intact AAA, which was also confirmed for the overall study population after sex covariate adjusting.

### 5.5. Role of Nox3 during Renal Diseases

Chen et al. conducted a GWAS for three phenotypes associated with risk of nephropathy, i.e., serum creatinine levels, creatinine clearance and the glomerular infiltration rate [1049,1050,1051] in 691 type 2 diabetes patients from West Africa to analyze potential factors for reduced renal functions as major consequence of diabetic diseases [998]. The screen detected linkage regions that contain genes, which might influence these three phenotypes. The most prominent candidate genes in these regions that have been implicated in diabetes-induced nephropathy and renal damage were the genes encoding p22^phox^, (linker region 16q24), Nox1 (linker region 10q22) and Nox3 (linker region 6q25.1–6q26). Together with the study from Ye et al. [492], only two studies investigated Nox3 during kidney-related diseases.

### 5.6. Role of Nox3 during Gastrointetinal Disaeses

The most dominant Nox isoform in the gastrointestinal tract is Nox1, which was long termed the “colon NADPH oxidase” [13,81,1052]. Nox1 was also detected in the stomach under normal and disease conditions [82,118,1053,1054]. In addition to Nox1, Duox2 is expressed in the rectum, cecum and ascending colon [92,364,1055], and Nox2 and Nox5 were detected in human gastric samples [1053]. However, so far, Nox3 has not been detected nor associated with the gastrointestinal tract.

### 5.7. Role of Nox3 in Other Diseases

Plantinga et al. investigated genetic variants associated with susceptibility to agranulocytosis [1056]. Agranulocytosis is defined as a reduced concentration of granulocytes in peripheral blood (<500 granulocytes/mL blood) [1057,1058]. Agranulocytosis can be induced by various factors, such as anti-psychotic drugs [1059] or antibiotics [1060,1061], but is also observed in rare events (0.1–0.35%) in patients during treatment with thionamides to medicate hyperthyroidism [1062,1063,1064]. This anti-thyroid drug-induced agranulocytosis (ATDAC) can be a life-threatening condition [1058,1065], especially after the usage of higher doses of anti-thyroid drugs [1066]. During the conducted GWAS of Plantinga et al., two independent families and six patients with Graves’ disease (GD) that developed ATDAC during treatment were analyzed. In 7 out of 11 GD-positive ATDAC patients, a variant of the *NOX3* gene were identified. The group reported that the *NOX3* gene variants p.Asn8Ser, p.Ala198Thr and p.Arg100Ile were absent in ATDAC-negative GD patients and were not detected in previous genetic screens for predisposition to GD [1067,1068]. Notably, all variants were located in regions of the membrane-spanning α-helices of the Nox3 protein.

CO poisoning is a consequence of malfunctioning oxygen supply due to carboxyhemoglobin forming in red blood cells [1069,1070]. The subsequent hypoxia leads to damage in various brain regions, such as the hippocampus or the striatum [1071]. However, several research groups have suggested that hypoxia alone cannot be addressed as solely responsible for the brain damage. The involvement of various ROS subspecies has been discussed by Hara et al. and others as possible damage-inducing molecules in this context [490,1072,1073,1074,1075]. A previous study already detected increased Duox2 mRNA after CO exposure (3000 ppm, 40 min) in the rat striatum [1076], but no mRNA of other Nox isoforms was detected. Hara et al. revised their findings [543] and used their well-established rat model in which CO exposition (1000 ppm or 3000 ppm) [1072,1077,1078] simulates CO poisoning and brain damage [1073,1079]. The group found a small increase in Nox3 mRNA, while Nox1, Nox2 and Nox4 remained unchanged.

Mikkola et al. performed a GWAS for identification of new gene loci associated with canine hip dysplasia [1080]. This canine skeletal disease is a hereditary disorder [1081,1082] of which the severity varies based on genetic variations [1083,1084,1085] and the dog breed [1084,1086,1087]. The group analyzed 750 German shepherd dogs and identified three new genetic loci associated with this disease. One of these newly identified loci is located on chromosome 1 in an intergenic position between the *NOX3* gene and the *ARDI1B* gene. The group identified the SNP BICF2P468585, which showed the strongest association with the disease and which was located approximately 196 kilobases upstream from the *NOX3* gene. Another detected SNP, BICF2S23248027 (also termed rs21911799), was located in the intron between the exons 9 and 10 of the *NOX3* gene.

During a study which investigated the therapeutic effects of Dimethyl fumarate (DMF) on relapsing-remitting multiple sclerosis (RRMS) in 564 participants, Carlströem et al. detected a SNP in the *NOX3* gene associated with a better DMF treatment outcome [1088]. RRMS is an autoimmune disease characterized by the entry of immune cells into the central nervous system (CNS), which leads to pro-inflammatory tissue damage accompanied by neurological dysfunction [1089,1090]. Like in many other autoimmune pathological settings [1091,1092], oxidative distress was reported to be a modulating factor in RRMS [1093,1094,1095]. DMF (Tecfidera^®^) is one of the most prescribed substances for patients that suffer from RRMS [1089,1096]. The identified SNP rs6919626 in the *NOX3* gene allele was associated with a probability of an insufficient DMF treatment response. The group stimulated CD14+ monocytes isolated from patients with the identified *NOX3* SNP rs6919626 with *Escherichia coli* in vitro and detected a reduced total cellular ROS production. This study suggested for the first time a possible link between Nox3-derived ROS and MS disease outcome and treatment.

Li et al. analyzed thyroid tissue samples from 11 patients who suffered from tertiary hyperparathyroidism (THPT) [1097]. Hyperparathyroidism manifests itself by an enlargement of the parathyroid gland, increased levels of circulating parathyroid hormone, as well as disturbed bone and mineral metabolism [1098,1099]. THPT develops during chronic kidney diseases and differs from hyperparathyroidism in an uncontrolled hypercalcemia, i.e., excessive Ca^2+^ levels in the blood [1100]. Since the molecular mechanisms of this process remain largely unknown, Li and colleagues investigated this topic by analyzing blood and thyroid tissue samples from 16 Chinese THPT patients. The group used whole-exome sequencing for the detection of SNPs and insertions or deletions variants. During the screen, 17,401 mutations (6690 missense variants, 3078 frameshift variants, 2005 stop-gained variants and 1630 synonymous variants) were detected in THPT patient samples. From this data set, a further driver mutation analysis identified 179 mutated genes, one of them being the *NOX3* gene. Expression quantification by qRT-PCR additionally revealed decreased levels of *NOX3* gene mRNA in thyroid gland samples from THPT patients.

## 6. Nox3 as Therapeutic Target

Although Nox3 and Nox3-derived ROS are associated with many different diseases (see Section 5), of course, due to the initial discovery in and research focus on the inner ear, most therapeutical approaches targeted Nox3 in this organ [651,1101]. Nevertheless, research was also conducted to develop therapeutic approaches for Nox3-related involvement during diabetes, cancer and MS.

### 6.1. Therapeutic Nox3 Targeting in the Inner Ear

#### 6.1.1. Therapeutic Treatment of Cisplatin-Induced Hearing Loss

The first study which investigated Nox3 as a therapeutical target was conducted by Mukherjea and colleagues [541]. They focused on treatment of cisplatin-induced ototoxicity but did not target Nox3 directly. Instead, the group focused on the protein Transient Receptor Potential Vanilloid 1 (TRPV1), which is expressed in the organ of Corti [1102] and can be activated by ROS [1103]. The group reported cell death of OHCs, IHCs, supporting cells, spiral ganglion cells and the stria vascularis in the rat cochlea after 72 h of cisplatin treatment. Cisplatin treatment increased mRNA levels of both TRPV1 (starting at 24 h and increasing over 72 h) and Nox3 (maximum at 24 h). Further in vitro studies with the UB-OC-1 hair cell line confirmed the in vivo observations. Cisplatin treatment induced TRPV1 protein expression and resulted in a higher channel activity, i.e. an increased Ca^2+^ influx. Nox3 mRNA and protein levels as well as total cellular ROS production were also increased after cisplatin treatment. siRNA-mediated knock-down of Nox3 reduced the cisplatin-induced ROS production, nicely confirming Nox3 as ROS source in this in vitro setting. Moreover, Nox3 down-regulation also decreased TRPV1 mRNA expression. Vice versa, down-regulation of TRPV1 via siRNA reduced the cisplatin-mediated increase of Nox3, suggesting a complex cross-talk between these two proteins. This is not very surprising, since ROS and Ca^2+^ are major factors, which influence various cellular signaling pathways. TRPV1 as a target for siRNA-mediated therapy was subsequently investigated in vivo. For the analysis of hearing loss, ABR threshold measurements were performed in Wistar rats, which were injected with scrambled or TRPV1-specific siRNA application into the cochlea. Animals were then either left untreated or injected with cisplatin (13 mg/kg, i.p.). In control animals, cisplatin treatment increased ABR thresholds at various frequencies (8, 16, 32 kHz) within 72 h, which indicates progressing hearing loss. In rats pre-treated with TRPV1 siRNA the ABR thresholds at 8 and 16 kHz were moderately reduced, while the cisplatin-induced ABR shifts at 32 kHz showed the strongest reduction after TRPV1 siRNA injection. Histochemical analysis of the cochlea showed damage to and/or loss of hair cells in rats after cisplatin treatment in the basal and middle turns of the cochlea, which was reduced after TRPV1 knock-down. While this study nicely establishes TRPV1 as potential target for therapeutic treatment against cisplatin-induced hearing loss, no in vivo experiments for analysis of Nox3 in this context were performed. Nevertheless, the study clearly showed that cisplatin treatment induces Nox3 expression in the inner ear and confirmed Nox3 as ROS source in the hair cell line UB-OC-1 in this context. Encouraged by their previous findings [541], Mukherjea and colleagues made the next logical step and focused on Nox3 as siRNA target [476]. Transfection of siRNA against Nox3 led to a strong reduction of Nox3 mRNA and protein expression in the cochlea of Wistar rats. As previously discovered [541], cisplatin treatment induced mRNA and protein expression of Nox3 in the cochlea, which was reduced after siRNA treatment. Similar observations were made for spiral ganglion cells and the stria vascularis. Furthermore, cisplatin-induced cell death of OHCs, spiral ganglion cells and the stria vascularis, which occurred after 3 days, was reduced after siRNA-mediated knock-down of Nox3. Finally, by analyzing ABR thresholds, the group observed that the cisplatin-induced ABR threshold shift to 35 dB was reduced to 23 dB after additional treatment with siRNA against Nox3. These two studies from Mukherjea and colleagues [476,541] not only confirm Nox3 expression in the rat cochlea, the rather harmful ROS production during ototoxic drug applications and cisplatin-mediated increase of Nox3 expression, but also nicely demonstrate that targeting of Nox3 via siRNA shows promising potentials for a therapeutic treatment of hearing loss.

The group among Mukherjea, Rakumar and colleagues further investigated possible treatment options for cisplatin-induced hearing loss and focused on the increased pro-inflammatory profile in the cochlea reported after cisplatin treatment [804,805,806,807,809]. The group investigated in this context a possible involvement of the cytosolic transcription factor STAT1 [831]. STAT1 is phosphorylated after the detection of various cellular stress factors, including pro-inflammatory cytokines, translocates to the nucleus and regulates expression of iNOS, TNF [1104,1105] and various factors involved in cell death [1106,1107]. The group observed STAT1 phosphorylation and STAT1 translocation into the nucleus after cisplatin treatment in UB-OC-1 cells and in rat cochlea explants. Increased phosphorylation of STAT1 was specifically detected in OHCs, the stria vascularis and spiral ganglion cells. The group nicely identified Nox3-derived ROS as important factor, since Nox3 knock-down via siRNA reduced the phosphorylation of STAT1 in UB-OC-1 cells and in rat cochlea explants. Moreover, the increased pro-inflammatory profile and cell death of UB-OC-1 cells and rat cochlea OHCs, and the induced hearing loss measured by ABR thresholds after cisplatin treatment all could be attenuated by STAT1 knock-down. Unfortunately, a direct protein target for the Nox3-derived ROS on the mechanistic level, which regulates STAT1, was not identified. The latest study from the Ramkumar lab investigated the role of the chemokine C-X-C motif chemokine ligand 1 (CXCL1) in a similar context [1108]. Cisplatin treatment led to CXCL1-mediated signaling in vitro and in vivo finally resulting in increased pro-inflammatory factors and hearing loss, as described before [1109,1110,1111]. Inhibition of CXCL1 signaling showed a protective effect against hearing loss. Among the up-regulated factors induced by CXCL1 signaling, the *NOX3* gene was also described. This study focused on the therapeutic possibilities by targeting the chemokine signaling during cisplatin-induced hearing loss. Therefore, the role of Nox3 in this setting was not further investigated.

Shin et al. designed and investigated a new synthetic compound, named KR-22332, as a treatment for cisplatin-induced hearing loss [542] 2013. Interestingly, the group analyzed zebra fish larvae for their evaluations as addition to the commonly used Wistar rat model. Zebra fish possess hair cells in their lateral line system [1112], which remarkably resembles mammalian inner ear hair cells [1112,1113,1114]. Ototoxic hair cell death can therefore be easily analyzed with this approach [807,1113,1115]. Cisplatin treatment induced significant hair cell loss in zebra fish, which could be reduced by additional treatment with KR-22332. The same results were obtained in vitro with the hair cell line HEI-OC1. Further in vivo experiments with Wistar rats were conducted. Similar to previous findings [355,476,541], the group reported increased cochlear damage and an increase of Nox3 protein levels after cisplatin exposure, which both were reduced by additional K-22332 treatment. Cisplatin-induced hearing loss, determined by an ABR threshold shift measurements at 67 dB, was also reduced to 38.5 dB after K-22332 treatment, altogether suggesting a promising compound for hearing loss treatment after cisplatin exposure. Among steroids, which showed some promising treatment options against cisplatin-induced ototoxicity [1116,1117,1118], the glucocorticoid dexamethasone exploited minimal side effects and protective effects against cisplatin-inflicted damage in the inner ear [1119,1120,1121,1122,1123,1124,1125]. Dinh et al. analyzed the effects of dexamethasone in vitro with rat explants of the organ of Corti [1126]. Cisplatin treatment induced Nox3 mRNA expression, total cellular ROS production and, subsequently, OHC death starting from the basal turn of the organ of Corti after 72 h, while IHCs were not affected. Dexamethasone treatment showed a dose-dependent protective effect in this context.

Kim and colleagues investigated a possible treatment option, which reverses peroxisomal and mitochondrial dysfunction during cisplatin-induced ototoxicity [1127]. They focused on fenofibrate, a pharmaceutical substance usually used to treat unbalanced lipid blood levels [1128]. Fenofibrate already displayed protection against gentamycin-induced ototoxicity [1129] and against cisplatin-induced nephrotoxicity [1130]. Fibrates, such as fenofibrate work mechanistically by binding to the peroxisome proliferation-associated receptor (PPAR), which regulates various cellular functions, mainly the cellular lipid and energy metabolism [1131,1132]. The group investigated hearing loss by ABR threshold shift measurements after cisplatin and fenofibrate treatment and observed, like before, that cisplatin induces higher ABR shifts at all frequency (4, 8, 16, 32 kHz) in mice. Additional fenofibrate treatment alone did not change ABR thresholds. Further analysis of cochlear rat explants showed destruction of OHCs and IHCs, which could be prevented by additional treatment with fenofibrate. These results were confirmed in vitro in the hair cell line HEI-OC1. The group also measured a strong increase in Nox3 and Nox4 protein levels and total cellular ROS production in murine cochlea explants after cisplatin treatment, which was reduced after fenofibrate treatment. They also saw a correlative increase in NF-κB p65 protein levels, which suggests a possible regulatory role for Nox3 expression.

#### 6.1.2. Therapeutic Treatment of Noise-Induced Hearing Loss

Building up on their previous findings, which showed a connection between TVRP1-mediated Ca^2+^ influx and Nox3-mediated ROS [541], as well as noise-induced Nox3-derived ROS production in rat cochleae [741], the group of Mukerjeah and colleagues further investigated the complex interplay of Ca^2+^, ROS and pro-inflammatory cytokines during noise-induced hearing loss [655]. ROS as critical drivers of cochlea damage in general have been described several times before [657,667,690,700]. Noise exposure results in increased Ca^2+^ levels in the cochlea, which, in turn, leads to chronically increased ROS levels [668,1133,1134,1135,1136]. Both factors, permanently increased Ca^2+^ and ROS levels, subsequently lead to an increased pro-inflammatory status of the cochlea [804,805,806], which attracts immune cells that also further progress the inflammation [807,1137,1138,1139]. Dhukhwa et al. focused on the pro-inflammatory cytokine TNF, which was associated with noise exposure in the rat cochlea before [745,1140,1141] as a possible therapeutic target. After noise exposure, the group measured ABR threshold shifts from 25 to 50 dB at frequencies of 8, 16 and 32 kHz in Wistar rats. This was associated with increased mRNA and protein expression of TRPV1, Nox3, TNF, COX2 and iNOS in the rat cochlea. Additional treatment of animals with capsaicin, the typical agonist of TRPV1 [1142], strongly reduced the mRNA and protein expression of mentioned proteins. Sequestration of TNF by treatment of animals with Etanercept, an IgG1 receptor covalently linked to two TNF receptors [806,1143], reduced TNF and Nox3 protein levels as well as ABR threshold shifts. Early administration (first 2 h of noise application) showed an even stronger otoprotection. However, neither in the in vivo model, nor in the in vitro cell culture experiments, Nox3 knock-out or knock-down animals or cells were used to provide evidence of Nox3 as activated ROS source. Moreover, the possibility that the noise application itself might activate the ROS production of Nox3 alone was not investigated, since no ROS measurements were performed.

After Oishi et al. successfully down-regulated Nox3 expression in the cochlea via direct injection into the murine inner ear [1144], Nacher-Soler and colleagues targeted Nox3 via siRNA in vivo as therapeutical option against sensorineural hearing loss [716]. The group developed a screening method for detecting especially effective Nox3-directed siRNA by establishing a co-expression system. This system resembles early research studies during discovery of the enzyme (Section 1), in which Nox3, p22^phox^, NOXO1 and NOXA1 of either mouse or human origin, were expressed in the cell lines HEK239, HeLa and CHO. ROS production measured by a water soluble triazonium salt (WTS) reduction assay was used as Nox activity output. Using this biomarker assay, the group identified two potent siRNAs out of ten tested in total, which showed strong down-regulation of Nox3 at very low concentration ranging between 0.1 and 1.13 nM. In mouse cochlear explants, a concentration of 80 nM was necessary to induce a reduction of Nox3 expression to 50% after 48 h. The siRNA #248 showed the most potent effect of the two selected siRNAs and fully matched with the human Nox3 sequence. In vivo delivery of siRNA via intracochlear injection resulted in 60% down-regulation of Nox3 siRNA in the mouse and might therefore bear a relevant human therapeutic approach.

Rousset et al. focused on the new therapeutical possibilities of in vitro designed microRNAs [872]. Like shRNA- or siRNA, microRNA-mediated knock-down of the targeted mRNA is commonly used techniques to investigate cellular processes [1145,1146,1147,1148] and depicts new opportunities for therapeutically uses in patients [1149,1150,1151]. Rousset and colleagues addressed the optimization of miRNAs for a better therapeutical use in general [1152,1153,1154] and chose the subunit p22^phox^ of the Nox isoforms Nox1-4 as one of the therapeutic targets. Indeed, they reported a decrease in p22^phox^ mRNA expression in hair cells, after transduction with an optimized miRNA. However, by only targeting p22^phox^ and not Nox3 specifically, this therapeutic approach will target all Nox isoforms, which can show a highly fluctuating expression profile in dependency on the biological context [6,361,505].

### 6.2. Therapeutic Nox3 Targeting as Diabetic Treatment

Type 2 diabetes is accompanied by vision loss due to diabetic retinopathy, which is a major complication in diabetic patients [960,961,1155,1156]. Vision loss during diabetic retinopathy is caused by a loss of pericytes and vascular endothelial cells, which leads to vascular dysfunction and neurological inflammation [1157]. There are several treatment options available already, such as the application of anti-VEGF or PKC inhibitors [1158,1159], which are, however, not fully satisfactory due to the complex processes involved in diabetes, such as hyperglycemia and increased oxidative distress [1160,1161,1162]. Cai et al. investigated the glucagon-like peptide 1 (GLP-1), an insulin tropic peptide, which showed potential for diabetes treatment [1163,1164] due to its anti-oxidative properties [1165,1166]. They induced type 2 diabetes in Wistar rats by applying a HFD to investigate this topic. The group described high glucose levels, reduced thickness of retinal cellular structures, namely the columnar and cone photoreceptors, the outer and inner nuclear layer, the inner plexiform layer and the retinal ganglion cell layer and an increased apoptotic cell death of the according retinal cells. All of these parameters were reduced after treatment with GLP-1. The authors also investigated a possible role of Nox3, however, only immunohistological staining of WT retinal explants was performed. A proper negative control staining for Nox3 in Nox3-deficient samples was not conducted. Furthermore, no mRNA or protein level expression was analyzed from retinal lysates to further validate Nox3 involvement and no ROS production was performed. As a result, neither the involvement nor the role of Nox3-derived ROS could be made in this study.

### 6.3. Therapeutic Nox3 Targeting during Cancer

Saleem et al. investigated a possible therapeutic role of Brevilin A, a plant-derived sesquiterpene lactone [1167,1168,1169], against breast cancer cells [548]. The group used the human breast cancer cell line MCF-7 in this context and reported a dose-dependent reduction of migratory abilities, induction of cell cycle arrest and subsequent cell death after Brevilin A treatment. Further supplementation of the globally working ROS scavenger NAC led to reversed effects suggesting the general involvement of ROS. Indeed, the group measured an increase in total cellular ROS levels after Brevilin A treatment starting at 1 hour and reaching its peak after 2 h. Notably, the group measured an increase of Nox2 and Nox3 protein levels after Brevilin A treatment and therefore suggested the involvement of these two Nox isoforms. However, in comparison to the ROS production, which peaks after two hours, the increase of Nox2 and Nox3 was reported earliest after 24 h, which is too late to explain the described early and quick ROS burst. Moreover, no Nox3-deficient cells or siRNA-mediated knock-down of Nox enzymes was conducted. Therefore, the role of Nox3 in this setting remains elusive.

### 6.4. Therapeutic Nox3 Targeting during Multiple Sclerosis

Choi et al. investigated an agonist of the lysophosphatidic acid (LPA) receptors as possible treatment option in multiple sclerosis (MS) [1170]. ROS overproduction plays a critical role for the pathological development of MS [1171,1172,1173,1174], e.g., disruption of the blood–brain barrier or acceleration of trans-endothelial migration of peripheral immune cells into the CNS, which further lead to tissue damage. Therefore, ROS indirectly contribute to lesion persistence and deterioration in MS. Several reports have mentioned an induction of ROS production after LPA treatment and subsequent signaling after binding to plasma membrane-located receptors [1175,1176,1177]. LPA functions as both a plasma membrane component and an extracellular signaling mediator in various tissues [1178]. As signaling molecule, LPA induces various processes, such as cell survival, angiogenesis, neurogenesis, and neuroplasticity in the nervous system [1178,1179,1180]. Choi et al. tested the LAP receptor antagonist Ki16425 on MS development and the role of ROS in this context. The group therefore used an established MS model in mice [1181]. Treatment with Ki16425 deteriorated the motor disability, spinal demyelination, enhanced the infiltration of immune cells, such as microglia and Th1 or Th17 helper cells into the spinal cord [1182] and progressed blood–brain barrier disruption [1183]. These events massively worsened the MS symptoms in treated mice. The group also detected increased levels of pro-inflammatory cytokines and of 4-Hydroxynonenal, a common marker of oxidative distress [1184], in the spinal cord of Ki16425-treated animals. They also reported increased mRNA levels of Nox2 and Nox3 after high doses Ki6425-treatment. On the contrary, treatment with the LPAR agonist 1-oleoyl-LPA alleviated the described parameters including Nox3 and Nox2 mRNA expression in the spinal cord of treated animals. While the effects of LAP-mediated signaling on the MS disease outcome were clearly demonstrated in this study, no evidence was given that Nox3-derived ROS were responsible for the observed enhanced oxidative distress in the spinal cord.

## 7. Concluding Remarks

According to PubMed there are 192 articles, which mention Nox3. The detailed knowledge about this Nox isoform, however, is surprisingly low. With exception of a few ground-breaking milestone articles, most of the studies have only described correlative increase or decrease of Nox3 mRNA in their research context. Confirmation of Nox3-derived ROS as an involved physiological factor on the genetic level or ROS measurements as representative enzymatic output were rare events, on average, of all conducted studies. Despite the fact that Nox3 is expressed not only in the inner ear but also in various cell types and organs, the “inner ear stigma” remains until today. Because of that, Nox3 might be the most underrated Nox isoform to date. Therefore, this review should not only be a helpful compendium of Nox3-associated research but should also function as an encouraging call to all researchers interested in Nox enzymes and Nox-dependent ROS production to focus more on this Nox isoform. The roles and functions of Nox3 are not just limited to the inner ear but extend far beyond it.

## Figures and Tables

**Figure 1 antioxidants-13-00219-f001:**
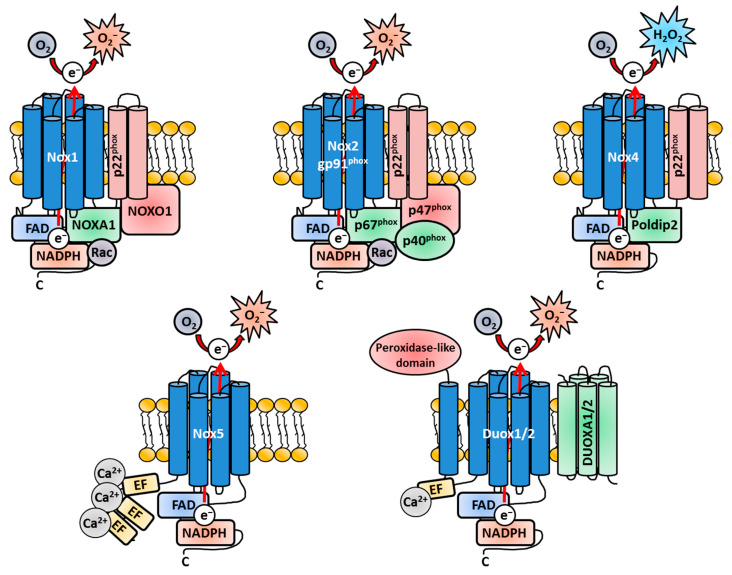
The enzyme family of NADPH oxidases (Nox) consists of seven members, namely Nox1, Nox2, Nox3, Nox4, Nox5 and the Dual oxidases Duox1 and Duox2. The core structure consists of six α-helical domains. At the C-terminal end of the core, the structure binding regions for nicotinamide adenine dinucleotide phosphate (NADPH) and flavin adenine dinucleotide (FAD) are located, which mediate electron delivery and translocation. The isoforms strongly vary in terms of activation and utilization of adaptor subunits necessary for ROS production. Nox1 recruits the organizer subunit NADPH oxidase organizer 1 (NOXO1) and the activator subunit NADPH oxidase activator 1 (NOXA1). Nox2 needs p47^phox^ as organizer and p67^phox^ as activator subunit, which both are tethered together via the scaffold-like protein p40^phox^. Nox1 and Nox2 strictly also need the guanosine triphosphate phosphohydrolase (GTPase) Ras-related C3 botulinum toxin substrate (Rac) for full activity. Without these adaptor subunits, both Nox isoforms are quiescent. Nox4, on the contrary, is constitutively active, but the ROS production can be regulated either via changes in Nox4 protein expression or via regulatory factor polymerase (DNA-directed) delta interacting protein 2 (Poldip2). The structural membrane-bound subunit p22^phox^ is crucial for enzymatic activity and maturation of Nox1, Nox2 and Nox4. Nox5, Duox1 and Duox2 do not need p22^phox^ or any activating subunit but are both activated by Ca^2+^-binding to their cytosolic EF-hand domains. The stabilizing factors Dual Oxidase Maturation Factor 1/2 (DUOXA1/2) are crucial for the maturation and transportation of Duox1/2 but not for activation of the enzymes. Duox1 and Duox2 also contain a peroxidase-like domain on the extracellular side, which utilizes H_2_O_2_ for oxidation.

**Figure 2 antioxidants-13-00219-f002:**
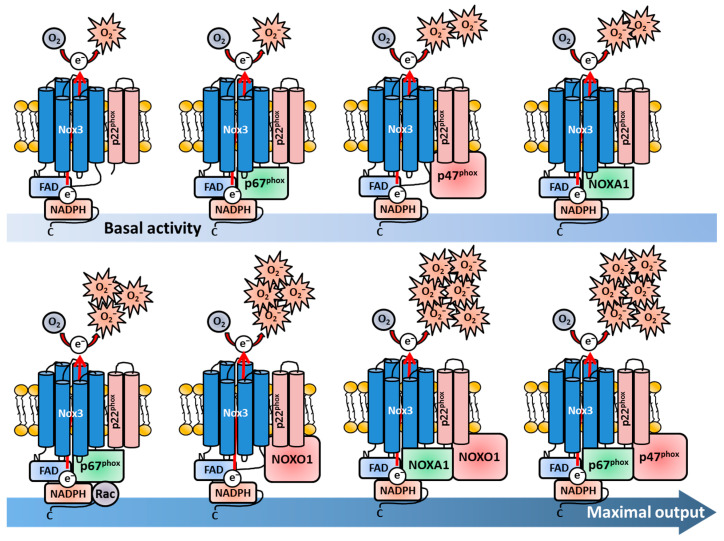
Nox3 is unique among the Nox isoforms since it shows a robust and constitutive ROS production without any organizer or activator subunit, similar to Nox4. Nevertheless, like Nox1, Nox2 and Nox4, Nox3 critically needs the membrane-bound subunit p22^phox^ for enzymatic activity as well as proper protein synthesis and cellular localization. Remarkably, Nox3 shows the most flexible possibilities of adaptor subunit usage among all Nox isoforms. Nox3 can utilize the adaptor subunits of both Nox1 and Nox2 in any thinkable combination. In human cells, the combination of adaptor subunits strongly affects the ROS production of Nox3. The subunits of Nox2, namely p67^phox^ or p47^phox^, either not or only weakly enhance Nox3-derived ROS production, respectively. The same applies for the activator subunit of Nox1 NOXA1. While Rac is crucial for Nox1 and Nox2 activation, it is not needed for Nox3 activation per se, but, in combination with p67^phox^, it can enhance Nox3-derived ROS production. The Nox1 organizer subunit NOXO1 induces the strongest ROS production, which can be initiated by a single adaptor subunit together with Nox3, while the combination of either the Nox2 or the Nox1 adaptor subunits together lead to maximal ROS output by Nox3.

**Figure 3 antioxidants-13-00219-f003:**
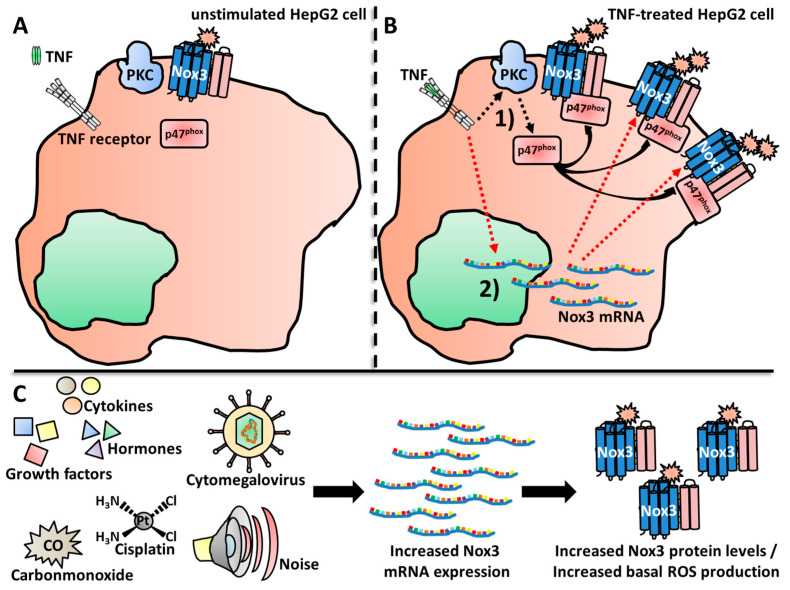
So far, only a few studies described detailed pathways, which lead to Nox3 activation and subsequent ROS production. (**A**) Li et al. demonstrated that Nox3 shows basal ROS production in unstimulated HepG2 cells. (**B**) Nox3-derived ROS production can be enhanced via tumor necrosis factor (TNF) by two different mechanisms. (**1**) TNF- and protein kinase C (PKC)-mediated signaling results in the translocation of p47^phox^ to the Nox3 core complex and subsequently activates Nox3-derived ROS production directly and (**2**) as many other exogenous or endogenous factors; also, TNF signaling leads to the up-regulation of Nox3 mRNA expression, protein synthesis and finally the increase of ROS production [396]. (**C**) Similar to TNF, other endogenous factors, such as growth factors or hormones, as well as exogenous factors like carbon monoxide, cisplatin or noise, lead to the up-regulation of Nox3 mRNA expression and subsequent increase of the Nox3 protein, which is often correlated to an increased ROS production.

**Figure 4 antioxidants-13-00219-f004:**
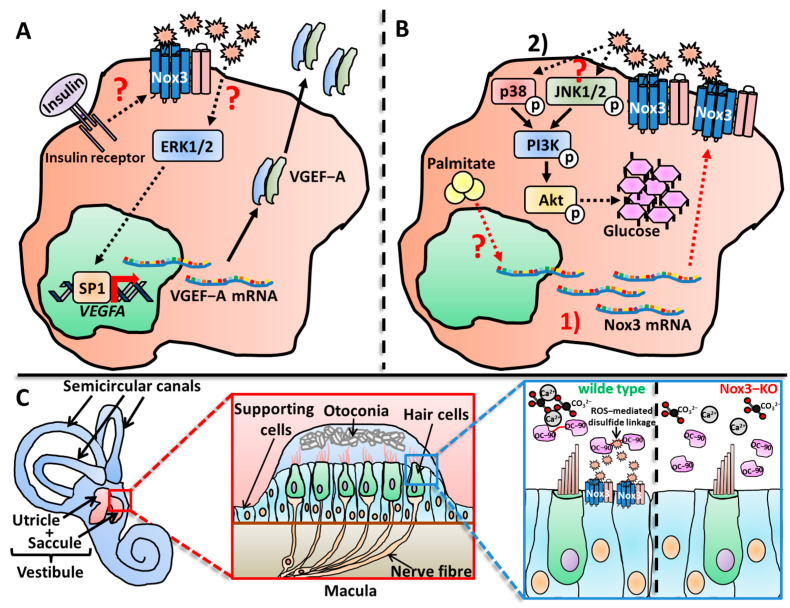
Nox3-derived ROS fulfill various important functions in the organism. (**A**) Insulin treatment of HepG2 cells leads to Nox3-derived ROS production, which, in turn, induce a extracellular signal-regulated kinase 1/2 (ERK1/2)-mediated translocation of the transcription factor Specific protein 1 (Sp1). Sp1 binds to the promoter of the *VGEFA* gene and induces expression of Vascular Endothelial Growth Factor (VGEF)-A mRNA and VGEF-A protein production. Neither the exact steps of the insulin-induced signaling cascade, which activates Nox3, nor the exact targets of Nox3-derived ROS that activate the ERK1/2 pathway are known in detail [487]. (**B**) Treatment of HepG2 cells with palmitate induces (**1**) Nox3 mRNA expression and protein synthesis by an unknown mechanism. The increased basal Nox3-derived ROS production then (**2**) activates a signaling cascade, which involves phosphorylation of the mitogen-activated protein kinases (MAPK) C-Jun-N-terminal Kinase 1/2 (JNK1/2), p38, phosphoinositide 3-kinases (PI3K) and the protein kinase B, also known as Akt, which ultimately leads to gluconeogenesis [537]. (**C**) In the inner ear, the vestibular system is responsible for detection of acceleration and gravity sensing. Three semicircular canals detect and rotational acceleration. In the vestibule, consisting of the saccule and the utricle, changes in gravity and linear acceleration are detected. In the vestibule, the maculae are responsible for this function. A gelatinous matrix is located on top of the maculae. Otoconia, solid crystalline structures, are formed in this matrix directly above the sensory hair cells. Movement of otoconia in this matrix stimulates the hair cells, which transfer the sensory information to the ganglion cells. The main inorganic compound in otoconia is calcium carbonate (CaCO_3_) but otoconia are not completely inorganic. Various proteins are necessary for proper otoconial formation. The major protein component is otoconin 90/95 (OC-90/95) [429,430,431], which is produced by the non-sensory epithelial cells. OC-90/95 is crucial for proper formation of the inorganic CaCO^3^ crystallites. The most current mechanism describes Nox3-derived ROS as crucial mediators of disulfide linkage and subsequent multimerization of OC-90. The OC-90 multimers then function as nucleation points for calcium ions (Ca^2+^) and CO_3_^2−^ to form CaCO^3^. Without Nox3-derived ROS, no OC-90 multimers are present as nucleation points, Ca^2+^ and CO_3_^2−^ remain in solution and otoconia are not formed [464,466]. The lack of otoconia leads to the most obvious phenotype of Nox3-deficient mice, i.e., strong balancing deficits [370].

**Figure 5 antioxidants-13-00219-f005:**
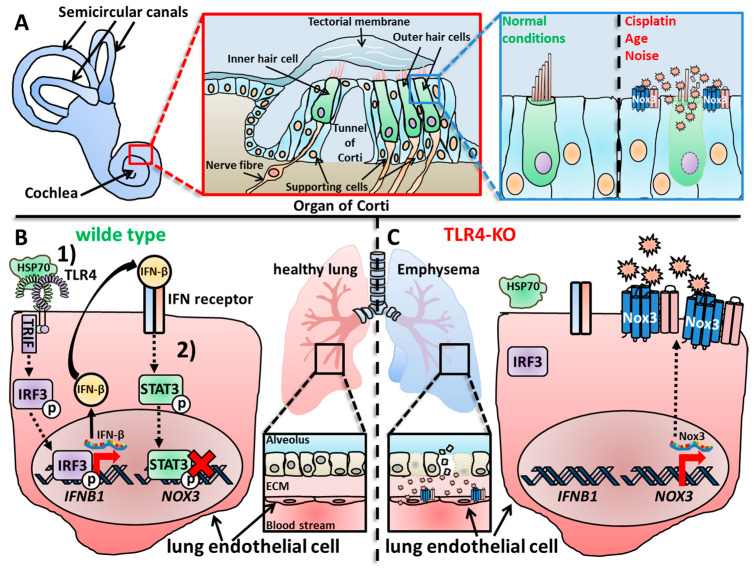
Overproduction of ROS or ROS production in the wrong location can lead to oxidative distress, cellular damage, malfunctioning of tissues and organs and finally manifest in diseases. (**A**) The cochlea is the organ responsible for hearing and in contrast to the vestibular system, loss of Nox3 leads to a rather protective outcome for the tissue and the hearing capacity. In the cochlea, the organ of Corti is responsible for detection of sound waves and neuronal processing. For that, outer and inner hair cells detect movements of the tectorial membrane, which are induced by incoming sound waves. Under healthy conditions, hair cells and supporting cells function normally; however, under exogenous or endogenous stress conditions, supporting cells up-regulate Nox3. The subsequent ROS overproduction leads to hair cell death and contributes to age-, noise- and drug-induced hearing loss [462,476,478]. (**B**) In WT mice, the development of lung emphysemas with increasing age is inhibited by a complicated signaling cascade in lung endothelial cells [395,483]. (**1**) Heat shock protein 70 (Hsp70) activates Toll-like receptor 4 (TLR4)-mediated signaling, which finally leads to activation and translocation of the transcription factor Interferon regulatory factor 3 (IFR3) into the nucleus. IFR3 induces the expression and production of Interferon-β (IFN-β), which is subsequently secreted and (**2**) activates the lung cells via binding to the IFN receptor in an autocrine manner. The IFN receptor-induced signaling cascade results in activation and translocation of the transcription factor Signal transducer and activator of transcription 3 (STAT3), which then binds to the promoter of the *NOX3* gene in result inhibiting the expression of Nox3. (**C**) In TLR4-deficient animals, this autocrine signaling cascade does not activate, which leads to increased mRNA expression and synthesis of Nox3 and subsequently to an increased ROS production of lung endothelial cells. The accumulating oxidative damage results in destruction of the alveolar structures and subsequently to the development of lung emphysemas in TLR4-deficient mice observed with increasing age.

## Data Availability

Not applicable.

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
