# Peer review of "NADPH Oxidase 3: Beyond the Inner Ear"

_antioxidants, 2024, doi:10.3390/antiox13020219_

Round 1

Reviewer 1 Report

Comments and Suggestions for Authors

This is a comprehensive review of literature related to Nox3, with an interesting perspective on the possible beneficial effects of Nox-3-derived ROS in the cochlea. The title is a bit misleading, given the amount of space devoted to studies of Nox in the inner ear. Instead of "NADPH-oxidase 3: Beyond the inner ear", a better title would be, "NADPH oxidase: Inner ear and beyond."  Alternatively, if this review is meant to focus on non-inner ear findings related to Nox 3, then the sections on the inner ear can be greatly condensed and the manuscript shortened considerably. Either way, the readability of this lengthy review can be improved substantially by correcting the overuse and misuse of commas. Additionally, the text needs careful editing for errors made in terminology and phrasing.

A few such errors are noted below:

L44, “felt” should be “fell”?

L66, author uses term “oxidative distress” which is fine, but most often the term used is “oxidative stress”. Perhaps use both here, “oxidative stress/distress” and thereafter just your preferred term, distress?

Throughout: why is “stria vascularis” italicized?

1.1.1. It would be helpful to refer to Figure 1 early (around L74) instead of at the end of the text description

L163, “as the most representative”

L239, “aroused” is not the word you need here

L256, “led to the highest”

L331 and “PX motive” should be “PX motif”?

L412 “by a co-expression”

L420, organisator?

L479, “as the first”

L500, “concert”??

L548, B57BL/6J mice should be C57BL/6J mice

L550, here and elsewhere, ”et al” should be “et al.”

L613, “as a location”

L638, extra “in”

L643, extra “detected in”

L663, hair cell not “hear cell”

L672, L673, 679, L943, sulcus cells, not “suculus cells”

L679 and many other places, “Organ of Corti” should be “organ of Corti”

L689 and L690 Claudius cells, not “Claudians cell”

L1259, citation not available?

L1303, “rotarod” should be “rotorod”

L1305, “did not adapt [not adapting]”

L1308, what is meant by “neuronal relays” of the vestibular nerve?

L1309, “where [not were] otoconia are located”

L1310, severe not sever

L1311, deficient not deficient

L1358, tatic??

L1376, “involvement of ROS”

L1401, auditory not auditorial

L1465, cases of hearing loss, not cases of hearing.

L1465, rephrase this sentence; sound at 85 dB is moderate, not extreme or severe

L1471 and elsewhere, severity not severeness

L1472, can also be influenced

L1480, sensitivity not sensibility

L1536, cell bodies of peripheral auditory neurons are located in Rosenthal’s canal, so “peripheral auditory neurons in Rosenthal’s canal”?

L1542 and L1544 are contradictory regarding 32 kHz differences

L1552, change to “results in increased oxidative stress and damage”

L1681, change “can be further progress by”

Figure 5 caption needlessly repeats information in text

Comments on the Quality of English Language

overuse and misuse of commas greatly decreases readability of this review

other: please see above

Author Response

Point-to-point response to reviewer 1

Reviewer 1:

This is a comprehensive review of literature related to Nox3, with an interesting perspective on the possible beneficial effects of Nox-3-derived ROS in the cochlea.

Response: First of all, I would like to thank this reviewer for the careful evaluation of my work and the positive feedback.

The title is a bit misleading, given the amount of space devoted to studies of Nox in the inner ear. Instead of "NADPH-oxidase 3: Beyond the inner ear", a better title would be, "NADPH oxidase: Inner ear and beyond." 

Response: I carefully analyzed the proportions of text parts, which solely focus on Nox3 and the inner ear (vestibular system and cochlea). Throughout the manuscript there are 15 ½ pages that focus on this matter (pages 11-14, 27-29, 30-38, 46, 49). In relation to the total page number of 50, the inner ear part makes up 31% of the total manuscript. Therefore I would like to keep the title as it is.

Alternatively, if this review is meant to focus on non-inner ear findings related to Nox 3, then the sections on the inner ear can be greatly condensed and the manuscript shortened considerably.

Response: This suggestion is appreciated; however, this review focuses neither only on the inner ear nor only on any specific other part, but wants to summarize all the research on Nox3 published to date. The reviewer is of course completely right by stating that Nox3-related research strongly focuses on the inner ear so far. Accordingly, since many readers will focus on Nox3 in the context of the inner ear in this review, sufficient information about the inner ear still has to be given. Based on this, I would politely decline and leave the sections about the inner ear unchanged.

  1. Either way, the readability of this lengthy review can be improved substantially by correcting the overuse and misuse of commas. Additionally, the text needs careful editing for errors made in terminology and phrasing.

Response: At this point, I want to express my gratitude for bearing with my writing and spelling errors. This is the first article that I completely conceived, wrote and financed by myself. Accordingly, for one person it is highly possible that errors in spelling, phrasing and punctuation occur in a lengthy article even after several rounds of screening. Again, I deeply thank the reviewer for the careful checking of the text and suggested corrections. I shortened or broke up many too long sentences. Some parts were considered unnecessary and were deleted completely.

A few such errors are noted below:

L44, “felt” should be “fell”?

Response: Corrected.

L66, author uses term “oxidative distress” which is fine, but most often the term used is “oxidative stress”. Perhaps use both here, “oxidative stress/distress” and thereafter just your preferred term, distress?

Response: Thank you for pointing this out. The reviewer is completely right, since oxidative stress describes an oxidative cellular milieu in general. The specific terms “oxidative eustress” for beneficial effects and “oxidative distress” for detrimental effects further describe this condition. I added a few sentences for clarification and used the term “oxidative distress” throughout the manuscript.

Throughout: why is “stria vascularis” italicized?

Response: I found the italicized version in some old publications, but the reviewer has a good point, since I did not right any other specific term in italics (like the organ of Corti) and also recent publications did not write it in italics. I corrected this issue throughout the text.

It would be helpful to refer to Figure 1 early (around L74) instead of at the end of the text description

Response: This suggestions is appreciated and was conducted.

L163, “as the most representative”

Response: Corrected.

L239, “aroused” is not the word you need here

Response: True. I changed it to “emerged”.

L256, “led to the highest”

Response: Corrected.

L331 and “PX motive” should be “PX motif”?

Response: Completely right. Corrected.

L412 “by a co-expression”

Response: Corrected.

L420, organisator?

Response: Corrected to organizer. Thank you.

L479, “as the first”

Response: Corrected.

L500, “concert”??

Response: Changed to “coordinate”.

L548, B57BL/6J mice should be C57BL/6J mice

Response: Corrected.

L550, here and elsewhere, ”et al” should be “et al.”

Response: Yes, thank you very much. It already came to my attention and I carefully checked the whole manuscript for this error.

L613, “as a location”

Response: Corrected.

L638, extra “in”

Response: Corrected.

L643, extra “detected in”

Response: Corrected.

L663, hair cell not “hear cell”

Response: Corrected.

L672, L673, 679, L943, sulcus cells, not “suculus cells”

Response: Corrected. Thank you very much.

L679 and many other places, “Organ of Corti” should be “organ of Corti”

Response: Corrected. Thank you.

L689 and L690 Claudius cells, not “Claudians cell”

Response: Corrected to “Claudius’ cells”.

L1259, citation not available?

Response: Yes, at least not on PubMed or as a normal published article. There is a free PDF of a short descriptive report of the mutant mouse line available for download on the Jackson Laboratory homepage. I rephrased the text passage to avoid confusion.

L1303, “rotarod” should be “rotorod”

Response: I used this word based on the scientific literature

https://app.jove.com/v/20782/the-accelerating-rotating-rod-assay-or-rotarod-test-a-method-to-test-motor-coordination-and-learning-in-mice

https://bmcbiol.biomedcentral.com/articles/10.1186/s12915-023-01679-y)

and the Wikipedia article (https://en.wikipedia.org/wiki/Rotarod_performance_test).

L1305, “did not adapt [not adapting]”

Response: Corrected.

L1308, what is meant by “neuronal relays” of the vestibular nerve?

Response: This text formulation was wrong. The sentence was rephrased to “The vestibular nerve and its information relay to the subsequent neuronal network depend on…”.

L1309, “where [not were] otoconia are located”

Response: Corrected.

L1310, severe not sever

Response: Corrected.

L1311, deficient not deficient

Response: Already corrected.

L1358, tatic??

Response: Corrected to “static”.

L1376, “involvement of ROS”

Response: Corrected.

L1401, auditory not auditorial

Response: Corrected.

L1465, cases of hearing loss, not cases of hearing.

Response: Corrected.

L1465, rephrase this sentence; sound at 85 dB is moderate, not extreme or severe

Response: Thank you for this correction.

L1471 and elsewhere, severity not severeness

Response: Corrected.

L1472, can also be influenced

Response: Corrected.

L1480, sensitivity not sensibility

Response: Corrected.

L1536, cell bodies of peripheral auditory neurons are located in Rosenthal’s canal, so “peripheral auditory neurons in Rosenthal’s canal”?

Response: I rephrased the sentence. Thank you very much.

L1542 and L1544 are contradictory regarding 32 kHz differences

Response: Indeed. Thank you very much. The sentence was corrected.

L1552, change to “results in increased oxidative stress and damage”

Response: The sentence was changed accordingly.

L1681, change “can be further progress by”

Response: Changed to “can further progress”.

Figure 5 caption needlessly repeats information in text

Response: This is true. However, Reviewer 2 on the other hand demanded Figures and Figure captions that “stand alone” and can be understood without the main text. I tried to “streamline the caption a little bit though. Moreover, especially the complex topic of panel C needs again a detailed description to understand the molecular mechanism of Nox3-derived ROS in this context.

Comments on the Quality of English Language

overuse and misuse of commas greatly decreases readability of this review

other: please see above

Response: I thank the reviewer very much for the laborious and detailed evaluation and correction of my manuscript. I carefully tried to remove any unnecessary words and commas in the text to increase the readability.

Reviewer 2 Report

Comments and Suggestions for Authors

With more than 50 text pages excluding the references and more than 1100 references, this comprehensive review fits with the scope of the journal and undoubtedly assists the interested reader with extensive information. The review is well-structured and well-written. I have the following few remarks. Apart from that, impressive work.

Figures: Figures are crucial in such a long review for the reader who wants to grasp the main message without reading the entire text. Therefore, they should "stand-alone". Please, revise the figure legends and spell out all abbreviations used in the figure (only as an example: FAD, EF, etc. in figure 1) and describe all elements shown in the figure. White labelings, especially white on rose (p22phox), are hard to read.

In figure 2, the two distinct blue arrows seem to refer to two distinct processes, instead of a continuum of the same process. Also, red on dark blue is hard to read. 

Figure 3 is cut on the right side. Panel B is not mentioned in the figure legend. Would processes 1) and 2) eventually converge in augmenting nox3 protein at the plasma membrane level? This is not shown in figure 3. 

Figure 4, text in panel c is too small

Figure 5, "Tunel" should be spelled "Tunnel"

Minor comments

- there are sometimes numbers in the main text that appear to be unformatted references or publication years, please double-check lines 147 and 455

- Period (.) missing at lines 191 and 386

- Lines 1901-1902, references 915 and 916 do not belong to Yasukoa, please revise. 

- Comments on language/grammar/style are given in the box below. 

Comments on the Quality of English Language

Some sentences are really long and hard to follow. Please, revise the abstract and conclusions for this issue. The use of parentheses is discouraged. Please revise the sentence in the abstract: "the discovery .... was a surprising discovery". 

Some sentences are rather unclear and should be rephrased:

lines 495-497

lines 570-572

lines 899-902

line 643, please correct the repetition

line 646, please correct as follows: Nox3 is expressed in avian liver

Line 783: amnichorions > amniochorions

Line 2210 and 2404, dervied > derived

Line 2285, meidated > mediated

Line 2316, disbalanced > unbalanced

Line 2323, frequency > frequencies 

Line 2343, progress > progresses

Line 2362, an > a 

Line 2366, bio marker > biomarker

Line 2388, lead do > leads to

Line 2448, mile stone > milestone

Author Response

Point-to-point response to reviewer 2

Reviewer 2

With more than 50 text pages excluding the references and more than 1100 references, this comprehensive review fits with the scope of the journal and undoubtedly assists the interested reader with extensive information. The review is well-structured and well-written. I have the following few remarks. Apart from that, impressive work.

Response: First of all, I would like thank this reviewer for the careful evaluation of my work and the highly encouraging and positive feedback.

Figures: Figures are crucial in such a long review for the reader who wants to grasp the main message without reading the entire text. Therefore, they should "stand-alone". Please, revise the figure legends and spell out all abbreviations used in the figure (only as an example: FAD, EF, etc. in figure 1) and describe all elements shown in the figure. White labelings, especially white on rose (p22phox), are hard to read.

Response: Thank you very much for these suggestions. I revised the figure legends and spelled out all abbreviations. The white labeling of p22phox was truly hard to read. I moved it a little bit and colored it black for better reading (also in the other figures).

In figure 2, the two distinct blue arrows seem to refer to two distinct processes, instead of a continuum of the same process. Also, red on dark blue is hard to read. 

Response: True! I changed the figure and tried to depict it now as one continuous arrow. I also changed the color of the text to white for better reading.

Figure 3 is cut on the right side. Panel B is not mentioned in the figure legend. Would processes 1) and 2) eventually converge in augmenting nox3 protein at the plasma membrane level? This is not shown in figure 3. 

Response: The cut-off was introduced during the uploading process. Unfortunately, I only discovered the error after 2 days. I corrected the position of all figures in this context. Indeed, I forgot to mention Panel B in the Figure legend, which includes the sub sections 1) and 2). This was corrected. Process 2), which is indeed an up-regulation of Nox3 mRNA, would subsequently lead to increased Nox3 protein levels. I depicted this matter by adding more Nox3 proteins in comparison to panel A). For clarity, I re-arranged panel B and added red dashed arrows which connect the increased mRNA levels with the increased protein levels of Nox3. I also increased the labeling of the Nox3 enzymes for better reading.

Figure 4, text in panel c is too small

Response: I increased all texts to the maximum of the possible size with exception of the molecules far on the right. I was also concerned about this issue during the first upload. However, since the manuscript itself, the PDF file and, more importantly the online version of the, hopefully, published article all allow high quality magnification by the reader; I would like to leave the text size as it is, since further increase of the molecules would lead to crowding of the panel c in this figure.

Figure 5, "Tunel" should be spelled "Tunnel"

Response: Corrected.

Minor comments

- there are sometimes numbers in the main text that appear to be unformatted references or publication years, please double-check lines 147 and 455

Response: Yes this is true. It already came to my attention. I carefully checked the whole manuscript for such errors and corrected them.

- Period (.) missing at lines 191 and 386

Response: Corrected. Thank you.

- Lines 1901-1902, references 915 and 916 do not belong to Yasukoa, please revise. 

Response: This is true. However, the references refer to articles about IGFBP-5 in general. Indeed the sentence is misleading and was rephrased.

- Comments on language/grammar/style are given in the box below. 

I want to express my high gratitude for bearing with my writing and spelling errors. This is the first article that I completely conceived, wrote and financed by myself. Accordingly, for one person it is highly possible that errors in spelling, phrasing and punctuation occur in a lengthy article. Again, I deeply thank the reviewer for the careful checking and suggested corrections.

Comments on the Quality of English Language

Some sentences are really long and hard to follow. Please, revise the abstract and conclusions for this issue. The use of parentheses is discouraged. Please revise the sentence in the abstract: "the discovery .... was a surprising discovery". 

Response: This is indeed a general issue concerning the whole manuscript. I carefully revised the text including the abstract and the conclusion and did my best to “cut” and “streamline” the text in general.

Some sentences are rather unclear and should be rephrased:

lines 495-497

Response: Rephrased.

lines 570-572

Response: Rephrased

lines 899-902

Response: Rewritten and supplemented.

line 643, please correct the repetition

Response: The repetition was deleted. Thank you very much.

line 646, please correct as follows: Nox3 is expressed in avian liver

Response: Corrected.

Line 783: amnichorions > amniochorions

Response: Corrected. Thank you.

Line 2210 and 2404, dervied > derived

Response: Thank you very much. I also corrected 19 similar misspellings.

Line 2285, meidated > mediated

Response: Corrected.

Line 2316, disbalanced > unbalanced

Response: Corrected.

Line 2323, frequency > frequencies 

Response: Corrected.

Line 2343, progress > progresses

Response: Corrected.

Line 2362, an > a 

Response: Corrected.

Line 2366, bio marker > biomarker

Response: Thank you. This was also corrected in another text passage.

Line 2388, lead do > leads to

Response: Corrected.

Line 2448, mile stone > milestone

Response: Corrected.

Round 2

Reviewer 1 Report

This comprehensive review paper has been improved. Editing is still required (e.g., L18 Abstract: "summarizes it’s the discovery") prior to publication. 

Some additional editing of English language will be required (e.g., L18 of Abstract).

Reviewer 2 Report

I appreciate the author fulfilling basically all of my requirements and suggestions. However, please make sure that the text corrections do not generate additional issues, for example, in the abstract: "surprisinge" or "it´s the discovery".

I am sure these misspellings can be fixed at the level of proof correction.

none